# Nested Slice Sampling: Vectorized Nested Sampling for GPU-Accelerated Inference

**David Yallup**[*][†]                                                                  *dy297@cam.ac.uk*
*Kavli Institute for Cosmology Cambridge and Institute of Astronomy, University of Cambridge*

**Namu Kroupa**[*]                                                                      *nk544@cam.ac.uk*
*Cavendish Laboratory and Department of Engineering, University of Cambridge*

**Will Handley**                                                                        *wh260@cam.ac.uk*
*Kavli Institute for Cosmology Cambridge and Institute of Astronomy, University of Cambridge*

**Reviewed on OpenReview:** *https://openreview.net/forum?id=5mF2eRl3gt*

## Abstract

Model comparison and calibrated uncertainty quantification often require integrating over parameters, but scalable inference can be challenging for complex, multimodal targets. Nested Sampling is a robust alternative to standard MCMC, yet its typically sequential structure and hard constraints make efficient accelerator implementations difficult. This paper introduces Nested Slice Sampling (NSS), a GPU-friendly, vectorized formulation of Nested Sampling that uses Hit-and-Run Slice Sampling for constrained updates. A tuning analysis yields a simple near-optimal rule for setting the slice width, improving high-dimensional behavior and making per-step compute more predictable for parallel execution. Experiments on challenging synthetic targets, high dimensional Bayesian inference, and Gaussian process hyperparameter marginalization show that NSS maintains accurate evidence estimates and high-quality posterior samples, and is particularly robust on difficult multimodal problems where current state-of-the-art methods such as tempered SMC baselines can struggle. An open-source implementation is released to facilitate adoption and reproducibility.

## 1 Introduction

Sampling from unnormalized probability distributions is a foundational task in machine learning and Bayesian computation. We consider targets of the form

$$P_\beta(x) = \frac{\exp(-\beta E(x))\,\Pi(x)}{Z(\beta)}\,, \qquad \beta \in [0, 1], \tag{1}$$

where $x$ are parameters of interest, $E$ is a scalar energy (e.g. negative log-likelihood up to an additive constant), and $\Pi$ is a reference density (often a prior). The normalizing constant $Z(\beta)$ is the partition function, given by

$$Z(\beta) = \int \exp(-\beta E(x))\,\Pi(x)\,dx\,. \tag{2}$$

The parameter $\beta$ can be interpreted as an inverse-temperature/tempering parameter: at $\beta = 0$ the target reduces to $\Pi$, and at $\beta = 1$ it corresponds to the distribution of interest. When $\exp(-E(x))$ is identified as a

---

[*]Equal contribution.
[†]Corresponding author.

likelihood and $\Pi$ as a prior, $Z(1)$ is the marginal likelihood (evidence) in Bayesian inference (Murphy, 2022). Estimating this quantity enables Bayesian model comparison (MacKay, 2002). While MCMC methods (Geyer, 1992) are widely used for posterior inference, most standard MCMC algorithms do not directly estimate $Z$, motivating specialised approaches.

Nested Sampling was introduced as a generic meta-algorithm for estimating the marginal likelihood (Skilling, 2004). At a high level, it is related to particle Monte Carlo methods such as Sequential Monte Carlo (Doucet et al., 2001), in that it evolves a population of particles through intermediate distributions bridging between $\Pi$ and the target. NS saw rapid adoption in the physical sciences, first in statistical physics (Murray et al., 2005) and cosmology (Mukherjee et al., 2006), and subsequently in gravitational-wave astronomy (Veitch & Vecchio, 2008), particle physics (Trotta et al., 2008), and materials science (Pártay et al., 2010), where it is valued for robustness on complex and multimodal distributions (Ashton et al., 2022). However, a single canonical implementation strategy has not emerged, leading to a variety of implementations with differing internal mechanics. Various components have been explored in the intervening years, including gradient-guided variants (Feroz & Skilling, 2013; Cai et al., 2022; Lemos et al., 2023) and implementations in autodiff-compatible frameworks (Albert, 2020; Anau Montel et al., 2024). Among these, PolyChord (Handley et al., 2015) demonstrated that slice sampling within NS scales to moderately high dimensions, establishing the approach we build on. However, PolyChord and other mature NS codes (e.g. MultiNest (Feroz & Hobson, 2008), UltraNest (Buchner, 2021b)) carry features designed for CPU architectures—clustering heuristics, priors implemented as unit-hypercube transforms, dynamic live-point populations, and MPI-based parallelism—that are poorly suited to accelerator hardware.

In this work, we address this gap by providing a clear, efficient, and modern implementation of Nested Sampling suitable for both physical science and ML practitioners. While many of the individual ingredients (nested sampling, slice sampling, constrained MCMC) are known, existing formulations and implementations do not automatically translate into an efficient *accelerator* algorithm: classical NS is often viewed as sequential, and constrained inner kernels typically have data-dependent control flow that maps poorly to SIMD hardware. Our contributions are therefore targeted at making NS *practically GPU-viable* end-to-end, by exposing parallelism in the outer loop and stabilizing the inner-kernel compute so that batched execution is effective. Our key contributions are as follows:

(i) We develop a massively parallel implementation of Nested Sampling that expresses the full update (energy evaluation, thresholding, resampling, and mutation) via vectorized execution targeting modern GPU hardware (section 3).

(ii) We show that using Hit-and-Run Slice Sampling as the constrained inner kernel, while discarding clustering heuristics, yields a robust and performant algorithm, and we provide a principled tuning analysis for the slice width. Crucially, operating near the derived optimum makes per-step costs concentrate, mitigating SIMD/warp divergence and enabling effective batching on accelerators (section 3.2).

(iii) We provide an empirical comparison against strong adaptive tempered SMC baselines using optimized implementations of both approaches (section 4). As well as an implementation that directly aligns in details and form with established SMC patterns. We demonstrate performance on challenging synthetic problems, high dimensional Bayesian inference and ML inference tasks.

Our implementation, Nested Slice Sampling (NSS) and the underlying nested sampling framework, is designed to be composable within modern probabilistic programming ecosystems and is available as open-source software.

## 2 Background and Theoretical Framework

Nested Sampling (NS) estimates the normalizing constant $Z(\beta) = \int \exp(-\beta E(x)) \, \Pi(x) \, dx$ and produces weighted samples for posterior expectations. A useful way to view NS is as the combination of (i) an *outer* scheme that turns evidence estimation into a one-dimensional quadrature with automatically chosen levels,

and (ii) an *inner constrained sampling* routine that approximately draws from $\Pi(x) \mathbf{1}_{\{E(x)<E_{\min}\}}$. This section reviews the background required for the outer NS construction (section 2.1), then (in section 2.2) discusses existing approaches to constrained sampling and motivates slice-based updates as our default inner kernel. Connections to particle methods and bridging distributions are summarized in section 2.3.

## 2.1 Nested Sampling

We follow the formulation of Skilling (2006). Figure 1 illustrates the core idea: NS iteratively restricts the reference density to lower and lower energy (higher likelihood) regions, producing a sequence of nested constraint sets and associated quadrature weights.

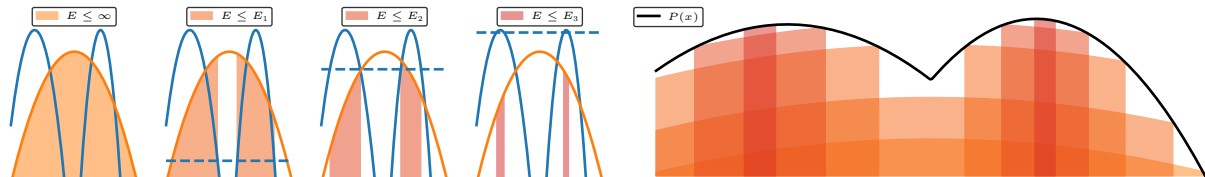

Figure 1: Nested Sampling shrinks a reference density by successive energy constraints (left), yielding a quadrature approximation of the normalizing constant (right).

Let $\Pi(x)$ denote a reference density (typically a prior) and $E(x)$ an energy (e.g. minus log-likelihood). Define the *prior volume*

$$X(E) := \int \Pi(x) \mathbf{1}_{\{E(x)<E\}} \, dx \in [0,1], \tag{3}$$

and let $E(X)$ be its inverse. Then the normalizing constant can be written as the one-dimensional integral

$$Z(\beta) = \int_0^1 \exp\big(-\beta \, E(X)\big) \, dX, \tag{4}$$

and NS approximates this integral by a quadrature over a decreasing sequence of energy levels $E_1 > E_2 > \cdots > E_n$ (equivalently, shrinking volumes $1 = X_0 > X_1 > \cdots > X_n$):

$$\widehat{Z}(\beta) = \sum_{i=1}^n \exp(-\beta E_i) \Delta X_i, \qquad \Delta X_i := X_{i-1} - X_i. \tag{5}$$

[1] Once the pairs $\{(E_i, \Delta X_i)\}$ are available, the same quadrature can be evaluated for different $\beta$ by reweighting. Operationally, NS maintains $m$ *live* particles intended to approximate draws from the constrained reference distribution

$$\Pi_{E_i}(x) \propto \Pi(x) \mathbf{1}_{\{E(x)<E_i\}}. \tag{6}$$

At iteration $i$, the algorithm identifies the $k$ live particles with largest energies, records them as *dead* points at level $E_i$, and replaces them with $k$ new particles approximately drawn from $\Pi_{E_i}$. Under the idealized assumption that live points are i.i.d. from $\Pi_{E_{i-1}}$, the shrinkage factor $t_i := X_i/X_{i-1}$ is an order statistic, giving the standard approximation

$$\mathbb{E}[\log t_i] \approx -k/m \qquad \Longrightarrow \qquad \Delta \log X_i \approx -k/m. \tag{7}$$

It is possible to derive errors on this approximation; details are given in section D.2. The resulting estimator and its error properties have been studied extensively (Chopin & Robert, 2010; Keeton, 2011; Fowlie et al., 2023), including extensions to dynamic live-set sizes (Higson et al., 2018) and likelihood plateaus (Fowlie et al., 2021). When $k > 1$ points are removed per iteration, rather than treating the batch as a single contraction

---

[1]Other quadrature rules (e.g. trapezoidal or Simpson) can be substituted with varying degrees of accuracy; in practice the evidence error is typically dominated by the stochastic estimation of the $X_i$ rather than the choice of quadrature.

step, we *unroll* it into $k$ sequential single-death events with effective live counts $m, m-1, \ldots, m-k+1$ (Fowlie et al., 2021), so each removal contributes its own order-statistic volume compression (section D.2.1). In practice, NS is terminated when an upper bound on the remaining contribution from the live set is below a user-specified tolerance.

The outer NS construction above is agnostic to how constrained draws from $\Pi_{E_i}$ are generated; pseudocode for this outer kernel is given in algorithm 1. The main practical bottleneck is therefore *constrained sampling* as the contour shrinks. We focus on this in section 2.2.

## 2.2 Constrained sampling

Nested Sampling (NS) reduces evidence estimation to a sequence of constrained sampling problems. At a given likelihood/energy threshold $E_{\min}$, the outer NS update requires approximate samples from the reference density $\Pi$ restricted to a hard constraint (Buchner, 2021a), as expressed in equation 6. As NS progresses, the feasible region typically becomes small, anisotropic, and may be disconnected. In moderate-to-high dimensions the boundary of $\{E(x) < E_{\min}\}$ becomes a dominant geometric feature, so the choice of constrained sampler largely determines both the efficiency and the practical correctness of NS.

Two broad families of constrained updates are commonly used in NS. First, *region-based rejection sampling* constructs a proposal distribution (often informed by the current live set) and draws candidates until the constraint is satisfied. This is effective in low-to-moderate dimensions and underlies several popular implementations (e.g. ellipsoidal bounds and related schemes) (Feroz & Hobson, 2008; Buchner, 2021b). More expressive proposals can be learned from the live points, including approaches based on normalizing flows or other ML density models (Lange, 2023; Williams et al., 2021; Torrado et al., 2023). However, rejection-based strategies typically degrade as dimension grows unless the proposal accurately matches the constrained geometry.

Second, *constrained MCMC* applies a Markov transition kernel that leaves $\Pi_{E_{\min}}$ invariant. This avoids explicit acceptance by a global proposal envelope, but introduces a different set of requirements: the kernel must mix within a compact, evolving, and potentially sharp-bounded region. Simple random-walk proposals can be dominated by boundary rejections, while naive reflection-based constrained dynamics can be sensitive to discretization choices and, in some implementations, exhibit dimension-dependent artifacts (Kroupa et al., 2025). In section F (fig. 13) we empirically compare several constrained kernels on a controlled family of problems; slice-based constrained updates remain accurate across dimensions in this stress test, while several reflection-based alternatives deviate as dimension increases (Kroupa et al., 2025).

Motivated by these considerations, we use *Hit-and-Run Slice Sampling* (HRSS) as the default inner kernel for constrained propagation. HRSS is a variant of slice sampling (Neal, 2003) specialised to high-dimensional updates: it chooses a random direction (hit-and-run) (Smith, 1984) and then performs an exact one-dimensional slice update along the resulting chord of the constrained set. The algorithm with recursive bound shrinkage was formalised by Kiatsupaibul et al. (2011), and scales well with dimensionality (Collins et al., 2013). This construction is well-suited to NS because it treats the hard constraint in equation 6 directly (by assigning zero density outside the feasible set), and inherits favourable dimension-dependent behaviour relative to naive constrained random walks in a range of settings (Rudolf & Ullrich, 2018; Power et al., 2024). We use a simple Gaussian family of direction proposals as a fast, robust baseline (Handley et al., 2015; Buchner, 2023), and note that richer direction/proposal mechanisms are compatible with the same framework (Moss, 2020). Both nested sampling and constrained (slice) updates exhibit adaptive, data-dependent control flow—e.g., variable iteration counts and on-the-fly stopping rules—so they are often viewed as ill-suited to GPU vectorization. In section 3 we address this and show that, with appropriate tuning and a batched nested sampling kernel, the per-step cost concentrates sufficiently to be well suited to vectorized execution, which we take forward to empirical validation in section 4.

## 2.3 Particle Monte Carlo and bridging distributions

Nested Sampling (NS) belongs to a broader family of *particle Monte Carlo* methods that propagate a population of particles through a sequence of intermediate ("bridging") distributions between an easy-to-

sample reference $\Pi$ and a target of interest, while accumulating an estimate of the normalizing constant (Del Moral et al., 2006; Chopin, 2002). The most prominent formalism in this family is Sequential Monte Carlo (SMC), which has become a standard tool for model comparison (Zhou et al., 2015) and admits highly parallel implementations (Murray et al., 2015). A central design choice in SMC is the *path*: the practitioner specifies (or adaptively constructs) the bridging distributions $\{\pi_t\}$, for example via tempering $\pi_\beta(x) \propto \Pi(x) \exp(-\beta E(x))$ or other problem-specific interpolations, with step sizes often chosen to control weight degeneracy (Fearnhead & Taylor, 2010).

NS shares structural similarities with rare-event SMC methods (Cérou et al., 2012), and can formally be viewed within the SMC framework (Salomone et al., 2024), but differs in important respects. Rather than selecting an externally parameterised annealing coordinate, NS induces a constraint-based path $\pi_t^{\mathrm{NS}}(x) \propto \Pi(x) \mathbf{1}_{\{E(x) < E_t\}}$, where the levels $\{E_t\}$ are set by order statistics of the live set. Crucially, NS estimates the normalizing constant via a one-dimensional quadrature over probabilistically estimated prior volumes (section 2.1), rather than via importance weights as in standard SMC. While the formal SMC unification is mathematically valid, it represents a singular case within the broader SMC framework: the probabilistic volume estimation that underpins NS has no natural analogue in standard SMC and is responsible for many of its distinctive properties, including the ability to reweight samples across temperatures and the characteristic geometric compression schedule. Consequently, NS follows a characteristic sequence of *nested likelihood regions* whose geometry is determined by the model and data (illustrated in fig. 1), yielding a bridging family that is essentially fixed by the NS update rule rather than chosen by the user. This distinction is practically important: the NS path turns inference into repeated *constrained sampling* problems, which places different demands on the inner MCMC kernel than unconstrained tempering paths. In our experiments we use adaptive tempered SMC as a baseline; pseudocode for this outer kernel is given in algorithm 2 and detailed configuration in section A.1.

## 3 Algorithm and implementation

Having established the theoretical background, we now present a generic vectorized Nested Sampling framework and a specific performant instantiation, Nested Slice Sampling (NSS). We implement both in the `blackjax` library (Cabezas et al., 2024) within the `jax` ecosystem (Bradbury et al., 2018), closely following the SMC abstractions of Chopin & Papaspiliopoulos (2020). A practical motivation for this choice is that the dominant cost in NS is repeated evaluation of the energy $E(x)$ (and hence the likelihood) over many particles. By expressing both the outer NS update and the inner constrained update as composable `jax` functions, we can batch energy evaluations across particles and compile the full update to CPU/GPU/TPU backends via JIT.

Our implementation mirrors the decomposition in section 2: an NS *outer kernel* that maintains the live set and performs evidence bookkeeping, and a pluggable *constrained update* kernel used to (approximately) sample from the constrained reference distribution $\Pi(x)\mathbf{1}_{\{E(x) < E_{\min}\}}$ at the current threshold. Section 3.1 describes the parallel outer kernel and its replacement interface; section 3.2 then specifies the constrained update and tuning choices that form NSS. Compared to typical NS implementations that parallelize primarily across CPU cores (e.g. via MPI), this enables a complementary form of massive parallelism based on batched likelihood evaluation and fused kernel execution.

### 3.1 Parallel Nested Sampling

We implement a batched NS outer loop that removes and replaces $k$ live points per iteration (Henderson & Goggans, 2014). Let $m$ denote the number of live particles and $k \in \{1, \ldots, m-1\}$ the batch size. Each outer iteration performs:

(i) **Reweight (delete):** evaluate energies $\{E(x_i)\}_{i=1}^m$, identify the $k$ worst live points, and set the batch threshold $E_{\mathrm{batch}}$ to the $k$-th worst energy. The deleted points are recorded as *dead* samples.

(ii) **Resample (duplicate):** select $k$ parent indices from the surviving $m-k$ live points (with replacement) and duplicate those states. This corresponds to multinomial resampling with binary weights. $w_i \propto \mathbf{1}\{E(x_i) < E_{\mathrm{batch}}\}$, i.e. uniform over survivors.

(iii) **Mutate (constrained update):** apply a user-specified constrained update operator targeting $\Pi(x)\mathbf{1}_{\{E(x)<E_{\text{batch}}\}}$ to each duplicated parent to obtain $k$ new live points satisfying the same constraint.

(iv) **Replace:** insert the $k$ new points into the live set.

The ratio $k/m$ controls the expected compression per iteration (hence the number of outer iterations), while $k$ also sets the exposed parallelism. In contrast to many SMC schemes, this decouples the degree of parallel mutation from the live-set resolution $m$ – an advantageous trade off for high memory footprint likelihoods. Resampling is a user defined step and simple extensions such as applying updates to all $m$ particles are also supported.

**Evidence bookkeeping.** The outer kernel stores dead-point energies (and any auxiliary statistics required for posterior reconstruction). For batched deletion, the expected volume contraction depends on $(m, k)$ via order statistics (section 2.1). In experiments we propagate the corresponding "geometric" uncertainty post hoc by shrinkage simulation (section D.2.2, using the within-batch unrolling in section D.2.1).

**Vectorised execution.** All operations in the outer update—energy evaluation, thresholding, indexing, and resampling— are implemented as array programs and can be JIT-compiled. In particular, energies for the full live set are evaluated in a single batched call to $E(x)$, and the $k$ constrained updates are applied in parallel via the replacement interface. We discuss the trade-offs of adaptive versus fixed-schedule execution in section A.2.

**Replacement strategies and adaptation.** The outer kernel is agnostic to how replacements are generated. Our default strategy, `from_mcmc`, applies a constrained MCMC kernel (HRSS in NSS, section 3.2) starting from each duplicated parent. The replacement kernels are synchronised at the likelihood level: all $k$ chains evaluate the likelihood in lockstep as a single batched operation. This is the first performant implementation of this form of parallelism in the nested sampling literature. To illustrate the flexibility of the interface we also implement a constrained Gaussian random-walk baseline using standard `blackjax` components. We additionally support an *adaptive* mode in which inner-kernel parameters (e.g. a conditioning metric estimated from the live set) are updated once per outer iteration, in the same spirit as SMC inner-kernel tuning utilities (Chopin & Papaspiliopoulos, 2020).

## 3.2 Nested Slice Sampling (NSS)

Nested Slice Sampling instantiates the replacement interface of section 3.1 using Hit-and-Run Slice Sampling (HRSS) (see section A.6 for practical implementation details) as the constrained update kernel. Following the batch deletion step, we resample (duplicate) $k$ surviving particles as parents. Each replacement starts from a duplicated survivor and applies a short HRSS chain to obtain a new (decorrelated) particle satisfying the constraint. Key design choices are detailed below.

**Hit-and-Run Slice Sampling (HRSS).** In NSS we generate new live points by applying short HRSS chains targeting the constrained density $\tilde{\pi}(x) \propto \Pi(x)\mathbf{1}_{\{E(x)<E_{\min}\}}$. Given a current state $x$, HRSS samples a random direction (velocity) $v$ and performs a one-dimensional slice update along the line $\{x + tv : t \in \mathbb{R}\}$ using stepping-out and shrinkage (Neal, 2003). Operationally, points outside the hard constraint (or outside the support of $\Pi$) are treated as having $\log \tilde{\pi}(x) = -\infty$, so stepping-out automatically discovers a finite chord within the feasible set whenever one exists. We run $p$ HRSS steps per replacement; we use $p = d$ as a default, with an ablation study in section B.7.

**Direction metric.** At each HRSS step, the velocity is randomized in a Mahalanobis geometry by sampling $\tilde{d} \sim \mathcal{N}(0, M^{-1})$ and setting $v = \tilde{d}/\|\tilde{d}\|$, where $M$ is a positive definite conditioning matrix. In practice $M$ is updated once per outer iteration from the empirical covariance of the current live set (with standard regularisation), providing inexpensive approximate whitening as the NS contour evolves. For very high dimensions it is likely to be beneficial to restrict this to be diagonal, or otherwise modify this to exploit

any structure in the model. This plays a role analogous to mass-matrix adaptation in HMC-based kernels (Buchholz et al., 2020).

**GPU scaling and practical performance.** To the best of our knowledge, NSS provides the only fully vectorized implementation of Nested Sampling, in which the entire outer kernel—including batched constrained mutation—maps to parallel execution on GPU hardware. Figure 2 demonstrates this on the GermanCredit logistic regression problem ($d$=25, $m$=1000), running on NVIDIA A100 hardware, with more detailed experimental results following in section 4.2.

We conduct two scaling experiments here: fixing the population size $m$ while varying the batch size $k$, and fixing the ratio of batch to population size $k/m$ while varying the population size $m$. In the first experiment, we observe nearly linear runtime speedup with $k$, with evidence uncertainty stable for $k/m \leq 0.5$ and degrading sharply beyond as shown in the rightmost panel. In the second experiment, we observe nearly flat runtime scaling as the live population is increased to 4000 points. In both cases we show the ideal *embarrassingly parallel* scaling and the worst-case scenario; we find that although perfect parallelism is not achieved, the implementation still provides significant performance benefits. Extended ablations including additional batch sizes, population sizes, and Gaussian process results are provided in section C. Comparisons with existing NS codes are given in section C.1. Based on these results we recommend setting $m$ as large as the available hardware allows (since additional particles are nearly free for fixed compression rate $k/m$), with $k/m \approx 0.1$ as a conservative default and $k/m \leq 0.5$ as a practical upper bound before evidence accuracy degrades.

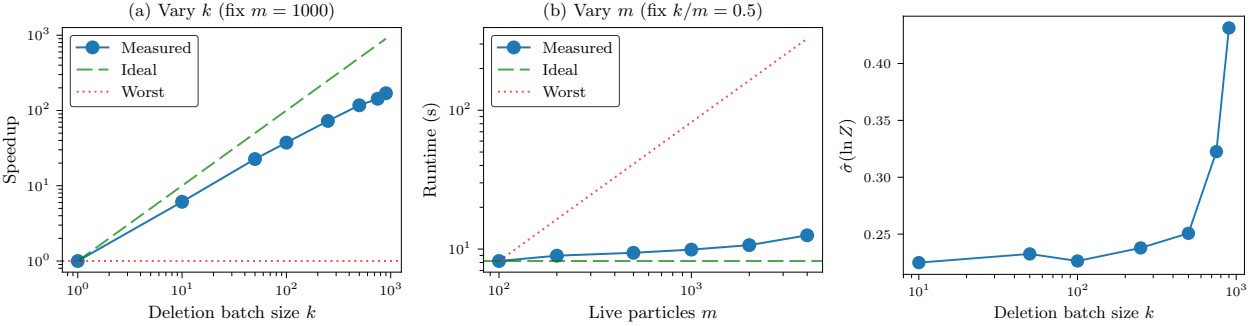

Figure 2: GPU scaling on GermanCredit logistic regression ($d$=25, $m$=1000). Left: runtime and $\ln Z$ accuracy vs. deletion batch size $k$, showing nearly linear runtime speedup with $k$. Right: evidence uncertainty $\hat{\sigma}(\ln Z)$ vs. $k$, confirming that accuracy is stable for $k/m \leq 0.5$ and degrades sharply beyond.

### 3.3 Optimal slice width tuning

Alongside the implementation, we analyse the theoretical properties of the constrained updates employed in NSS and provide optimal tuning recommendations.

**Computational cost of slice sampling.** Section E derives the computational cost for the stepping-out and shrinkage procedures and the resulting optimal tuning. Conditional on the (random) chord length $\ell$ induced by the current point, direction, and constrained region, the expected number of likelihood evaluations for one HRSS update is

$$\mathbb{E}[N_{\text{evals}} \mid \ell] = \frac{\ell}{w} + 1 + 2\phi\left(\frac{w}{\ell}\right), \tag{8}$$

for an explicit function $\phi$ (theorem 1). Importantly, $\mathbb{E}[N_{\text{evals}} \mid \ell]$ grows as $\frac{\ell}{w}$ as $w \to 0$ and logarithmically as $\ln\left(1 + \frac{w}{\ell}\right)$ when $\frac{w}{\ell} \to \infty$. This matches the intuition. For small $\frac{w}{\ell}$, the chord length is always underestimated, and we require $\frac{\ell}{w}$ stepping-out iterations to cover the chord. For large $\frac{w}{\ell}$, we expect exponential contraction of the shrinkage step as it is effectively a randomised bisection. This then implies that the number of likelihood evaluations scales logarithmically with the slice width $w$. Minimising eq. (8) yields a unique optimum. For a

fixed (non-stochastic) $\ell$, this gives $w_* \approx 1.36\,\ell$ (theorem 2). Notably, this value is larger than $\ell$, which stems from the fact that the logarithmic growth of the cost favours overestimation of the chord length.

**Ellipsoidal level sets.**   For ellipsoidal level sets $E_d = \{x : x^\top A_d x \leq 1\}$ in dimension $d$, and assuming uniform initialisation with such level sets with an isotropic direction law of slice sampling, the chord length is a random variable which concentrates in high dimensions (lemma 10). From this, one obtains the explicit high-dimensional scaling

$$w_* \;=\; 4\kappa_\infty\sqrt{\frac{2}{\pi\mu_d d}}\,[1 + o(1)], \qquad \mu_d := \frac{1}{d}\mathrm{Tr}(A_d), \tag{9}$$

with a universal constant $\kappa_\infty \approx 1.3035$ (theorem 3). In particular, the optimal step size decreases like $d^{-1/2}$, and the leading dependence on anisotropy enters primarily through $\mu_d$ rather than the full spectrum.

Another way to state the observation of the $d^{-1/2}$-scaling is that the prefactor $\propto \mu_d^{-1/2}$ is determined by the mean eigenvalue $\mu_d$ of the matrix $A_d$. The anisotropy of the ellipsoid, i.e. the spread of the eigenvalues, only enters through negligible higher-order corrections. Correspondingly, we may say that the anisotropy is "washed out" in high dimensions. The ellipsoid with semi-axes $(s_1, \ldots, s_d)$, such that $A_d = \mathrm{diag}(\frac{1}{s_1^2}, \ldots, \frac{1}{s_d^2})$, behaves like a ball with effective radius $R_{\mathrm{eff}}$ given by the harmonic mean,

$$\frac{1}{R_{\mathrm{eff}}^2} := \frac{1}{d}\sum_{i=1}^d \frac{1}{s_i^2}. \tag{10}$$

Note that this is a global statement, since the expectations were taken over the entire ellipsoid, and should not be taken to apply, for example, to a Markov chain exploring an ellipsoid locally.

As a further consequence, the optimal slice width is maximised in the isotropic case. This motivates the whitening strategy used in practice: By sampling directions in a Mahalanobis metric based on the live-particle covariance, we aim to keep the constrained region approximately isotropic in the proposal geometry, so that a dimension-aware width rule remains stable as the NS contour evolves.

**Isotropic level sets.**   In the case where the level set is an isotropic ball, the above expression may be specialised (corollary 1) and the optimal scaling of $w_*$ remains $d^{-1/2}$. We remark that the $d^{-1/2}$-scaling is also exhibited by the standard deviation of the uniform distribution in a ball in high dimensions, since the covariance matrix of a ball of radius $R$ is $\frac{R^2}{d+2}I$. In low dimensions, however, we expect that the dominant chord length is on the scale of the diameter. Overall, this motivates a method for tuning $w_*$ for all $d$: In low dimensions, we compute the covariance matrix of the (uniformly distributed) live point population and rescale the slice width to the diameter of the whitened ball. In high dimensions, the asymptotic value of $w_*$ above applies, so that we can use the raw entries of the covariance matrix without a dimension-dependent rescaling.

**Numerical tests.**   We validate the theoretical predictions above numerically. Details are given in section E.2. First, the theoretical expressions for both the cost and the optimal width agree with numerical experiments (fig. 3). Second, the variability of the per-step cost remains modest at the optimum (section E.2.2; see fig. 12), which explains why vectorising many short constrained chains can be effective despite the adaptive control flow of slice sampling (see section A.7 for an empirical comparison with random walks). These two empirical consequences of this analysis are leveraged by NSS.

**Remarks on assumptions.**   In our derivation, it was assumed that the eigenvalues of $A_d$ are bounded away from 0 and $\infty$ (lemma 9). In practice, this is a weak assumption, especially for statistical models used in the physical sciences where nested sampling is widely applied (Ashton et al., 2022). By the Bernstein–von Mises theorem (Van der Vaart, 2000, Section 10.2), under standard regularity conditions and in the large-data limit, the posterior is locally well approximated by a Gaussian around a regular mode. This provides asymptotic intuition for modelling the constrained region as approximately ellipsoidal. In highly nonlinear targets or landscapes with widely separated modes, these assumptions may break down. However, we show empirically in section 4 that in such cases, slice sampling remains robust. In particular, the variance of the chain length

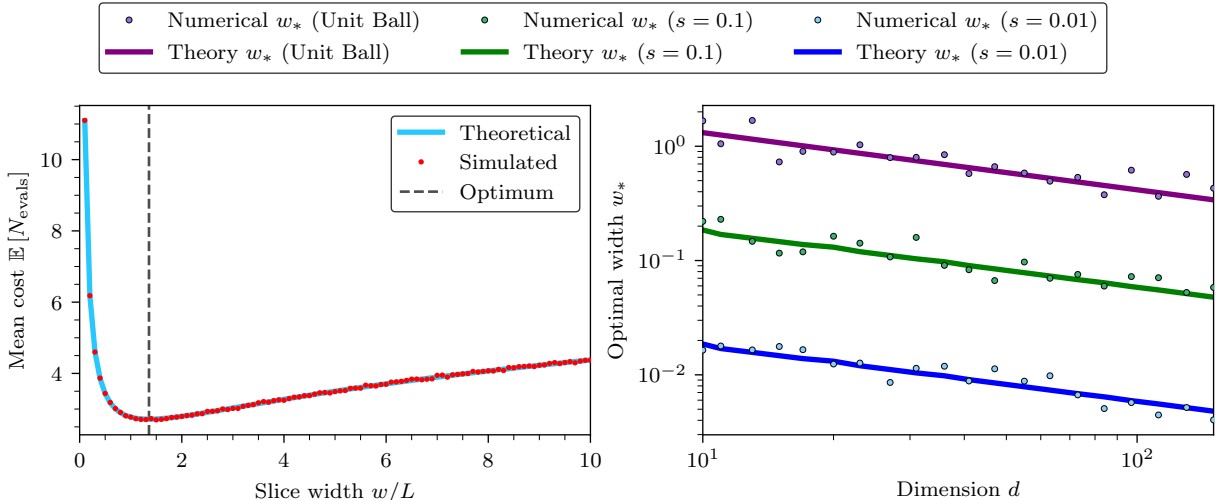

Figure 3: Validation of theory against numerics. **Left:** The theoretical prediction (eq. (125)) is compared against the cost obtained from numerical simulation of slice sampling. The optimum is given by eq. (159). **Right:** The theoretically obtained optimal slice width $w_*$ (eq. (165)) is compared against the numerically calculated optimal $w_*$, obtained by scanning through a range of values of $w_*$ and simulating slice sampling to compute the cost.

may become larger since the number of slice sampling steps depends on the local geometry, which is expected to lower the efficiency. However, this inefficiency can be measured and serve as a practical indicator that the theoretical assumptions are becoming less reliable.

### 3.4 SMC and MCMC baselines

We benchmark NSS against particle and MCMC baselines implemented in the same `blackjax` stack. For particle methods we use adaptive tempered SMC with an ESS-controlled temperature schedule (Fearnhead & Taylor, 2010), and compare multiple mutation kernels: HMC (SMC-HMC), random-walk Metropolis–Hastings (SMC-RW), slice sampling (SMC-SS), and independent Metropolis–Hastings (SMC-IRMH). For posterior-only comparisons (where evidence estimation is not required) we additionally run standalone slice sampling (SS) and NUTS. Full configuration details are given in sections A.4 and A.5.

## 4 Experiments

In this section we review the performance of NSS on synthetic benchmarks, Bayesian inference problems from Inference Gym (Sountsov et al., 2020), which serves a similar aim to posteriordb (Magnusson et al., 2025) with overlapping problems and construction, and an ML application to Gaussian Process hyperparameter marginalization. We compare against adaptive tempered SMC with several mutation kernels (SMC-RW, SMC-IRMH, SMC-SS, SMC-HMC) and, where appropriate, posterior-only baselines (SS and NUTS). We evaluate posterior quality using Maximum Mean Discrepancy (MMD) (Gretton et al., 2012) and the sliced 2-Wasserstein distance ($W_2$) against reference samples (analytic where available, otherwise long-run MCMC; see section D.1). For problems without analytic ground truth, all NSS posterior estimates are broadly consistent with long-run NUTS. For efficiency, we report ESS normalised by wall-clock time and by the number of energy evaluations. For the GP Regression task we evaluate performance using a held out test set comprised of the last 30% of the data, measuring the log-likelihood of the test set under the posterior predictive distribution averaged over hyperparameter posterior samples. All GPU experiments were run on a Google Cloud A2 Standard instance equipped with $1\times$ NVIDIA A100 GPU (40 GB HBM2), 12 vCPUs, and 85 GB host memory. We run experiments in single precision (float32) unless otherwise noted. Extended results and more experimental details are provided in section B.

For evidence estimation, NSS reports geometric shrinkage uncertainties from volume simulations (section D.2), which quantify the stochastic error from the NS compression process. SMC normalizing-constant estimators are unbiased with variance $O(1/m)$ under standard conditions (Beskos et al., 2011); at $m = 1000$ particles (the value we run most experiments at), this statistical error is negligible compared to the mixing uncertainty that arises when mutation kernels fail to fully explore the target. We report all metrics, including $\ln Z$, with run-to-run error bars from 5 independent seeds; for SMC the dominant source of error is mixing failure rather than within-run variance (see section B.6).

### 4.1 Synthetic Benchmarks

We evaluate NSS on synthetic benchmarks designed to probe sampler performance on pathological features common in inference problems: pronounced multimodality (Mixture of Gaussians in $2d$ with 40 modes and in $10d$ with 5 modes and randomized covariances) and hierarchical structures (Neal's funnel). These targets test both posterior sampling quality and evidence estimation accuracy. We compare against adaptive tempered SMC baselines with random-walk, independent, slice-sampling, and HMC mutation kernels (SMC-RW/IRMH/SS/HMC). Results are summarized in table 1, with full problem descriptions, visualizations, and extended baselines in section B.

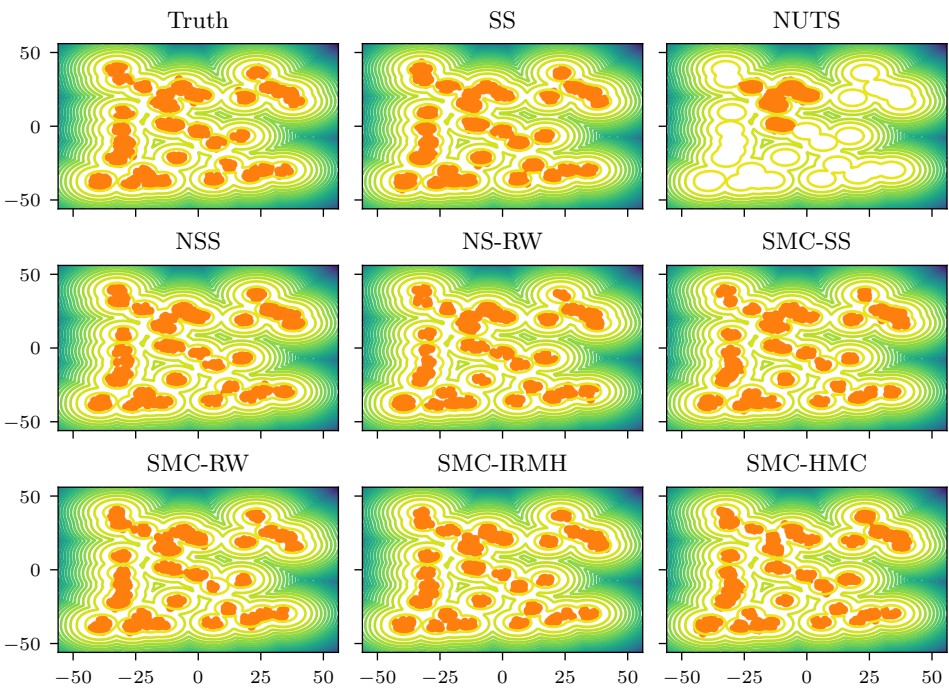

Figure 4: Posterior samples for a mixture of 40 bivariate Gaussians on a bounded uniform prior. NSS (orange) recovers all modes with correct weights. This $2d$ example, despite its low dimensionality, challenges many sampling methods due to the large number of well-separated modes.

The results demonstrate consistent patterns across problems. On the $2d$ MoG with 40 modes—a challenging benchmark from the ML sampling literature (Midgley et al., 2023)—all our particle methods recover the correct mode structure (fig. 4), confirming that our SMC implementations are competitive baselines and likely underrepresented in this literature. The interesting performance gaps between methods emerge on harder problems involving high-dimensional multimodality, invalid likelihood regions, and hierarchical structure.

On the $10d$ funnel, NSS has the lowest MMD score, perhaps unsurprising as slice sampling was introduced in the literature to solve this exact problem. For multimodal targets, NSS maintains significantly better posterior quality metrics indicating proper mode discovery and weighting as shown in table 1. All methods

Table 1: Performance on synthetic benchmarks. NSS vs SMC-RW vs SMC-HMC, averaged over 5 runs. $W_2$ and MMD computed against ground truth samples. Full results with additional baselines in section B.

| Problem | Method | Energy evals | Time (s) | ESS/s | ln $Z$ | MMD | $W_2$ |
|---|---|---|---|---|---|---|---|
| MoG | Truth | - | - | - | -9.21 | $0.021 \pm 0.009$ | $3.51 \pm 0.71$ |
| [$d$=2, 40 modes] | NSS | $7.8 \times 10^4$ | 5.1 | $7.4 \times 10^2$ | $-9.19 \pm 0.02$ | $0.029 \pm 0.014$ | $4.06 \pm 0.87$ |
| | SMC-RW | $1.9 \times 10^5$ | 3.6 | $2 \times 10^3$ | $-9.21 \pm 0.01$ | $0.032 \pm 0.012$ | $4.22 \pm 0.75$ |
| | SMC-HMC | $1.9 \times 10^5$ | 8.1 | $8.6 \times 10^2$ | $-9.22 \pm 0.06$ | $0.034 \pm 0.019$ | $4.48 \pm 1.2$ |
| MoG | Truth | - | - | - | -46.05 | $0.036 \pm 0.02$ | $4.70 \pm 0.95$ |
| [$d$=10, 5 modes] | NSS | $1.5 \times 10^6$ | 8.1 | $9.9 \times 10^2$ | $-45.97 \pm 0.11$ | $0.19 \pm 0.05$ | $9.56 \pm 0.89$ |
| | SMC-RW | $2.7 \times 10^6$ | 4.1 | $7.3 \times 10^2$ | $-47.93 \pm 1.33$ | $1.9 \pm 0.65$ | $19.0 \pm 2.7$ |
| | SMC-HMC | $5.4 \times 10^5$ | 8.5 | $3.5 \times 10^2$ | $-46.13 \pm 0.46$ | $0.5 \pm 0.23$ | $12.3 \pm 1.8$ |
| Funnel | Truth | - | - | - | - | $(8.3 \pm 1.5) \times 10^{-3}$ | $7.35 \pm 0.40$ |
| [$d$=10] | NSS | $2 \times 10^6$ | 7.7 | $4 \times 10^3$ | $-62.34 \pm 0.10$ | $0.033 \pm 0.002$ | $9.17 \pm 0.06$ |
| | SMC-RW | $2.5 \times 10^6$ | 4.1 | $5.6 \times 10^3$ | $-63.56 \pm 0.17$ | $0.066 \pm 0.005$ | $8.92 \pm 0.01$ |
| | SMC-HMC | $5.2 \times 10^5$ | 8.2 | $2.9 \times 10^3$ | $-62.71 \pm 1.42$ | $0.059 \pm 0.010$ | $9.03 \pm 0.22$ |

have similar evaluation overheads, and produce large numbers of posterior samples per second. The $10d$ MoG starts to differentiate tested methods, with slice sampling embedded in particle methods, particularly NSS, proving generally the most reliable at mode recovery. We note that the broadly similar metrics across methods on these benchmarks are expected: the purpose here is to validate that NSS keeps pace with performant baselines from the same class of particle methods, rather than to claim superiority on problems where all methods perform well. Section B.6 provides additional analysis of evidence estimation accuracy and reliability as dimension increases.

## 4.2 Inference Gym

To complement the results on synthetic benchmarks, we evaluate on real-data problems from the Inference Gym (Sountsov et al., 2020) repository. These Bayesian probabilistic models include ground truth posteriors derived from exhaustive NUTS runs. We compare NSS against adaptive tempered SMC with multiple mutation kernels (SMC-RW/IRMH/SS/HMC). Results are shown in table 2, with posterior quality measured by $W_2$ and MMD against the NUTS reference samples. No ground truth marginal likelihood estimates are available for these problems, so we compare ln $Z$ estimates across methods for consistency.

The results reveal several patterns. On lower-dimensional problems (EightSchools, GermanCredit), all methods achieve similar posterior quality and compatible evidence estimates. NSS matches SMC-RW and SMC-HMC on EightSchools in both MMD and $W_2$, and achieves the best posterior quality on GermanCredit logistic regression. For the higher-dimensional problems (S&P500, RadonIndiana), NSS remains competitive. On S&P500, NSS achieves comparable evidence variance to SMC-HMC and substantially outperforms SMC-RW on all metrics; SMC-HMC achieves the best posterior quality, benefiting from gradient information in this smooth stochastic volatility model. On RadonIndiana, NSS achieves the best performance across all metrics, with substantially lower MMD and $W_2$ than SMC-HMC and tighter evidence estimates; SMC-RW fails on this hierarchical model, substantially underestimating the evidence. These results demonstrate that NSS scales effectively to moderately high-dimensional problems; all NSS posteriors are broadly consistent with the NUTS reference draws on these examples. The purpose of these benchmarks is to validate that NSS remains competitive with state-of-the-art GPU-accelerated particle methods, rather than to claim superiority; scaling further will likely require gradient-based constrained samplers.

## 4.3 Gaussian Process hyperparameter marginalization

A practical application of Nested Sampling in machine learning is Bayesian Gaussian Process (GP) regression with hyperparameter marginalization (Rasmussen & Williams, 2006; Simpson et al., 2021). Standard practice often fits GP hyperparameters by maximizing the marginal likelihood (type-II maximum likelihood), but this point estimate ignores hyperparameter uncertainty and can lead to miscalibrated predictive uncertainties. A full Bayesian treatment instead marginalizes hyperparameters (Lalchand et al., 2022). Particle methods such as NSS additionally provide an estimate of the model evidence (marginal likelihood) $Z$, which enables

Table 2: Results on Inference Gym benchmarks. NSS vs SMC-RW vs SMC-HMC, averaged over 5 runs. $W_2$ and MMD computed against NUTS reference chains.

| Problem | Method | Energy evals | Time (s) | ESS/s | $\ln Z$ | MMD | $W_2$ |
|---|---|---|---|---|---|---|---|
| EightSchools [$d$=10] | NSS | $1.6 \times 10^6$ | 10 | $3.8 \times 10^2$ | $-36.15 \pm 0.09$ | $(2.0 \pm 0.8) \times 10^{-3}$ | $0.86 \pm 0.12$ |
| | SMC-RW | $6.6 \times 10^6$ | 3.9 | $4.2 \times 10^3$ | $-36.11 \pm 0.16$ | $(1.8 \pm 0.8) \times 10^{-3}$ | $0.85 \pm 0.07$ |
| | SMC-HMC | $6.8 \times 10^6$ | 7.6 | $2.3 \times 10^3$ | $-36.16 \pm 0.10$ | $(2.0 \pm 0.3) \times 10^{-3}$ | $0.95 \pm 0.09$ |
| GermanCredit [$d$=25] | NSS | $2.0 \times 10^7$ | 16 | $5.5 \times 10^2$ | $-529.12 \pm 0.26$ | $(1.4 \pm 0.3) \times 10^{-3}$ | $(8.9 \pm 0.5) \times 10^{-3}$ |
| | SMC-RW | $2.4 \times 10^7$ | 6.7 | $2.5 \times 10^2$ | $-529.53 \pm 0.18$ | $(2.3 \pm 0.6) \times 10^{-3}$ | $(11 \pm 1.0) \times 10^{-3}$ |
| | SMC-HMC | $9.8 \times 10^6$ | 11 | $1.6 \times 10^2$ | $-529.19 \pm 0.08$ | $(4.1 \pm 0.6) \times 10^{-3}$ | $(13 \pm 0.7) \times 10^{-3}$ |
| S&P500 [$d$=103] | NSS | $3.7 \times 10^7$ | 79 | $1.1 \times 10^2$ | $-571.25 \pm 0.18$ | $0.014 \pm 0.002$ | $0.14 \pm 0.008$ |
| | SMC-RW | $5.7 \times 10^7$ | 53 | 47 | $-572.87 \pm 0.62$ | $0.023 \pm 0.006$ | $0.18 \pm 0.015$ |
| | SMC-HMC | $5.4 \times 10^6$ | 90 | 24 | $-571.02 \pm 0.19$ | $(2.2 \pm 0.6) \times 10^{-3}$ | $0.072 \pm 0.005$ |
| RadonIndiana [$d$=97] | NSS | $1.2 \times 10^8$ | 63 | $2.6 \times 10^2$ | $-2591.13 \pm 2.43$ | $(2.0 \pm 0.3) \times 10^{-3}$ | $0.029 \pm 0.002$ |
| | SMC-RW | $1.9 \times 10^8$ | 29 | $5.4 \times 10^2$ | $-3033.96 \pm 376.35$ | $0.5 \pm 0.55$ | $0.67 \pm 0.47$ |
| | SMC-HMC | $1.5 \times 10^7$ | 14 | $1.1 \times 10^3$ | $-2596.62 \pm 8.59$ | $0.04 \pm 0.039$ | $0.1 \pm 0.051$ |

principled Bayesian model comparison between kernel choices (Kroupa et al., 2024). Beyond its scientific interest, GP regression also serves as a useful proxy for models that place non-trivial computational load on the likelihood evaluation, making it a sharper test of GPU acceleration.

We demonstrate NSS on two standard GP regression benchmarks (Mauna Loa $CO_2$ and Airline Passengers), using a composite kernel with constant, linear, and spectral mixture components (Wilson & Adams, 2013), resulting in an 11-dimensional hyperparameter space (see section B.5 for kernel details). We again compare against adaptive tempered SMC with random-walk and HMC mutation kernels, and against NUTS as a posterior-only baseline. Because evaluating the GP log marginal likelihood involves repeated Cholesky factorisations, this problem benefits from double precision for numerical stability which does not vectorize as efficiently (see section C) across particles. We therefore use a reduced particle population of $m = 500$. This also limits the benefit of batch deletion, but it still provides a significant acceleration. In preliminary single-precision tests, NSS remained stable, consistent with HRSS being gradient-free and based on log-density comparisons (see section A.6). We use the GPJax library (Pinder & Dodd, 2022) for GP likelihood evaluations.

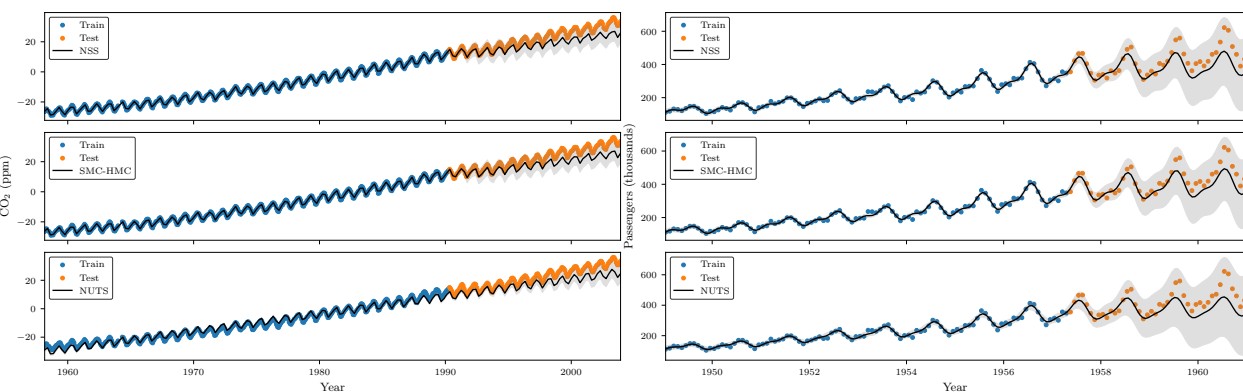

Figure 5: Posterior predictive distributions for GP regression on Mauna Loa $CO_2$ [left] and Airline passengers [right]. Shaded regions show 95% credible intervals from NSS samples.

The results in table 3 and posterior predictive distributions in fig. 5 reveal several important findings. First, Spectral Mixture kernels provide an extremely challenging posterior landscape to sample from. This surface is highly non-linear, multimodal and features sweeping degeneracies, which are all features NS is proposed to handle particularly well. To evaluate predictive performance we use a held-out test set (the last 30% of the data) and compute the test negative log-likelihood (NLL) and root-mean-square error (RMSE) under the posterior predictive distribution (see section D.1 for metric definitions). NSS provides the best predictive

Table 3: GP regression results on Mauna Loa $CO_2$ and Airline Passengers datasets ($d$=11 spectral mixture kernel). Results averaged over 5 runs. NSS provides marginal likelihood estimates with uncertainty, enabling model comparison.

| Problem | Method | $\ln Z$ | Time (s) | ESS | ESS/s | Test NLL | RMSE |
|---|---|---|---|---|---|---|---|
| Mauna Loa [$d$=11] | NSS | 874.42 ± 2.42 | 423 | 5511 | 13 | 38.01 ± 7.82 | 0.25 |
| | SMC-RW | 880.83 ± 1.36 | 130 | 495 | 3.8 | 57.87 ± 3.02 | 0.25 |
| | SMC-HMC | 875.34 ± 2.66 | 100 | 896 | 9.0 | 49.00 ± 12.31 | 0.25 |
| | NUTS | — | 277 | 1000 | 3.6 | 67.86 ± 52.42 | 0.30 |
| Airline Passengers [$d$=11] | NSS | 24.72 ± 0.57 | 32 | 4203 | 131 | 15.68 ± 0.77 | 0.46 |
| | SMC-RW | 24.02 ± 1.30 | 13 | 976 | 75 | 15.61 ± 0.80 | 0.45 |
| | SMC-HMC | 23.50 ± 0.92 | 19 | 563 | 30 | 15.24 ± 0.85 | 0.44 |
| | NUTS | — | 117 | 1000 | 8.5 | 25.23 ± 17.58 | 0.49 |

performance on Mauna Loa and competitive performance on Airline, where all particle methods achieve similar NLL. NSS's high effective sample size (ESS) indicates thorough posterior exploration; visually in fig. 5, all methods capture similar trends. As no ground-truth posterior is available for these GP problems, techniques such as simulation-based calibration (Modrák et al., 2025) could provide further validation of uncertainty calibration if required. We provide additional comparison with legacy NS implementations in section C.1, where NSS achieves over an order of magnitude speedup on identical hardware by fully exploiting batch likelihood evaluation.

## 5 Discussion

Both NSS and our SMC baselines rely on *mutation* steps that evolve particles via short Markov chains, and a key practical hyperparameter is the chain length $p$. If $p$ is too small, replacement particles remain strongly correlated with their duplicated parents, reducing the effective resolution of the particle population; if $p$ is too large, runtime is wasted on over-mixing relative to the outer-loop progress. This trade-off is present in many particle Monte Carlo methods. In terms of parallel structure, the SMC baselines apply an embarrassingly parallel update that attempts a fixed number of steps for all particles, while classical NS (with $k = 1$) is at the other extreme, updating one particle at a time. Our batched NS outer loop (with $k > 1$) lies between these extremes, and is reminiscent of waste-free particle schemes (Dau & Chopin, 2021).

Empirically, NSS is often efficient in terms of energy evaluations per effective sample, but can be slower in wall-clock time than SMC on hardware where strict SIMD synchronization penalizes variable-cost inner updates. In NSS, HRSS steps have adaptive control flow (stepping-out and shrinkage) and therefore a variable number of energy evaluations; when many chains are executed in lockstep this induces some wasted computation, as discussed in section C. Our tuning analysis helps reduce this effect: near the optimal slice width, HRSS step costs concentrate (section 3.2, fig. 12), making batched execution substantially more effective than naive constrained random walks. An important direction for future work is to adapt $p$ online using diagnostics of within-chain decorrelation across the population (Margossian et al., 2024). The NS literature has also explored *dynamic* schemes that vary the number of live particles (Higson et al., 2018), but such approaches are less compatible with a static-memory, accelerator-oriented implementation; adapting $p$ and inner-kernel parameters appears more tractable in this setting.

Across the chosen benchmarks, NSS provides robust performance with minimal manual tuning, consistent with the view that NS is particularly effective when the induced intermediate targets have hard constraints and complex geometry (section 2.2). In summary, NSS is a natural default when the target is multimodal or has constrained support, when the likelihood is simulator-like or non-differentiable, and when evidence estimation is required alongside posterior samples. When reliable gradients are available and the target is smooth and unimodal, gradient-based alternatives such as SMC-HMC or NUTS will typically be more efficient, as our own experiments confirm (section 4.2). We expect the proposed implementation to be useful in scientific and ML workflows where the forward model is already vectorized (or emulated)—a setting that is increasingly common in the physical sciences (Wong et al., 2023; Spurio Mancini et al., 2022)—and where estimating $\log Z$ is valuable for principled model comparison (Lovick et al., 2025; Leeney, 2025; Yallup et al.,

2025b). Alternative approaches to evidence estimation include bridge sampling and path sampling (Gelman & Meng, 1998); we do not address popular scalable approaches to model comparison based on stacking or cross-validation (Vehtari et al., 2017; Fong & Holmes, 2020; Yao et al., 2022), which primarily post-process draws from an existing inference procedure and are not themselves a sampling technique.

Our benchmarks are deliberately chosen to overlap with those used in the growing literature on neural samplers (e.g. Midgley et al., 2023; Wu et al., 2025; Blessing et al., 2025), which evaluate often on similar regression problems in 10–100 dimensions and benchmark against SMC. The synthetic experiments we highlight, multimodality, funnels, and other geometries where HMC has known difficulties, are precisely where there is general interest in improved samplers. An implicit goal of this work is to provide strong, well-tuned classical baselines for such comparisons: NSS and our SMC implementations are GPU-native, operate in the same JAX ecosystem as many neural samplers, and can serve as reference methods against which learned approaches should be evaluated.

In higher dimensions ($d > 10^3$), more sophisticated constrained mutation kernels (e.g. gradient-guided variants) may be required to maintain efficiency (Lemos et al., 2023); at the same time, exact evidence estimation for highly multimodal non-convex targets is challenging for any method, and understanding practical scaling limits remains an open problem. We also provide practical accelerator friendly versions of slice sampling, optionally embedded within SMC, and note that for highly non-linear problems slice sampling is highly performant. This embodies a broader contribution of this work, beyond just NSS, of well tested and efficient constrained sampling methods for modern ML ecosystems.

## 6  Conclusion

Nested Sampling is a widely used tool for Bayesian model comparison in the physical sciences, notably in astrophysics (Trotta, 2008), but has seen comparatively less adoption in modern ML software ecosystems. We presented an accelerator-oriented implementation of Nested Sampling in the `jax` ecosystem, structured as a batched NS outer kernel with a pluggable constrained-update interface. Instantiating this interface with Hit-and-Run Slice Sampling yields Nested Slice Sampling (NSS), a simple and robust variant designed for hard constraints and massively parallel likelihood evaluation.

Besides this engineering contribution, our main methodological contribution is a principled treatment of the HRSS slice-width hyperparameter: we provide an optimal-width analysis (including the $w_* \approx 1.36\,\ell$ rule and its high-dimensional scaling for ellipsoidal sets) and show that operating near this regime leads to well-behaved per-step costs that support vectorized execution. We empirically compare NSS to strong adaptive tempered SMC baselines and demonstrate competitive performance on challenging synthetic targets, High dimensional Bayesian inference, and GP hyperparameter marginalization (section 4.3), where evidence estimation enables Bayesian model comparison and improved uncertainty calibration. Our implementation of NSS is particularly performant for moderate-dimensional problems that exhibit complex constrained geometries, and where the likelihood is already vectorized or emulated.

The accompanying open-source implementation aims to make NS easier to use and to benchmark in ML settings, while remaining compatible with established NS practice (e.g. evidence termination) and removing several legacy complexities (e.g. clustering heuristics; see section C.1). Open problems include better automatic selection of the mutation length $p$, more asynchronous batched execution strategies on accelerators (Bou-Rabee et al., 2025), and integrating gradient information into constrained updates when it is available.

### Acknowledgments

We thank Sam Power, Adam Ormondroyd, Sam Leeney, Toby Lovick, and Metha Prathaban for helpful discussions and code testing. An early version of this work was presented at the ICLR Frontiers of Probabilistic Inference workshop (Yallup et al., 2025a). We used OpenAI GPT-5.2 to refine portions of the draft, and Claude Opus 4.5 was used in refining the code. The authors take full responsibility for the final content. This material is based upon work supported by the Google Cloud research credits program, with the award GCP442577929. The authors were supported by the research environment and infrastructure of the Handley

Lab at the University of Cambridge. N. K. was supported by the Harding Distinguished Postgraduate Scholarship.

**Code Availability**

The Nested Sampling implementation is a pending contribution to the `blackjax` library. Experiment code scripts, and a current working nested sampling implementation, are available at `https://github.com/yallup/nss/`.

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

# A    Baseline Methods and Configuration

## A.1    Adaptive Tempered SMC

SMC methods estimate ratios of normalizing constants by moving a particle system through a sequence of bridging distributions $\{\pi_t\}_{t=0}^{T}$ with unnormalized densities $\{\gamma_t\}$, using incremental importance weights, resampling, and MCMC mutation (Del Moral et al., 2006; Chopin, 2002). While many paths are possible (e.g. tempering, data-tempering, or other interpolations), for benchmarking we use a standard *tempered* (annealed) path targeting the power-posterior family

$$\pi_\beta(x) \ \propto \ \gamma_\beta(x) := \Pi(x)\exp(-\beta E(x)), \qquad \beta \in [0,1], \tag{11}$$

with $\beta_0 = 0$ and $\beta_T = 1$.

Given particles $\{x_i^{(t)}\}_{i=1}^{m}$ approximating $\pi_{\beta_t}$, we choose the next temperature $\beta_{t+1} > \beta_t$ adaptively using an ESS criterion following Fearnhead & Taylor (2010). For a candidate $\beta$, define incremental weights

$$\tilde{w}_i(\beta) := \exp\big( - (\beta - \beta_t)E(x_i^{(t)}) \big), \qquad \bar{w}_i(\beta) := \frac{\tilde{w}_i(\beta)}{\sum_{j=1}^{m} \tilde{w}_j(\beta)}, \tag{12}$$

and the effective sample size $\mathrm{ESS}(\beta) := 1/\sum_{i=1}^{m} \bar{w}_i(\beta)^2$. We then set $\beta_{t+1}$ so that $\mathrm{ESS}(\beta_{t+1}) \approx \rho m$ for a fixed target $\rho \in (0,1)$. The incremental normalizing-constant ratio has the standard estimator

$$\frac{\widehat{Z}(\beta_{t+1})}{\widehat{Z}(\beta_t)} \ = \ \frac{1}{m}\sum_{i=1}^{m} \tilde{w}_i(\beta_{t+1}), \tag{13}$$

so $\log\widehat{Z}$ accumulates as a sum of log-means over SMC steps.

After reweighting, we resample particles according to $\bar{w}_i(\beta_{t+1})$ and then apply an MCMC *mutation* kernel that leaves $\pi_{\beta_{t+1}}$ invariant. The inner kernel is a design choice; common options include random-walk or independent Metropolis–Hastings proposals (Robert & Casella, 2005; Andrieu et al., 2010) and Hamiltonian Monte Carlo (Duane et al., 1987; Buchholz et al., 2020). In our benchmarks we choose inner kernels and tuning rules to match NSS as closely as possible (e.g. comparable particle counts and comparable numbers of inner-kernel steps), so that differences primarily reflect the bridging path (tempered versus constraint-based) and the resulting geometry of the intermediate targets. Prior comparisons between NS and SMC have used a range of NS inner kernels (Grosse et al., 2015; Salomone et al., 2024), and inner-kernel choice can strongly affect the practical performance of either method.

The adaptive tempering sequence is closest in spirit to the automatic bridging sequence of NS, and the ESS criterion plays a similar role to the compression factor $k/m$ in NSS.

## A.2    Adaptive non-equilibrium samplers and parallelism

The nested and SMC samplers studied in this paper can be viewed as *adaptive non-equilibrium samplers*: they use a termination criterion, their outer loops run for a data-dependent (non-deterministic) number of steps, and they target an evolving sequence of distributions. This structure is less immediately friendly to accelerators; however, it offers distinct advantages in robustness and ease of use. It is possible to express the computation using fixed-length scans, or to use `jax.lax.while_loop` to lower the full loop into a single JIT-compiled function. In practice, however, one often wants access to intermediate states, which makes this less desirable.

We quantify the practical overhead associated with adaptive execution, including host–accelerator communication and schedule discovery, on two 20-dimensional test problems. We compare the full adaptive loop, in which the schedule is discovered on the fly, with a fixed `scan` over a precomputed schedule (assuming it is known *a priori*, which is not the case in practice). We implement both variants for NSS and SMC-IRMH. To assess the additional overhead introduced by particle reweighting, we also include an Annealed Importance Sampling (AIS) baseline (Neal, 1998), which can be viewed as the no-resampling analogue of tempered SMC,

here using the same independent-MH transition kernel for comparability. Optimised AIS (OAIS) (Syed et al., 2025) extends this framework with optimised tempering schedules and could serve as a more advanced parallel baseline in future work.

The targets are: (i) a 20-dimensional Gaussian model with prior $\Pi(x) = \mathcal{N}(x; 0, \mathbf{I})$ and Gaussian likelihood proportional to $\mathcal{N}(x; \mathbf{1}, 0.01\,\mathbf{I})$; and (ii) a 6-component mixture of Gaussians with a broad uniform prior, in which one mode carries substantially more mass than the others. Results are shown in table 4. Samplers use the default settings described in section B, run on the same hardware, and are averaged over 5 seeds. In these examples the likelihood is deliberately cheap, so host–accelerator communication overhead is comparatively most visible; even so, the extra overhead from the adaptive implementation is modest. On the Gaussian target, all methods recover similar evidence values, whereas on the mixture target SMC-IRMH and AIS substantially underestimate the evidence, highlighting that the comparison reflects both overhead and robustness. The NSS loop is computationally more expensive, but it delivers more robust estimates on the more challenging mixture problem.

Table 4: Runtime overhead and evidence estimates on two 20-dimensional problems ($m = 1000$, averaged over 5 runs). ($\beta$) denotes replay over a fixed schedule with `jax.lax.scan`.

| | Gaussian | | MoG | |
|---|---|---|---|---|
| Method | Runtime (s) | $\ln Z$ | Runtime (s) | $\ln Z$ |
| NSS | 21.52 | -28.19 $\pm$ 0.14 | 25.47 | -72.49 $\pm$ 1.03 |
| NSS ($\beta$) | 15.69 | -27.92 $\pm$ 0.13 | 18.65 | -72.38 $\pm$ 0.98 |
| SMC-IRMH | 5.15 | -28.35 $\pm$ 0.04 | 5.51 | -91.87 $\pm$ 0.06 |
| SMC-IRMH ($\beta$) | 4.49 | -28.39 $\pm$ 0.06 | 4.77 | -91.93 $\pm$ 0.12 |
| AIS | 3.76 | -28.39 $\pm$ 0.10 | 4.21 | -91.83 $\pm$ 0.38 |
| Analytic | — | -28.38 | — | -73.68 |

## A.3 Algorithm Pseudocode

For reference, algorithm 1 and algorithm 2 give pseudocode for the outer kernels of Nested Sampling and tempered SMC respectively, corresponding to the descriptions in sections 2.1 and 2.3.

---

**Algorithm 1** Nested Sampling outer kernel

---

**Require:** Live particles $\{x_i\}_{i=1}^m$, deletion batch size $k$
1: Evaluate energies $\{E(x_i)\}_{i=1}^m$
2: Sort particles by $E(x_i)$ and identify the $k$ highest-energy (lowest-likelihood) particles
3: Record batch threshold $E_{\text{batch}} \leftarrow$ minimum energy among the top-$k$
4: Store the $k$ deleted particles and their energies as dead points at level $E_{\text{batch}}$
5: *Resample* $k$ parents from the remaining $m-k$ live particles
6: *Mutate* each parent with a constrained kernel targeting $\Pi_{E_{\text{batch}}}(x) \propto \Pi(x)\,\mathbf{1}_{\{E(x) < E_{\text{batch}}\}}$ for $p$ steps
7: Replace the $k$ dead particles with the $k$ new samples
**Ensure:** Updated live particles $\{x_i\}_{i=1}^m$, updated dead-point record

---

## A.4 NSS Default Configuration

Unless otherwise noted, NSS experiments use population size $m = 1000$ and batch deletion size $k = 100$ (i.e., $k/m = 0.1$). We use $p = d$ MCMC steps per replacement, where $d$ is the parameter dimension. The theoretical analysis in theorem 3 shows that the optimal slice width scales as $w^\star \propto d^{-1/2}$ for a $d$-dimensional isotropic target. In practice, we combine a fixed base width $w = 1.0$ with adaptive whitening: at each outer iteration, we estimate the live-set covariance and sample HRSS directions in the corresponding Mahalanobis

---

**Algorithm 2** Adaptive tempered SMC outer kernel

---

**Require:** Particles $\{x_i\}_{i=1}^m$ approximating $\pi_{\beta_t}$, temperature $\beta_t$
1: Select next temperature $\beta_{t+1}$ (e.g. by targeting a desired ESS from incremental weights)
2: Compute incremental weights $\tilde{w}_i \leftarrow \exp\big(-(\beta_{t+1}-\beta_t)\,E(x_i)\big)$
3: Accumulate $\log \widehat{Z} \leftarrow \log \widehat{Z} + \log\big(\frac{1}{m}\sum_{i=1}^m \tilde{w}_i\big)$
4: Normalize $\bar{w}_i \leftarrow \tilde{w}_i / \sum_{j=1}^m \tilde{w}_j$
5: *Resample* $\{x_i\}_{i=1}^m$ with replacement according to $\{\bar{w}_i\}$
6: *Mutate* each particle with an MCMC kernel targeting $\pi_{\beta_{t+1}}$ for $p$ steps
**Ensure:** Updated particles $\{x_i\}_{i=1}^m$ approximating $\pi_{\beta_{t+1}}$, temperature $\beta_{t+1}$

---

metric (see section 3.2). This effectively rescales the slice width to the local geometry, so that a simple fixed $w$ in the whitened space approximates the optimal scaling. The stepping-out procedure then adapts to the actual slice length, ensuring robustness when the whitening is imperfect.

Termination occurs when the estimated remaining evidence contribution falls below $e^{-3}$ (synthetic) or $e^{-5}$ (Inference Gym) of the current estimate. For GP regression experiments requiring double precision, we use $m = 500$ with $k = 250$. For the higher-dimensional Inference Gym problems (S&P500, RadonIndiana), we use $m = 1000$ with $k = 500$ and triple the MCMC steps to $p = 3d$; the same tripling is applied to SMC baselines for consistency (although this is likely conservative for both methods).

## A.5   SMC Inner Kernel Configuration

This section details the specific inner kernels used for our SMC baselines. We deliberately use settings that are stronger than typical library defaults to ensure fair comparisons: high ESS targets, long mutation chains, and adaptive proposals. Across all SMC variants we use $m = 1000$ particles (or $m = 500$ for GP experiments) and adaptive tempering with an ESS target of $0.9m$ (Fearnhead & Taylor, 2010). Within each SMC step we apply $p = 5d$ inner-kernel transitions per particle for random-walk kernels and $p = 2 - 5$ HMC transitions with trajectory length $L = 5$–$10$ for HMC kernels. As noted in section A.4, we triple inner step counts for the higher-dimensional Inference Gym problems.

**SMC-RW (random-walk Metropolis–Hastings).**   We use a Gaussian random-walk proposal with covariance estimated from the current particle population and scaled as $2.38^2/d$, following standard optimal-scaling results for random-walk MH (Geyer, 1992). The resulting Metropolis–Hastings kernel targets the current tempered density $\pi_{\beta_t}$ (Robert & Casella, 2005).

**SMC-IRMH (independent Metropolis–Hastings).**   We use an independent Gaussian proposal fitted from the current particle cloud (empirical mean and covariance) and apply an independent MH correction (Robert & Casella, 2005; South et al., 2019). This can be interpreted as an importance-sampling-like mutation step inside a particle Monte Carlo method (Andrieu et al., 2010), and is closely related in spirit to Annealed Importance Sampling when used along a tempered path (Neal, 1998).

**SMC-HMC (Hamiltonian Monte Carlo).**   We use an HMC mutation kernel with mass-matrix adaptation based on the particle covariance, and step size scaled using the Expected Squared Jump Distance of the previous ($\beta_{t-1}$) temperature step trajectory (Buchholz et al., 2020). This provides a gradient-based baseline on problems where gradients are available and the geometry is amenable to Hamiltonian dynamics.

**SMC-SS (slice sampling).**   We also consider slice sampling as an SMC mutation kernel (Neal, 2003). This targets the tempered density $\pi_{\beta_t}(x) \propto \Pi(x)\exp(-\beta_t E(x))$ without hard constraints, and provides a gradient-free alternative to MH/HMC within the same SMC outer loop. This is an important control as the inner kernel exactly matches that of NSS, with identical tuning (slice width, number of steps).

**Additional baselines.**   We did not include more expressive learned proposals (e.g. flow-based repeated annealing) (Matthews et al., 2023) or alternative classical baselines such as parallel tempering (Syed et al.,

2021), in order to keep the comparison focused on strong, standard kernels. We report point estimates of $\log Z$ from SMC; unbiasedness and error bounds for SMC normalizing-constant estimators are available under standard conditions (Beskos et al., 2011). Finally, we use standalone SS and NUTS as posterior-only baselines where appropriate; these do not provide marginal-likelihood estimates. NUTS is run with warmup window adaption (Carpenter et al., 2017), and SS is run for a similar number of steps (1000-2000) with covariance set from the initial particle sample covariance. We remark that the slice sampler in NSS can be combined with the recent reformulation of slice sampling as a finite state machine as a means to soften the synchronization penalty (Dance et al., 2025) to potentially yield an even larger speedup of slice sampling. For problems with explicit linear (polytope) constraints, the chord endpoints along a Hit-and-Run direction can instead be computed analytically, enabling more specialised higher-order or preconditioned Hit-and-Run variants (Paul et al., 2026).

### A.6 Hit-and-Run Slice Sampling Implementation

Our constrained update kernel is Hit-and-Run Slice Sampling (HRSS), implemented using the stepping-out and shrinkage procedures of slice sampling (Neal, 2003). For a current state $x$ and (randomised) direction $v$, HRSS performs a one-dimensional slice update along the line $\{x + tv : t \in \mathbb{R}\}$. We work in log space throughout and treat points outside the constraint (or outside the support of $\Pi$) as having log density $-\infty$.

**Stepping-out.** Given a slice height (auxiliary) variable, stepping-out expands an interval until it brackets the horizontal slice along the chosen line. We use the standard *linear stepping-out* procedure with a randomised initial bracket of width $w$ (Neal, 2003), which yields a transition kernel that leaves the target invariant under the usual slice-sampling conditions.

To ensure bounded-loop execution under JIT compilation, we cap the number of stepping-out expansions at 10 steps in each direction. This cap exists purely for safety against unbounded `while` loops and is not expected to affect the sampler under normal operation.

**Shrinkage.** Given a bracketing interval, we sample a candidate $t'$ uniformly from the current interval and accept it if it lies in the slice; otherwise we shrink the interval by replacing the left or right endpoint with $t'$ and repeat. This shrinkage loop produces an exact slice update along the line in the idealised setting, and in particular it does not require a separate Metropolis–Hastings accept/reject step.

As with stepping-out, we cap the maximum number of shrinkage iterations at 100 for robustness. The stepping-out cap is implemented as in Neal (2003) and the resulting algorithm is unbiased regardless of whether this limit is reached. The shrinkage cap is more consequential: if reached, the update returns the current state as a null move. In practice, reaching the shrinkage cap signals a pathological condition. We have identified two scenarios where this can occur: (a) the likelihood surface reaches a flat plateau (more likely in reduced precision), in which case the cap serves as an augmented termination criterion alongside compression-based stopping, and the result remains unbiased; (b) the likelihood is non-deterministic (e.g. consuming a random seed), which violates the standard assumption of a deterministic target and would require a pseudo-marginal extension (Murray & Graham, 2016). Across all experiments in section 4 neither the stepping-out cap (10 per direction) nor the shrinkage cap (100) was ever reached.

**Reduced-precision execution.** HRSS requires only comparisons of (log-)densities and does not use gradients or numerical integration. This makes it straightforward to run in single or mixed precision, provided the log-density computation itself is stable. We did not observe the same sensitivity to reduced precision that can arise in gradient-based inner kernels. We found relatively robust performance on all problems, including GP hyperparameter marginalization, when using single precision. Due to potential instabilities of other algorithms on the GP tasks, we used double precision for all methods in that section to ensure a fair comparison.

## A.7 Slice Sampling vs Random Walk

A key advantage of slice sampling over random-walk Metropolis–Hastings for constrained sampling is the *concentration of per-step computational cost.* This property is critical for efficient GPU execution: when many chains run in parallel via `jax.vmap`, the wall-clock time per batch is determined by the slowest chain. High variance in per-step cost leads to wasted computation as faster chains wait for stragglers.

We compare NSS (using HRSS) against constrained random walk (RW) on an ill-conditioned Gaussian target with condition number $\kappa = 100$. For RW, we use a Gaussian proposal with covariance matched to the target and scaled optimally, rejecting proposals that violate the constraint. Table 5 shows that NSS maintains nearly constant cost ($\approx 5$ evaluations per step with standard deviation $\approx 1.2$) across dimensions 10–100, while RW has both higher mean cost and dramatically higher variance (standard deviation $\approx 30$–$34$).

Table 5: Evaluations per step for constrained sampling on an ill-conditioned Gaussian ($\kappa = 100$). NSS uses slice sampling; RW uses random walk with rejection. NSS maintains constant cost with low variance across dimensions, while RW exhibits high variance that degrades batched execution efficiency.

| Dimension | Mean $\pm$ std Evals per successful sample | |
| --- | --- | --- |
| | NSS | RW |
| 10 | $4.9 \pm 1.2$ | $19.6 \pm 30.7$ |
| 50 | $5.0 \pm 1.2$ | $25.3 \pm 33.7$ |
| 100 | $5.1 \pm 1.2$ | $24.1 \pm 32.7$ |

The distribution of per-step costs reveals why this matters for vectorized execution. Figure 6 shows that NSS cost is tightly concentrated around the mean, while RW exhibits a heavy tail extending to 100+ evaluations per step. In a batched setting, even a small fraction of high-cost chains forces all other chains to wait, severely degrading parallel efficiency.

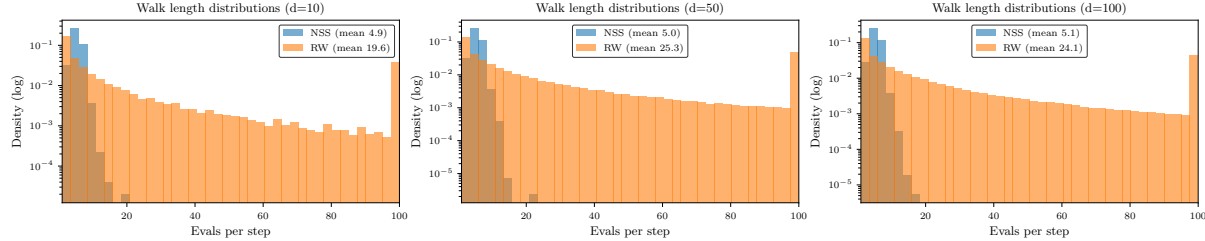

Figure 6: Distribution of likelihood evaluations per constrained step for NSS (slice sampling, blue) vs RW (random walk, orange) on an ill-conditioned Gaussian ($\kappa = 100$) in dimensions 10, 50, and 100. NSS maintains concentrated cost ($\approx 5$ evals, with standard deviation $\approx 1$) while RW exhibits heavy tails that degrade batched execution efficiency.

This empirical finding aligns with the theoretical analysis in section E.2.2, which shows that HRSS cost variance remains $O(1)$ across a wide range of dimensions when operating near the optimal slice width. The concentration property is a direct consequence of the stepping-out procedure automatically adapting to the local geometry, whereas rejection-based methods accumulate rejections at the constraint boundary.

# B Extended Experimental Results

This section provides detailed problem descriptions and complete results for the synthetic benchmarks discussed in section 4.1. For each problem, we provide the target distribution definition, ground truth computation methodology, and extended performance comparisons. Across all synthetic experiments we report error bars on key quality metrics ($W_2$, MMD) computed over 5 independent draws from the reference density. We run each algorithm with 5 independent seeds to measure the run-to-run variability on a subset

of the problems. As all experiments are run on identical hardware, and we are comparing across different algorithms with different computational profiles, we report results in terms of number of target density evaluations, wall clock time, and the ESS per wall clock time as a measure of efficiency. The ESS for NS is measured as described in section D.2, and for SMC variants it is computed by summing contributions from all annealing steps using the standard formula (Le Thu Nguyen et al., 2014). All reported wall-clock times include `jax` JIT compilation overhead, which represents a significant proportion of total runtime on these problems; production use would amortize this cost across multiple runs.

### B.1  Mixture of Gaussians ($d$=2, 40 modes)

This pedagogical example, introduced by Midgley et al. (2023), consists of 40 bivariate normal distributions arranged on a bounded uniform reference density. The ground truth evidence is $\ln Z = -9.21$, computed analytically from the mixture weights and component normalizations. Interestingly, standalone SS also mixes well despite being a single chain method. The full results are shown in table 6 and fig. 4.

Table 6: Performance on the 40-component mixture of Gaussians. Results averaged over 5 runs. $\ln Z$ reported where available.

| Method | Energy evals | Time (s) | ESS/s | $\ln Z$ | MMD | $W_2$ |
|---|---|---|---|---|---|---|
| Truth | - | - | - | -9.21 | $0.021 \pm 0.009$ | $3.51 \pm 0.71$ |
| SS | $6.6 \times 10^4$ | 9.2 | $1.1 \times 10^3$ | - | $0.027 \pm 0.015$ | $3.84 \pm 0.94$ |
| NUTS | $4.4 \times 10^4$ | 37 | $2.7 \times 10^2$ | - | $2.8 \pm 0.27$ | $34.7 \pm 2.1$ |
| *Particle Methods* | | | | | | |
| NSS | $7.8 \times 10^4$ | 5.1 | $7.4 \times 10^2$ | $-9.19 \pm 0.02$ | $0.029 \pm 0.014$ | $4.06 \pm 0.87$ |
| NS-RW | $6.4 \times 10^4$ | 3.3 | $1.1 \times 10^3$ | $-9.20 \pm 0.04$ | $0.032 \pm 0.012$ | $4.42 \pm 0.63$ |
| SMC-SS | $2 \times 10^5$ | 5.9 | $1.2 \times 10^3$ | $-9.20 \pm 0.01$ | $0.036 \pm 0.012$ | $4.43 \pm 0.54$ |
| SMC-RW | $1.9 \times 10^5$ | 3.6 | $2 \times 10^3$ | $-9.21 \pm 0.01$ | $0.032 \pm 0.012$ | $4.22 \pm 0.75$ |
| SMC-IRMH | $1.9 \times 10^5$ | 3.5 | $2 \times 10^3$ | $-9.21 \pm 0.01$ | $0.023 \pm 0.008$ | $3.82 \pm 0.79$ |
| SMC-HMC | $1.9 \times 10^5$ | 8.1 | $8.6 \times 10^2$ | $-9.22 \pm 0.06$ | $0.034 \pm 0.019$ | $4.48 \pm 1.2$ |

### B.2  Mixture of Gaussians ($d$=10)

This benchmark extends multimodal sampling challenges to higher dimensions. The target is a mixture of five 10-dimensional Gaussian distributions with randomized means and covariances, designed to test mode discovery and proper weighting across well-separated regions of parameter space.

Ground truth samples are drawn directly from the known mixture components, providing exact reference distributions for quality metrics. This problem starts to differential the particle methods, where both SMC-SS and NSS are particularly strong. This pattern holds on random re-seeding. The full results of this experiment are shown in table 7 and fig. 7.

Table 7: Performance on the $d$=10 mixture of Gaussians. Results averaged over 5 runs.

| Method | Energy evals | Time (s) | ESS/s | $\ln Z$ | MMD | $W_2$ |
|---|---|---|---|---|---|---|
| Truth | - | - | - | -46.05 | $0.036 \pm 0.02$ | $4.70 \pm 0.95$ |
| SS | $6.7 \times 10^4$ | 9.5 | $1.1 \times 10^3$ | - | $2.7 \pm 0.24$ | $22.2 \pm 0.53$ |
| NUTS | $3.6 \times 10^5$ | 89 | $1.1 \times 10^2$ | - | $2.8 \pm 0.06$ | $22.5 \pm 0.12$ |
| *Particle Methods* | | | | | | |
| NSS | $1.5 \times 10^6$ | 8.1 | $9.9 \times 10^2$ | $-45.97 \pm 0.11$ | $0.19 \pm 0.05$ | $9.56 \pm 0.89$ |
| NS-RW | $1.5 \times 10^6$ | 5.3 | $1.3 \times 10^3$ | $-46.09 \pm 0.44$ | $0.94 \pm 0.41$ | $15.4 \pm 2.0$ |
| SMC-SS | $2.6 \times 10^6$ | 6.8 | $4.4 \times 10^2$ | $-46.07 \pm 0.05$ | $0.044 \pm 0.03$ | $5.43 \pm 1.4$ |
| SMC-RW | $2.7 \times 10^6$ | 4.1 | $7.3 \times 10^2$ | $-47.93 \pm 1.33$ | $1.9 \pm 0.65$ | $19.0 \pm 2.7$ |
| SMC-IRMH | $2.7 \times 10^6$ | 4.4 | $6.7 \times 10^2$ | $-46.88 \pm 0.22$ | $0.93 \pm 0.3$ | $15.3 \pm 1.8$ |
| SMC-HMC | $5.4 \times 10^5$ | 8.5 | $3.5 \times 10^2$ | $-46.13 \pm 0.46$ | $0.5 \pm 0.23$ | $12.3 \pm 1.8$ |

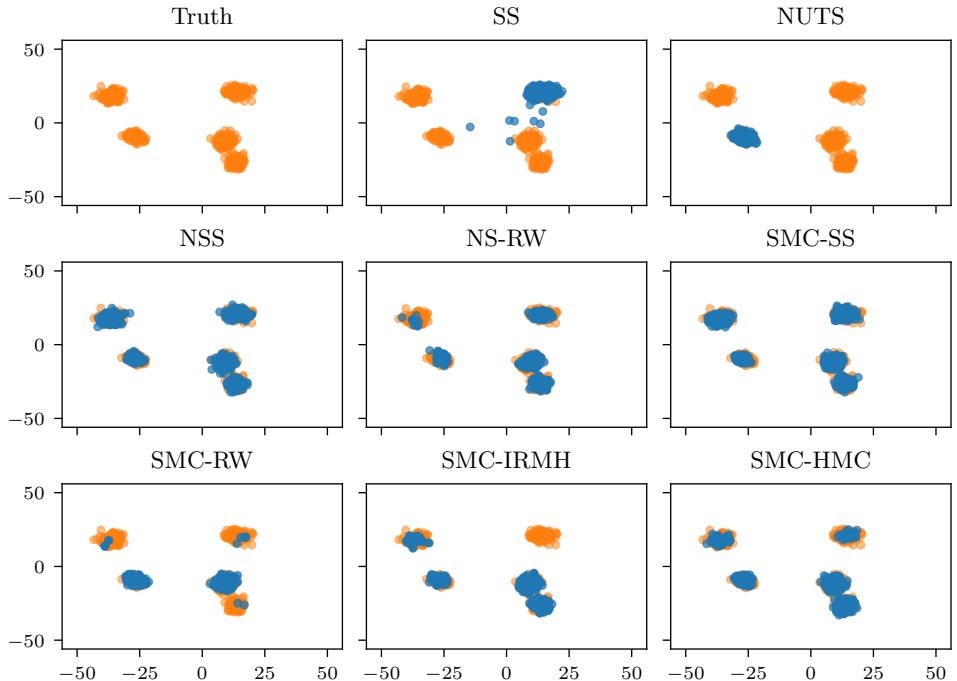

Figure 7: Marginal distribution of the first two parameters of the $d$=10 mixture of Gaussians. Ground truth samples (blue) compared with NSS samples (orange).

### B.3 Neal's Funnel

The funnel distribution (Neal, 2003) is a canonical example of a hierarchical structure that presents significant challenges for sampling algorithms. In 10 dimensions, the target density is defined as:

$$P(y, \mathbf{x}) = \mathcal{N}(y \mid 0, 3) \prod_{n=1}^{9} \mathcal{N}(x_n \mid 0, \exp(y/2)), \tag{14}$$

with a uniform prior distribution on all parameters. The hierarchical structure creates a "funnel" shape where the variance of the $x$ variables depends exponentially on $y$, causing the distribution to span many orders of magnitude in scale.

Ground truth samples are obtained by running HMC with a non-centered parameterization (Gorinova et al., 2019), which removes the problematic coupling. All algorithms under test use the more challenging centered parameterization to evaluate robustness to difficult geometries. Although HRSS has limited tuning requirements (primarily the slice width), embedding it within a particle method like NSS makes tuning effectively automatic: the adaptive whitening based on the live-set covariance (section 3.2) continuously rescales the sampler to the local geometry, removing the need for manual tuning even on problems with extreme scale variation like the funnel. The difference between NSS and SMC-SS on this problem highlights that NS provides a more natural framework to embed slice sampling within, compared to the tempered SMC approach. The full results are shown in table 8 and fig. 8.

### B.4 Inference Gym Models

We run a reduced, more targeted subset of the full algorithmic range on the remaining problems explored, restricting to SMC-RW, SMC-HMC and NSS. The NUTS reference chains used to assess quality were run with 1000 warm-up steps and for 5000 sampling steps, before thinning to 1000 posterior samples. The NUTS chains were run in double precision on the machine CPU as this executed significantly faster than using the

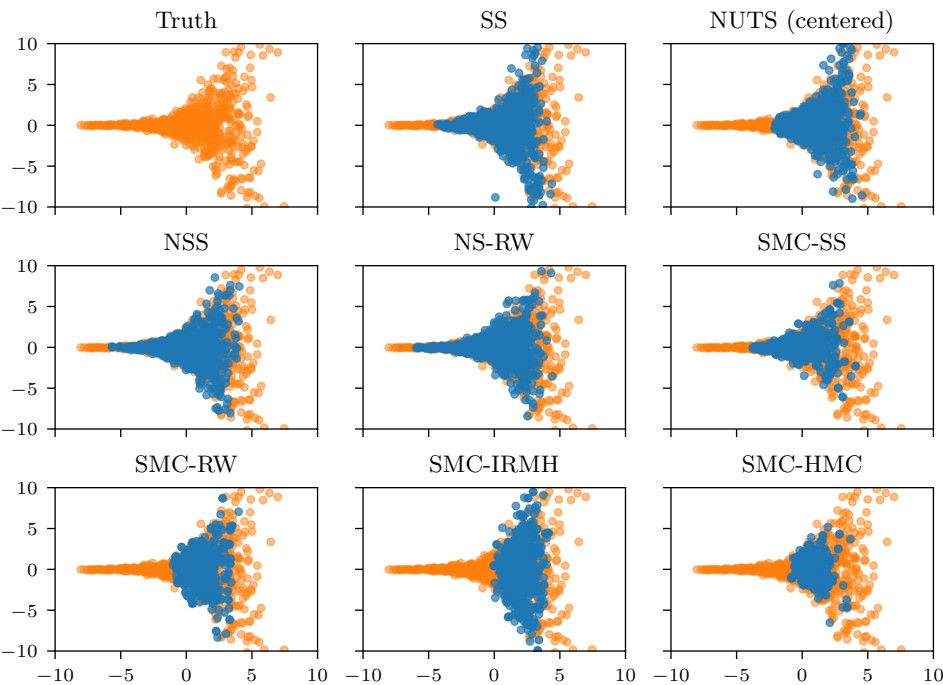

Figure 8: Marginal distribution of the first two parameters of Neal's $d=10$ funnel. Ground truth samples (blue) compared with NSS samples (orange).

Table 8: Performance on the $d=10$ funnel target. Results averaged over 5 runs. $\ln Z$ reported where available.

| Method | Energy evals | Time (s) | ESS/s | $\ln Z$ | MMD | $W_2$ |
|---|---|---|---|---|---|---|
| Truth | - | - | - | - | $(8.3 \pm 1.5) \times 10^{-3}$ | $7.35 \pm 0.40$ |
| NUTS (centered) | $4.1 \times 10^4$ | 20 | $1 \times 10^2$ | - | $0.031 \pm 0.004$ | $9.07 \pm 0.03$ |
| SS | $5.8 \times 10^4$ | 6.3 | $1.6 \times 10^3$ | - | $0.041 \pm 0.011$ | $9.22 \pm 0.18$ |
| *Particle Methods* | | | | | | |
| NSS | $2 \times 10^6$ | 7.7 | $4 \times 10^3$ | $-62.34 \pm 0.10$ | $0.033 \pm 0.002$ | $9.17 \pm 0.06$ |
| NS-RW | $2.2 \times 10^6$ | 5.6 | $5.9 \times 10^3$ | $-62.34 \pm 0.05$ | $0.033 \pm 0.003$ | $9.18 \pm 0.06$ |
| SMC-SS | $2.6 \times 10^6$ | 6.3 | $3.7 \times 10^3$ | $-62.63 \pm 0.18$ | $0.033 \pm 0.002$ | $9.08 \pm 0.05$ |
| SMC-RW | $2.5 \times 10^6$ | 4.1 | $5.6 \times 10^3$ | $-63.56 \pm 0.17$ | $0.066 \pm 0.005$ | $8.92 \pm 0.01$ |
| SMC-IRMH | $2.6 \times 10^6$ | 4.4 | $5.3 \times 10^3$ | $-63.53 \pm 0.10$ | $0.063 \pm 0.003$ | $8.95 \pm 0.03$ |
| SMC-HMC | $5.2 \times 10^5$ | 8.2 | $2.9 \times 10^3$ | $-62.71 \pm 1.42$ | $0.059 \pm 0.010$ | $9.03 \pm 0.22$ |

hardware accelerator, highlighting the fact that the throughput of a GPU is best exploited with ensemble methods. The standard prior and likelihood definitions provided by the library were used throughout Sountsov et al. (2020), with a filtering to prior samples applied to ensure valid support on all initial prior samples.

The S&P500 stochastic volatility model presents unique challenges due to its heavy-tailed Cauchy priors on volatility parameters, which produce high-variance samples during prior initialization. Additionally, this model requires filtering-based likelihood evaluation, which can amplify sensitivity to initialization. Without aggressive filtering of prior samples to ensure valid likelihood support, all particle methods fail entirely on this problem. With proper initialization, NSS achieves comparable evidence variance to SMC-HMC and substantially outperforms SMC-RW; SMC-HMC benefits from gradient information on this smooth target, achieving the best posterior quality metrics.

### B.5 Gaussian Process Hyperparameter Marginalization

This section provides details for the GP regression experiments in section 4.3. We use a composite kernel combining trend and periodic components, following the spectral mixture approach of Wilson & Adams (2013).

**Kernel specification.** The covariance function is a sum of three components:

$$k(x, x') = k_{\text{const}}(x, x') + k_{\text{linear}}(x, x') + k_{\text{SM}}(x, x'), \tag{15}$$

where the constant and linear kernels capture the long-term trend:

$$k_{\text{const}}(x, x') = \sigma_c^2, \tag{16}$$

$$k_{\text{linear}}(x, x') = \sigma_\ell^2 \, x^\top x', \tag{17}$$

and the spectral mixture kernel captures periodic structure:

$$k_{\text{SM}}(x, x') = \sum_{q=1}^{Q} w_q \exp\left(-2\pi^2 (x - x')^2 / \lambda_q^2\right) \cos(2\pi f_q (x - x')). \tag{18}$$

We use $Q = 3$ mixture components, giving 11 hyperparameters in total: $\{\sigma_c^2, \sigma_\ell^2, \sigma_n^2, w_{1:3}, \lambda_{1:3}, f_{1:3}\}$, where $\sigma_n^2$ is the observation noise variance.

**Priors.** All variance and scale parameters are assigned standard normal priors in log-space: $\log \sigma_c^2, \log \sigma_\ell^2, \log \sigma_n^2, \log w_q, \log \lambda_q \sim \mathcal{N}(0, 1)$. The frequencies $f_1 < f_2 < f_3$ are constrained to be sorted and lie in $[0, f_{\text{Nyquist}}]$, where $f_{\text{Nyquist}} = N/(2T)$ for $N$ observations over time span $T$. We place a uniform prior on the sorted frequencies, implemented via a Dirichlet-based bijection with appropriate Jacobian correction.

**Data preprocessing.** Both inputs $X$ and outputs $y$ are standardized (zero mean, unit variance) before inference. Reported predictions are transformed back to the original scale for visualization. We apply the methodology to two standard datasets: the Mauna Loa $CO_2$ dataset (Keeling et al., 1976) and the airline passenger dataset (Box & Jenkins, 1976).

**Experiment configuration.** We use $m = 500$ particles with $k = 250$ for NSS, and $m = 500$ particles for SMC baselines. NSS uses $p = d$ MCMC steps; SMC-RW uses $p = 5d$ steps; SMC-HMC uses $p = 2$ transitions with trajectory length $L = 10$. NUTS uses 1000 warmup and 1000 sampling iterations. Results are averaged over 5 independent runs with a 70/30 train/test split.

### B.6 Dimension Scaling of Evidence Estimation

Beyond per-step efficiency, we evaluate the accuracy of evidence estimation as dimension increases. This experiment also illustrates the distinction between two sources of uncertainty in NS evidence estimates:

1. Geometric uncertainty: Stochastic variation from the volume shrinkage process, computable via Monte Carlo simulation of Beta-distributed shrinkage factors (section D.2).

2. Mixing uncertainty: Systematic bias from imperfect constrained sampling that fails to maintain the i.i.d. assumption on the live set. This appears as run-to-run variation *beyond* the geometric uncertainty.

When the constrained sampler mixes well, empirical run-to-run variation should be consistent with the reported geometric uncertainty. When mixing fails, runs will exhibit systematic bias.

We use the $10d$ Mixture of Gaussians experimental design (section B.2) with an analytically known evidence, varying the dimension from 10 to 20. Figure 9 and Table 9 show results from averaging 10 independent runs per method.

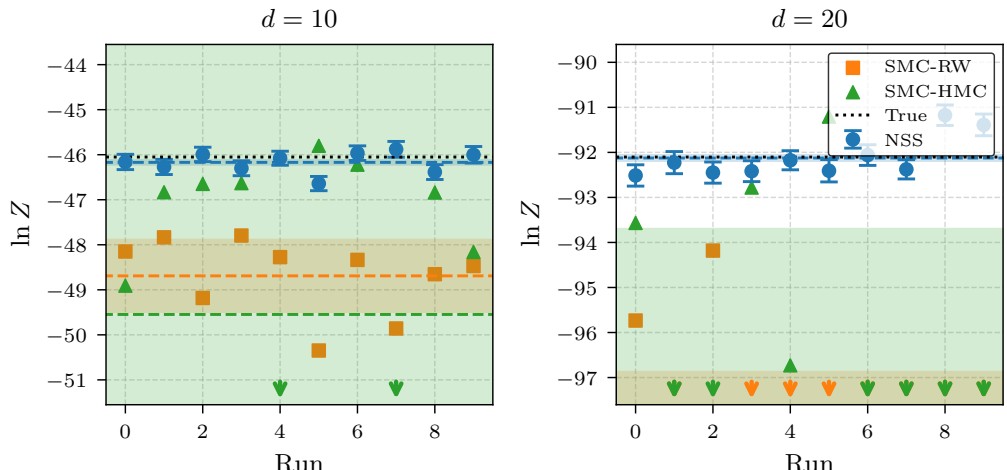

Figure 9: Evidence estimates ($\ln Z$) across 10 independent runs for MoG targets in $d{=}10$ (left) and $d{=}20$ (right). NSS (blue circles) consistently tracks the true value (black dotted line), while SMC-RW (orange squares) and SMC-HMC (green triangles) exhibit systematic negative bias that worsens with dimension. Arrows indicate runs that failed to reach $\beta = 1$ within the budget.

Table 9: Evidence estimation accuracy on MoG targets at $d = \{10, 20\}$. Mean absolute error $|\ln Z - \ln Z_{\text{true}}|$ from 10 independent runs. NSS* denotes the estimate obtained by unrolling the order statistics across all 10 runs into a single combined evidence estimate section D.2.

| | | | $|\ln Z - \ln Z_{\text{true}}|$ | |
|---|---|---|---|---|
| $d$ | NSS | NSS* | SMC-RW | SMC-HMC |
| 10 | $0.19 \pm 0.16$ | 0.12 | $2.64 \pm 0.81$ | $3.55 \pm 6.19$ |
| 20 | $0.35 \pm 0.27$ | 0.02 | $7.86 \pm 3.09$ | $8.04 \pm 6.05$ |

NSS maintains accurate evidence estimates across dimensions, with mean absolute error $\approx 0.2$–$0.35$ that is consistent with the NS geometric uncertainty—indicating that the constrained slice sampler mixes well and the reported uncertainties are reliable. When combining all 10 runs (as would be done in practice for uncertainty reduction), NSS achieves errors of 0.12 ($10d$) and 0.02 ($20d$). In contrast, both SMC-RW and SMC-HMC exhibit systematic negative bias that far exceeds their nominal uncertainties: errors of 2.6–3.6 in $10d$ worsen to 7.9–8.0 in $20d$. This demonstrates mixing uncertainty dominating: the samplers fail to discover all modes, producing biased evidence estimates regardless of the number of particles. The bias is particularly severe for SMC-HMC in $20d$, where most runs fail to discover all modes (indicated by arrows in fig. 9). The constraint-based path of NS, combined with the robustness of slice sampling, provides more reliable evidence estimation than tempered SMC with standard mutation kernels on this problem.

## B.7 Dimension Scaling of Inner mutation step

In section A.7 we showed that relative to a constrained random walk baseline, slice sampling maintains a constant per-step cost distribution as dimension increases. However, this doesn't address the mixing rate of the chain itself which will exhibit scaling with dimension. Hamiltonian Monte Carlo is known to have a favorable $d^{1/4}$ scaling for unconstrained targets (Beskos et al., 2010), while random walk methods typically scale as $d$ or worse (Geyer, 1992). The impact of this is that we keep the number of MCMC mutations per replacement proportional to dimension in algorithms that display this random walk scaling (NSS, SMC-RW) to maintain a roughly constant effective sample size per replacement. For SMC-HMC we make the realistic choice of using a fixed number (2-5) of HMC transitions per replacement, as the $d^{1/4}$ scaling is mild enough that this remains effective in practice.

**Mutation Chain Length** The number of MCMC steps $p$ per particle replacement controls the tradeoff between decorrelation and cost. Figure 10 shows ablations varying $p$ on the $d$=10 MoG problem (5 independent runs each). All methods use $m = 1000$ particles. NSS uses batch deletion $k = 100$ and sweeps $p \in \{1, 3, 5, \ldots, 20\}$. SMC-RW uses ESS target 0.9 and sweeps $p \in \{5, 15, \ldots, 250\}$. SMC-HMC uses ESS target 0.9, trajectory length $L = 5$, and sweeps $p \in \{1, 2, 4, 6, 8, 10\}$.

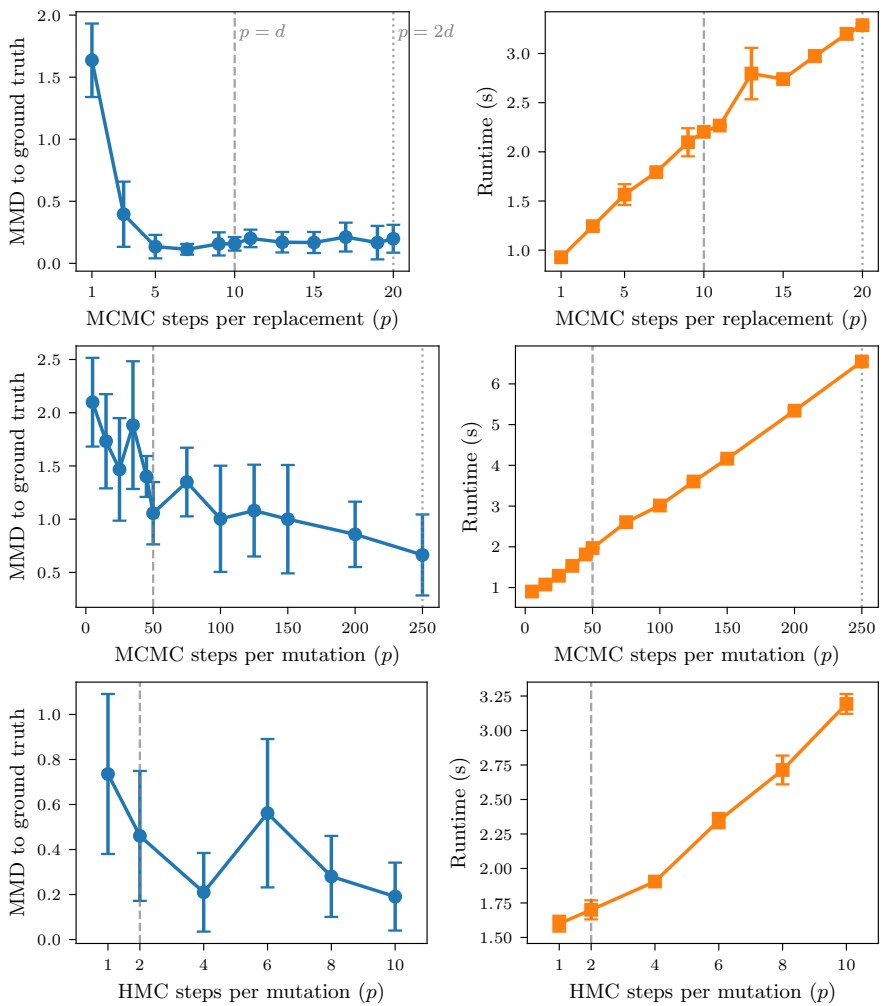

Figure 10: Ablation of MCMC chain length $p$ on $d$=10 MoG. Top: NSS ($p = 1$–$20$). Middle: SMC-RW ($p = 5$–$250$). Bottom: SMC-HMC ($p = 1$–$10$, trajectory length $L = 5$).

NSS plateaus around $p \approx 7$ with MMD $\approx 0.11$; we use $p = d$ for consistency. SMC-RW and SMC-HMC improve more slowly with increased $p$. The optimal $p$ is problem-specific; for challenging problems, a similar ablation is recommended. Assuming a near optimally tuned Gaussian Random Walk, with acceptance rate around 0.234 (Geyer, 1992), attempting $p = 5d$ moves will yield a similar effort to NSS (which is rejection free) making $p = d$ moves. We note that the mean values of slice walk lengths in table 5 are around 5, so the total evaluations are indeed similar between the Gaussian walk and HRSS, with some loss of parallel efficiency incurred for HRSS. Despite this random walk scaling, NSS can still extend to hundreds of dimensions (Yallup, 2025). Going much beyond this requires more sophisticated constrained samplers, which we explore more in section E.

## C  GPU Scaling and Vectorization

With the increased proliferation of Neural Network surrogates, neural Simulation-Based Inference, and the general utilization of GPU acceleration in scientific problems, sampling algorithms that efficiently leverage parallel hardware have become essential. However, implementing typically control-flow-heavy MCMC methods effectively on architectures like GPUs presents challenges (Hoffman & Sountsov, 2022; Hoffman et al., 2021).

The efficiency of this parallelization is limited by the inner MCMC step, here Hit-and-Run Slice Sampling. As discussed in section A.7, slice sampling involves a variable number of likelihood evaluations per call (e.g., during stepping-out). When executing $k$ chains in parallel (e.g., via `jax.vmap`), the time for the collective step is determined by the chain requiring the maximum number of evaluations. However, the cost concentration property of HRSS (section E) ensures that the per-chain variability remains low, enabling significant wall-clock time reductions on highly parallel hardware.

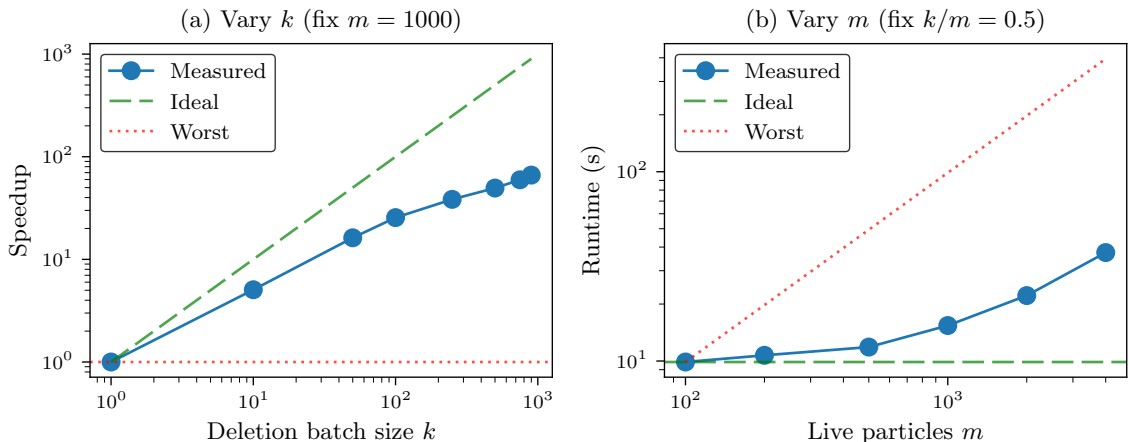

Figure 11: GPU scaling ablation for NSS on GP hyperparameter inference (Airline Passengers, $d$=11). Left: runtime vs. deletion batch size $k$. Right: $\ln Z$ accuracy vs. $k$. The corresponding GermanCredit scaling results are shown in fig. 2 in the main text.

Detailed scaling results are provided in table 10. Note that timings in this section exclude JIT compilation overhead to isolate the scaling behavior of the algorithm itself. For the GP task (fig. 11), increasing the deletion batch size $k$ from 10 to 900 (with fixed $m = 1000$) reduces runtime by 13× (152s to 12s) while maintaining consistent $\ln Z$ estimates until $k/m > 0.5$. The GermanCredit scaling (fig. 2) shows even better parallel speedups (28×), consistent with the higher per-evaluation compute cost of this problem.

The population size scaling is particularly striking for GermanCredit: increasing $m$ from 100 to 4000 (a 40× increase in particles) only increases runtime by 1.5× (8.2s to 12.6s), while reducing $\sigma(\ln Z)$ from 0.80 to 0.13. This near-constant runtime with increasing population size demonstrates effective GPU utilization on problems that parallelize well on the chosen hardware. Gaussian Process inference is somewhat bottlenecked by the $O(N^3)$ Cholesky factorization, leading to more modest scaling. The potential for GPU acceleration on new tasks depends on the per-evaluation compute cost and parallelizability of the likelihood function.

### C.1  Comparison with existing Nested Sampling implementations

Existing prominent Nested Sampling implementations, such as PolyChord (Handley et al., 2015) and UltraNest (Buchner, 2021b), were built to target CPU architectures and rely on MPI-based parallelism, with features (clustering, unit-hypercube prior transforms, dynamic population sizes) that are not well-suited to GPU execution (see section 3 for discussion). As such it is hard to construct a fair comparison with our GPU-accelerated NSS implementation. Despite these architectural differences, we find the performance (in terms of quality metrics) on difficult problems to be potentially superior to these existing codes, with a significant runtime advantage.

Table 10: GPU scaling ablation comparing GP hyperparameter inference (Airline Passengers, $d$=11) and GermanCredit logistic regression ($d$=25). For each problem: top rows vary deletion batch size $k$ with fixed $m = 1000$; bottom rows vary population size $m$ with fixed $k/m = 0.5$.

| Problem | Study | Parameters | Time (s) | $\ln Z$ | $\sigma(\ln Z)$ | Evals ($\times 10^6$) |
|---|---|---|---|---|---|---|
| GP Airline Passengers [$d$=11] | Deletion batch ($m$=1000) | $k$=10, $k/m$=0.01 | 152.3 | 25.21 | 0.17 | 1.85 |
| | | $k$=50, $k/m$=0.05 | 47.5 | 24.88 | 0.19 | 1.82 |
| | | $k$=100, $k/m$=0.10 | 30.2 | 26.12 | 0.18 | 1.75 |
| | | $k$=250, $k/m$=0.25 | 20.0 | 26.03 | 0.21 | 1.61 |
| | | $k$=500, $k/m$=0.50 | 15.6 | 25.38 | 0.19 | 1.31 |
| | | $k$=750, $k/m$=0.75 | 12.9 | 24.19 | 0.23 | 1.02 |
| | | $k$=900, $k/m$=0.90 | 11.6 | 23.03 | 0.38 | 0.79 |
| | Population size ($k/m$=0.5) | $m$=100, $k$=50 | 9.9 | 22.57 | 0.67 | 0.15 |
| | | $m$=200, $k$=100 | 10.7 | 23.73 | 0.40 | 0.27 |
| | | $m$=500, $k$=250 | 11.9 | 25.27 | 0.27 | 0.67 |
| | | $m$=1000, $k$=500 | 15.4 | 24.97 | 0.24 | 1.33 |
| | | $m$=2000, $k$=1000 | 22.2 | 26.50 | 0.14 | 2.57 |
| | | $m$=4000, $k$=2000 | 37.4 | 26.21 | 0.10 | 5.15 |
| German Credit [$d$=25] | Deletion batch ($m$=1000) | $k$=10, $k/m$=0.01 | 193.9 | -528.99 | 0.23 | 21.5 |
| | | $k$=50, $k/m$=0.05 | 52.4 | -529.17 | 0.21 | 21.1 |
| | | $k$=100, $k/m$=0.10 | 31.8 | -529.00 | 0.24 | 20.5 |
| | | $k$=250, $k/m$=0.25 | 16.4 | -529.62 | 0.21 | 19.0 |
| | | $k$=500, $k/m$=0.50 | 10.1 | -529.91 | 0.29 | 16.0 |
| | | $k$=750, $k/m$=0.75 | 8.3 | -529.77 | 0.33 | 11.9 |
| | | $k$=900, $k/m$=0.90 | 7.0 | -529.95 | 0.47 | 8.6 |
| | Population size ($k/m$=0.5) | $m$=100, $k$=50 | 8.2 | -528.09 | 0.80 | 1.6 |
| | | $m$=200, $k$=100 | 9.0 | -529.56 | 0.62 | 3.2 |
| | | $m$=500, $k$=250 | 9.4 | -528.43 | 0.39 | 7.7 |
| | | $m$=1000, $k$=500 | 9.9 | -529.47 | 0.25 | 15.8 |
| | | $m$=2000, $k$=1000 | 10.7 | -529.01 | 0.19 | 31.2 |
| | | $m$=4000, $k$=2000 | 12.6 | -529.17 | 0.13 | 62.3 |

To demonstrate this we consider the Mauna Loa dataset hyperparameter marginalization problem from section 4.3, comparing NSS against UltraNest and JAXNS (Albert, 2020). UltraNest implements a variety of NS methods; we pick two that closest match NSS: SliceSampler and a vectorized PopSliceSampler. JAXNS is a JAX-native NS implementation, but executes its MCMC chains sequentially and lacks the batched parallelism of NSS; it also forces all computation into 64-bit precision. We use the same population size and termination criterion across all methods, and run on the same NVIDIA A100 hardware with the same JIT-compiled likelihood (enabling vectorized likelihoods in UltraNest).

Table 11: Comparison of NSS with existing NS implementations on Mauna Loa GP ($d$=11 spectral mixture kernel).

| Method | $\ln Z$ | Time (s) | Evals | Test NLL | Test RMSE |
|---|---|---|---|---|---|
| NSS | 870.02 $\pm$ 0.41 | 227 | 1050579 | 32.6466 | 0.2690 |
| UltraNest (PopSliceSampler) | 803.40 $\pm$ 0.73 | 2813 | 15823697 | 141.1388 | 0.3519 |
| UltraNest (SliceSampler) | 881.91 $\pm$ 0.69 | 6785 | 2711842 | 50.6655 | 0.2522 |
| JAXNS | 799.86 $\pm$ 0.36 | 2611 | 1977399 | 125.1613 | 0.4160 |

The results in table 11 show NSS achieves a 12$\times$ to 30$\times$ speedup over UltraNest depending on configuration, while achieving substantially better predictive NLL and using fewer likelihood evaluations. JAXNS, despite operating in the same JAX ecosystem, is over 10$\times$ slower than NSS and produces substantially worse evidence estimates and predictive metrics. This likelihood is relatively bottlenecked by Cholesky factorization, so we expect even larger speedups on problems with more parallelizable likelihoods.

# D    Technical Details

## D.1    Sample Quality Metrics

To provide a principled comparison of posterior sample quality, we employ two integral probability metrics to measure the statistical distance between the samples $X = \{x_i\}_{i=1}^n$ generated by a given method and our ground truth reference samples $Y = \{y_j\}_{j=1}^m$ (either analytically known or from exhaustive NUTS runs). For our analysis, we draw $n = m = 1000$ samples for each comparison.

**Maximum Mean Discrepancy (MMD).**   We use the kernel two-sample discrepancy MMD Gretton et al. (2012), which measures the distance between the mean embeddings of two sample sets in a reproducing kernel Hilbert space (RKHS). For a characteristic kernel $k$, the squared MMD is

$$\mathrm{MMD}^2(X, Y) = \mathbb{E}_{x,x' \sim X}\big[k(x, x')\big] - 2\,\mathbb{E}_{x \sim X,\, y \sim Y}\big[k(x, y)\big] + \mathbb{E}_{y,y' \sim Y}\big[k(y, y')\big].$$

With a characteristic kernel, $\mathrm{MMD}(X, Y) = 0$ if and only if $X$ and $Y$ are drawn from the same distribution. We use a Gaussian (RBF) kernel

$$k(x, y) = \exp\big(-\|x - y\|^2/(2\sigma^2)\big),$$

with bandwidth $\sigma$ set by the median heuristic. MMD is particularly effective at detecting differences in the shape and moments of distributions.

**Sliced 2-Wasserstein Distance ($W_2$).**   We compute the sliced 2-Wasserstein distance (Bonneel et al., 2015), which provides a computationally efficient and statistically robust approximation to the full Wasserstein distance, particularly in higher dimensions. The sliced $W_2$ distance projects both sample sets onto random one-dimensional directions and computes the exact 1D Wasserstein distance along each projection.

For a unit vector $\theta \in \mathbb{S}^{d-1}$, let $X_\theta = \{\langle x_i, \theta \rangle\}$ and $Y_\theta = \{\langle y_j, \theta \rangle\}$ denote the projected samples. The 1D Wasserstein distance admits a closed-form solution: for sorted samples, $W_2^2(X_\theta, Y_\theta) = \frac{1}{n}\sum_{i=1}^n (X_\theta^{(i)} - Y_\theta^{(i)})^2$, where superscripts denote order statistics. The sliced $W_2$ distance averages over random projections:

$$W_2(X, Y) = \big(\mathbb{E}_{\theta \sim \mathrm{Uniform}(\mathbb{S}^{d-1})}\big[W_2^2(X_\theta, Y_\theta)\big]\big)^{1/2}.$$

We approximate this expectation using 200 random projection directions. This approach avoids the regularization bias of entropic optimal transport methods (Cuturi & Doucet, 2014) and scales linearly in sample size, making it well-suited for comparing high-dimensional posterior samples.

**Predictive metrics for GP regression.**   For the GP hyperparameter marginalization experiments (section 4.3), we evaluate predictive performance on a held-out test set $\{(x_{*,i}, y_{*,i})\}_{i=1}^{N_*}$ using two standard metrics.

The *test negative log-likelihood* (Test NLL) measures how well the posterior predictive distribution explains unseen observations. For each posterior hyperparameter sample $\theta_s$ ($s = 1, \ldots, S$), the GP predictive distribution at a test input $x_*$ is Gaussian with mean $\mu_s(x_*)$ and variance $\sigma_{f,s}^2(x_*) + \sigma_{n,s}^2$, where $\sigma_{f,s}^2$ is the latent GP predictive variance and $\sigma_{n,s}^2$ is the observation noise (both functions of $\theta_s$). We marginalize over samples pointwise via a mixture:

$$\log p(y_{*,i} \mid \mathcal{D}) = \log\left(\frac{1}{S}\sum_{s=1}^S \mathcal{N}\big(y_{*,i} \mid \mu_s(x_{*,i}),\ \sigma_{f,s}^2(x_{*,i}) + \sigma_{n,s}^2\big)\right),$$

computed numerically using logsumexp for stability, and the total NLL is

$$\mathrm{NLL} = -\sum_{i=1}^{N_*} \log p(y_{*,i} \mid \mathcal{D}).$$

Lower NLL indicates better-calibrated predictive uncertainty. The *root mean square error* (RMSE) evaluates point prediction accuracy:

$$\text{RMSE} = \sqrt{\frac{1}{N_*} \sum_{i=1}^{N_*} \left(y_{*,i} - \hat{y}_{*,i}\right)^2},$$

where $\hat{y}_{*,i} = \frac{1}{S} \sum_{s=1}^{S} \mu_s(x_{*,i})$ is the posterior predictive mean averaged over hyperparameter samples. While RMSE captures the accuracy of the mean prediction, it does not assess uncertainty calibration; the combination of NLL and RMSE therefore distinguishes methods that produce similar point estimates but differ in their uncertainty quantification (table 3).

## D.2 Nested Sampling Volume Estimation

### D.2.1 Computation of the effective number of live particles

Our implementation performs *batched* NS updates: at each outer iteration we remove the $k$ live points with highest energies and then generate $k$ replacements, all under the same constraint threshold $E_{\text{batch}}$ (the $k$-th worst energy). In the code, all replacements are therefore assigned the same *birth level* $E^{\text{birth}} = E_{\text{batch}}$.

For volume accounting it is convenient to *unroll* each batched deletion into a sequence of $k$ single-death events *without replenishment* (Fowlie et al., 2021). Under the usual NS idealisation that the live set is i.i.d. from the constrained prior, the $k$ removed points are precisely the top $k$ order statistics of $m$ draws, and this unrolling yields the correct joint distribution for the associated prior volumes. This is mathematically equivalent to the joint volume contraction $t \sim \text{Beta}(m-k+1, k)$ derived by Fowlie et al. (2021) for likelihood plateaus, and applies generally to the joint compression of any $k$ removed points, ensuring that evidence estimation remains unbiased. Concretely, within a batch we assign the $j$-th removed point ($j = 1, \ldots, k$, ordered from worst to best within the batch) an effective live count

$$n_{\text{live},(j)} = m - (j - 1). \tag{19}$$

After the $k$ deaths, the $k$ births at the batch threshold restore the live set to size $m$ for the next outer iteration.

This "within-batch" live-count schedule is what we use in the post-hoc shrinkage simulation in section D.2.2. (For $k = 1$ it reduces to the standard constant-$m$ setting.)

### D.2.2 Estimation of prior volumes and propagation of geometric uncertainty

The deterministic approximation $\Delta \log X \approx -k/m$ in the main text is useful for exposition, but in experiments we propagate the canonical NS *geometric* uncertainty by Monte Carlo simulation of volume shrinkage factors.

For a single death event with $n_{\text{live}}$ live points, the shrinkage ratio $t := X_i/X_{i-1}$ satisfies $t \sim \text{Beta}(n_{\text{live}}, 1)$, equivalently

$$\Delta \log X_i := \log X_i - \log X_{i-1} = \frac{1}{n_{\text{live}}} \log s, \qquad s \sim \mathcal{U}(0,1),$$

(or $\Delta \log X_i = -e/n_{\text{live}}$ with $e \sim \text{Exp}(1)$). Starting from $X_0 = 1$, we obtain a full volume trajectory by iterating $X_i = X_{i-1} t_i$.

In the batched setting, we apply this shrinkage update to each of the $k$ unrolled deaths within a batch using $n_{\text{live},(j)} = m - (j - 1)$, producing volumes $X_{b,0} \to X_{b,1} \to \cdots \to X_{b,k}$ for batch $b$.

Repeating the shrinkage simulation $R$ times (we use $R = 100$) yields an ensemble of volume trajectories; after concatenating all dead points in likelihood order we form (trapezoidal) prior-mass elements

$$dX_i^{(r)} := \frac{X_{i-1}^{(r)} - X_{i+1}^{(r)}}{2}, \qquad X_0^{(r)} := 1, \quad X_{N+1}^{(r)} := 0,$$

and corresponding quadrature weights at inverse temperature $\beta$,

$$w_i^{(r)}(\beta) := \exp(-\beta E_i)\, dX_i^{(r)}, \qquad \widehat{Z}^{(r)}(\beta) = \sum_{i=1}^{N} w_i^{(r)}(\beta).$$

We compute uncertainty estimates for $\log Z$ from the empirical dispersion of $\{\log \widehat{Z}^{(r)}(\beta)\}_{r=1}^{R}$.

For posterior diagnostics (ESS and resampling), we collapse the $R$ stochastic weights by averaging in log-space (geometric-mean weights)

$$\tilde{w}_i(\beta) := \exp\left(\frac{1}{R}\sum_{r=1}^{R}\log w_i^{(r)}(\beta)\right),$$

and report the Kish effective sample size

$$\mathrm{ESS}(\beta) = \frac{\left(\sum_{i=1}^{N}\tilde{w}_i(\beta)\right)^2}{\sum_{i=1}^{N}\tilde{w}_i(\beta)^2} = \frac{1}{\sum_{i=1}^{N}\bar{w}_i(\beta)^2}, \qquad \bar{w}_i(\beta) := \frac{\tilde{w}_i(\beta)}{\sum_{j=1}^{N}\tilde{w}_j(\beta)}.$$

This captures the standard NS geometric uncertainty induced by random volume contraction, but does not account for additional error sources from imperfect constrained mixing; we therefore complement it with repeated runs on different seeds and initial conditions for some experiments (section 4.3 and section B.6). Various diagnostics and corrections to account for these effects have been proposed (Prathaban & Handley, 2024; Higson et al., 2019).

## E  Optimal scaling of hit-and-run slice sampling

This appendix develops a principled tuning theory for Hit-and-Run Slice Sampling (HRSS), with the goal of explaining how its per-step computational cost behaves and how the slice-width parameter should scale with dimension in the regimes relevant to Nested Slice Sampling. The key results are a closed-form expression for the expected HRSS cost in terms of the chord length of the one-dimensional slice (theorem 1); a unique optimal width when the slice length is fixed (theorem 2) and an asymptotically optimal global width for sampling uniformly from high-dimensional ellipsoids, yielding the characteristic $d^{-1/2}$ scaling and identifying the leading dependence on the ellipsoid geometry (theorem 3). The section is organised as follows. Section E.1 collects technical probabilistic lemmas and establishes the convergence results needed to control expectations in high dimension, which it then specialises to ellipsoids to obtain chord-length asymptotics and related geometric inputs for the tuning rule. Section E.1.4 derives the HRSS cost formula and the optimal-width results (fixed-slice and ellipsoid-wide). Section E.2 validates the theoretical predictions for both the cost and the optimal width, and section E.2.2 quantifies how strongly per-step costs concentrate near the optimum, motivating efficient vectorised implementations.

### E.1  Proofs

#### E.1.1  Technical lemmas

A *Polish* space is a topological space which is separable and admits a complete metrisation (Kallenberg (1997), Chapter 1). This is not a particularly restrictive assumption, all spaces we consider are Polish. We denote convergence in distribution by $\xrightarrow{\mathcal{D}}$, almost sure convergence by $\xrightarrow{a.s.}$ and convergence in $L^1$ by $\xrightarrow{L^1}$. We say that a family of random variables $\{X_n\}_{n\geq 1}$ is *uniformly integrable* (Rao & Ren (1991), Theorem 2) if

$$\lim_{x\to\infty}\sup_{n\geq 1}\mathbb{E}\left[|X_n|1_{|X_n|>x}\right] = 0. \tag{20}$$

The Gamma$(\alpha, \theta)$ distribution with shape and scale parameters $\alpha$ and $\theta$, respectively, is supported on $[0, \infty)$ and has density

$$x \mapsto \frac{1}{\Gamma(\alpha)\theta^\alpha}x^{\alpha-1}e^{-x/\theta}, \tag{21}$$

where $\Gamma$ is the Gamma function. For any real $p > -\alpha$, the moments are obtained by direct integration,

$$\mathbb{E}[X^p] = \theta^p \frac{\Gamma(\alpha + p)}{\Gamma(\alpha)}. \tag{22}$$

The Beta$(\alpha, \beta)$ distribution is supported on $[0, 1]$ and has density

$$x \mapsto \frac{1}{\mathrm{B}(\alpha, \beta)} x^{\alpha - 1} (1 - x)^{\beta - 1}, \tag{23}$$

where B is the Beta function. For any real $p > -\alpha$, the moments are obtained by direct integration,

$$\mathbb{E}[X^p] = \frac{\Gamma(\alpha + \beta)\Gamma(\alpha + p)}{\Gamma(\alpha + \beta + p)\Gamma(\alpha)}. \tag{24}$$

**Lemma 1.** *For each $p \geq 0$, there exists a finite constant $C_p$ such that for all $z \geq 1$,*

$$\frac{\Gamma(z)}{\Gamma(z + p)} \leq C_p z^{-p}, \tag{25}$$

*where $\Gamma$ is the Gamma function.*

*Proof.* It suffices to show $\Gamma(z + p)/\Gamma(z) \geq c_p z^p$ for some $c_p > 0$ and then set $C_p = 1/c_p$.

Fix $p \geq 0$. Write $p = n + a$ with $n = \lfloor p \rfloor \in \mathbb{N}$ and $a = p - n \in [0, 1)$. For any $z \geq 1$,

$$\frac{\Gamma(z + p)}{\Gamma(z)} = \frac{\Gamma(z + n + a)}{\Gamma(z + n)} \frac{\Gamma(z + n)}{\Gamma(z)}. \tag{26}$$

To bound the first ratio, we use Gautschi's inequality (Olver (2010), 5.6.4), which states that for $x > 0$ and $a \in (0, 1)$

$$x^{1-a} < \frac{\Gamma(x + 1)}{\Gamma(x + a)} < (x + 1)^{1-a}. \tag{27}$$

Inverting the middle ratio and multiplying by $\frac{\Gamma(x+1)}{\Gamma(x)} = x$, we obtain

$$\frac{x}{(x + 1)^{1-a}} < \frac{\Gamma(x + a)}{\Gamma(x)} < x^a. \tag{28}$$

Now using the bound

$$\frac{x}{(x + 1)^{1-a}} = x^a \left( \frac{x}{x + 1} \right)^{1-a} \geq 2^{a-1} x^a, \tag{29}$$

valid for $x \geq 1$, because $\frac{x}{x+1} \geq \frac{1}{2}$ and $1 - a \in (0, 1]$. Therefore,

$$\frac{\Gamma(z + n + a)}{\Gamma(z + n)} \geq 2^{a-1}(z + n)^a \geq 2^{a-1} z^a, \tag{30}$$

since $z + n \geq z$ and $a > 0$.

To bound the second ratio, we use the recurrence relation of the Gamma function:

$$\frac{\Gamma(z + n)}{\Gamma(z)} = \prod_{k=0}^{n-1} (z + k) \geq z^n. \tag{31}$$

Combining the two bounds, we have shown that for all $z \geq 1$ and $a \in (0, 1)$,

$$\frac{\Gamma(z + p)}{\Gamma(z)} \geq z^n \cdot 2^{a-1} z^a = 2^{a-1} z^p. \tag{32}$$

To handle the edge case $a = 0$, in which case $p$ is an integer, we note that

$$\frac{\Gamma(z+p)}{\Gamma(z)} = \prod_{k=0}^{p-1}(z+k) \geq z^p. \tag{33}$$

Therefore, for all $p \geq 0$ and $z \geq 1$,

$$\frac{\Gamma(z+p)}{\Gamma(z)} \geq c_p z^p \tag{34}$$

with

$$c_p = \begin{cases} 1, & \text{if } p \in \mathbb{N}, \\ 2^{a-1}, & \text{if } p \notin \mathbb{N}, \text{ where } a = p - \lfloor p \rfloor. \end{cases} \tag{35}$$

$\square$

**Lemma 2.** *Let $\{X_n\}_{n \geq 1}$ be a family of random variables and fix $\delta > 0$. If $\sup_{n \geq 1} \mathbb{E}[|X_n|^{1+\delta}] < \infty$, then $\{X_n\}_{n \geq 1}$ is uniformly integrable.*

*Proof.* By Vallée-Poussin's theorem (Rao & Ren (1991), Theorem 2), uniform integrability is equivalent to the existence of a convex function $\phi : \mathbb{R} \to [0, \infty)$ with $\phi(0) = 0$, $\phi(-x) = \phi(x)$, $\phi(x)/x \to \infty$ as $x \to \infty$ and $\sup_{n \geq 1} \mathbb{E}[\phi(X_n)] < \infty$. Therefore, a sufficient condition is that $\sup_{n \geq 1} \mathbb{E}[|X_n|^{1+\delta}] < \infty$ for some $\delta > 0$, since $x \mapsto |x|^{1+\delta}$ with $\delta > 0$ is convex and satisfies the growth condition. $\square$

**Lemma 3.** *Let $S$ be a Polish space and $X_n \xrightarrow{\mathcal{D}} X$ in $S$. Let $f : S \to \mathbb{R}$ be continuous. Assume that the family $\{f(X_n)\}_n$ is uniformly integrable. Then, $\mathbb{E}[f(X_n)] \to \mathbb{E}[f(X)]$.*

*Proof.* Because $S$ is Polish (hence separable) and $X_n \xrightarrow{\mathcal{D}} X$, Skorokhod's representation theorem (Kallenberg (1997), Theorem 5.31) gives a probability space on which there exist random variables $\tilde{X}_n$ and $\tilde{X}$ with the same laws as $X_n$ and $X$, respectively, i.e. $\tilde{X}_n \sim X_n$ and $\tilde{X} \sim X$, respectively, such that $\tilde{X}_n \xrightarrow{a.s.} \tilde{X}$.

Since $f$ is continuous, it follows that $f(\tilde{X}_n) \xrightarrow{a.s.} f(\tilde{X})$ by the continuous mapping theorem (Van der Vaart (2000), Theorem 2.3). As $\{f(\tilde{X}_n)\}$ have the same laws as $\{f(X_n)\}$ and uniform integrability depends only on the laws (eq. (20)), they are also uniformly integrable.

Set $Y_n = f(\tilde{X}_n)$ and $Y = f(\tilde{X})$. We have $Y_n \xrightarrow{a.s.} Y$ (hence $Y_n \xrightarrow{p} Y$) and $\{Y_n\}$ is uniformly integrable, so by the Vitali convergence theorem (Bogachev (2007), Theorem 4.5.4), $Y_n \xrightarrow{L^1} Y$ and therefore $\mathbb{E}[Y_n] \to \mathbb{E}[Y]$. Because $\tilde{X}_n \sim X_n$ and $\tilde{X} \sim X$, we have $\mathbb{E}[f(\tilde{X}_n)] = \mathbb{E}[f(X_n)]$ and $\mathbb{E}[f(\tilde{X})] = \mathbb{E}[f(X)]$. Hence, $\mathbb{E}[f(X_n)] \to \mathbb{E}[f(X)]$. $\square$

**Lemma 4.** *Fix $a, b > 0$ and let $U \sim \text{Gamma}(a, 1)$ and $V_b \sim \text{Gamma}(b, 1)$ be independent. Define $X_b := \frac{U}{U+V_b}$. Then, $bX_b \xrightarrow{\mathcal{D}} U$ as $b \to \infty$.*

*Proof.* Write $bX_b = \frac{U}{U/b + V_b/b}$. Since $\mathbb{E}[V_b] = b$ and $\text{Var}[V_b] = b$, it holds by Chebychev's inequality that for any $\epsilon > 0$,

$$P\left(\left|\frac{V_b}{b} - 1\right| > \epsilon\right) \leq \frac{1}{\epsilon^2} \text{Var}\left(\frac{V_b}{b}\right) = \frac{1}{b\epsilon^2} \to 0, \tag{36}$$

as $b \to \infty$. Hence, $\frac{V_b}{b} \xrightarrow{p} 1$. Since $U$ has a fixed distribution independent of $b$, $\mathbb{E}[U] = a < \infty$ for all $b$. By Markov's inequality, it follows that for all $\epsilon > 0$,

$$P\left(\left|\frac{U}{b}\right| > \epsilon\right) = P(U > \epsilon b) \leq \frac{\mathbb{E}[U]}{\epsilon b} \to 0. \tag{37}$$

Thus, $\frac{U}{b} \xrightarrow{p} 0$. It follows that $\frac{V_b}{b} + \frac{U}{b} \xrightarrow{p} 1$. Slutsky's theorem (Van der Vaart (2000), Lemma 2.8) finally gives $bX_b \xrightarrow{\mathcal{D}} U$. $\square$

### E.1.2 Convergence of expectations in a ball

Let the random variable $y_d$ be uniform in the $d$-dimensional unit ball $\{x \in \mathbb{R}^d : \|x\| \leq 1\}$, where $\|\cdot\|$ is the $L^2$ norm. Write this as the polar decomposition $y_d = \rho_d U_d$ with radius $\rho_d = \|y_d\| \in [0,1]$ and unit vector $U_d = \frac{y_d}{\|y_d\|} \in S^{d-1}$, where $S^{d-1} = \{x \in \mathbb{R}^d : \|x\| = 1\}$ is the unit sphere. $\rho_d$ and $U_d$ are independent. For any fixed unit vector $u$, define

$$X_d := (U_d \cdot u)^2 \in [0,1] \quad \text{and} \quad Y_d := 1 - \rho_d^2 \in [0,1] \tag{38}$$

It follows that $Y_d$ and $X_d$ are independent.

Define

$$Z_d := \rho_d^2(U_d \cdot u)^2 + (1 - \rho_d^2) = X_d + Y_d - X_d Y_d \in [0,1]. \tag{39}$$

Introduce the scaled variables

$$S_d := dX_d \in [0,d], \qquad T_d := dY_d \in [0,d] \quad \text{and} \quad Q_d := dZ_d \in [0,d]. \tag{40}$$

It follows that $S_d$ and $T_d$ are independent.

An exact algebraic identity is given by $Q_d = S_d + T_d - \frac{T_d S_d}{d}$. Let

$$P_d := S_d + T_d \quad \text{and} \quad \epsilon_d := T_d S_d. \tag{41}$$

Then we have the identity

$$Q_d = P_d - \frac{\epsilon_d}{d}. \tag{42}$$

**Lemma 5.** $X_d \sim \text{Beta}\left(\frac{1}{2}, \frac{d-1}{2}\right)$ and $Y_d \sim \text{Beta}\left(1, \frac{d}{2}\right)$.

*Proof.* The first statement follows from $V = U_d \cdot u$ having density $f_V(v) \propto (1 - v^2)^{(d-3)/2}$ by rotational symmetry, taking the square and comparing with the Beta density (eq. (23)). The second statement follows from the radius $\rho_d$ having distribution $f_{\rho_d}(r) = dr^{d-1}$ on $[0,1]$. $\square$

**Lemma 6.** $Q_d \xrightarrow{\mathcal{D}} Q$ as $d \to \infty$, where $Q \sim \text{Gamma}\left(\frac{3}{2}, 2\right)$.

*Proof.* If $X \sim \text{Beta}(a,b)$, then $X = \frac{A}{A+B}$ where $A \sim \text{Gamma}(a,1)$ and $B \sim \text{Gamma}(b,1)$ are independent.

By lemma 5, we write $X_d$ as $X_d = \frac{A_X}{A_X + B_X}$ with $A_X \sim \text{Gamma}(\frac{1}{2}, 1)$ and $B_X \sim \text{Gamma}(\frac{d-1}{2}, 1)$. By lemma 4, $\frac{d-1}{2} X_d \xrightarrow{\mathcal{D}} A_X$ and hence $S_d = dX_d = \frac{d}{(d-1)/2} \frac{d-1}{2} X_d \xrightarrow{\mathcal{D}} 2A_X =: S$, where the random variable $S$ has distribution $\text{Gamma}(\frac{1}{2}, 2)$.

Similarly, write $Y_d$ as $Y_d = \frac{A_Y}{A_Y + B_Y}$, where $A_Y \sim \text{Gamma}(1,1)$ and $B_Y \sim \text{Gamma}(\frac{d}{2}, 1)$. It follows that $T_d = dY_d = \frac{d}{d/2} \frac{d}{2} Y_d \xrightarrow{\mathcal{D}} 2A_Y =: T$, where $T$ has distribution $\text{Gamma}(1, 2)$.

We have shown that marginal convergence of $S_d \xrightarrow{\mathcal{D}} S$ and $T_d \xrightarrow{\mathcal{D}} T$ holds, i.e. $S_d$ and $T_d$ converge individually in distribution. Moreover, we have that for any finite $d$, $S_d$ and $T_d$ are independent. We proceed to show that $S_d$ and $T_d$ converge jointly as the random vector $(S_d, T_d)$ and that convergence preserves independence. Since $S_d$ and $T_d$ are independent, the joint characteristic function factorises:

$$\phi_{(S_d, T_d)}(s,t) = \phi_{S_d}(s)\phi_{T_d}(t). \tag{43}$$

From marginal convergence, we have pointwise $\phi_{S_d}(s) \to \phi_S(s)$ and $\phi_{T_d}(t) \to \phi_T(t)$ for all $s$ and $t$. Therefore,

$$\phi_{(S_d, T_d)}(s,t) = \phi_{S_d}(s)\phi_{T_d}(t) \to \phi_S(s)\phi_T(t) =: \phi(s,t). \tag{44}$$

The function $\phi(s,t)$ is the characteristic function of the product measure, i.e. of the independent pair $(S, T)$, since it factorises. Note that the characteristic functions $\phi_S$ and $\phi_T$ are continuous everywhere so that

the product $\phi$ is continuous at zero. By Lévy's continuity theorem (Van der Vaart (2000), Theorem 2.13), pointwise convergence of characteristic functions to a characteristic function implies joint convergence in distribution. Therefore, $(S_d, T_d) \xrightarrow{\mathcal{D}} (S, T)$ with $S$ and $T$ independent, as required.

A consequence thereof is that $P_d := S_d + T_d \xrightarrow{\mathcal{D}} S + T$ by the continuous mapping theorem since the map $(x, y) \mapsto x + y$ is continuous. Importantly, by the properties of the Gamma distribution, $S + T \sim \text{Gamma}(\frac{3}{2}, 2)$.

Using independence and $\mathbb{E}[S_d] = d\mathbb{E}[X_d] = 1$ and $\mathbb{E}[T_d] = d\mathbb{E}[Y_d] = \frac{2d}{d+2}$, we have

$$\mathbb{E}\left[\frac{\epsilon_d}{d}\right] = \frac{1}{d}\mathbb{E}[S_d]\mathbb{E}[T_d] = \frac{2}{d+2} \to 0. \tag{45}$$

Since $\frac{\epsilon_d}{d} \geq 0$, $\mathbb{E}\left[\left|\frac{\epsilon_d}{d}\right|\right] = \mathbb{E}\left[\frac{\epsilon_d}{d}\right] \to 0$. By definition, this means that $\frac{\epsilon_d}{d} \xrightarrow{L^1} 0$. Hence, $\frac{\epsilon_d}{d} \xrightarrow{p} 0$.

Using eq. (42), it follows that $Q_d - P_d = -\epsilon_d/d \xrightarrow{p} 0$. By Slutsky's theorem, we have $Q_d \xrightarrow{\mathcal{D}} S + T =: Q$ where $Q \sim \text{Gamma}(\frac{3}{2}, 2)$. $\qquad\square$

**Lemma 7.** *The families $\{Q_d^{1/2}\}_{d\geq 1}$ and $\{Q_d^{-1/2}\}_{d\geq 1}$ are uniformly integrable.*

*Proof.* For $p \geq 0$, we have

$$\sup_d \mathbb{E}[S_d^p] = \sup_d d^p \mathbb{E}[X_d^p] \tag{46}$$

$$= \sup_d d^p \frac{\Gamma\left(\frac{1}{2} + p\right)\Gamma\left(\frac{d}{2}\right)}{\Gamma\left(\frac{1}{2}\right)\Gamma\left(\frac{d}{2} + p\right)} \qquad \text{(by lemma 5 and eq. (24))} \tag{47}$$

$$\leq \max\left\{\mathbb{E}[S_1^p], \ \sup_{d\geq 2} B_p d^p \left(\frac{d}{2}\right)^{-p}\right\} \qquad \text{(for some } B_p < \infty \text{ by lemma 1 with } z = \frac{d}{2} \geq 1\text{)} \tag{48}$$

$$\leq \max\left\{\mathbb{E}[S_1^p], \ 2^p B_p\right\} \tag{49}$$

$$< \infty. \tag{50}$$

Similarly, for $p \geq 0$, we have

$$\sup_d \mathbb{E}[T_d^p] = \sup_d d^p \mathbb{E}[Y_d^p] \tag{51}$$

$$= \sup_d d^p \frac{\Gamma\left(1 + p\right)\Gamma\left(1 + \frac{d}{2}\right)}{\Gamma\left(1\right)\Gamma\left(1 + \frac{d}{2} + p\right)} \qquad \text{(by lemma 5 and eq. (24))} \tag{52}$$

$$\leq \sup_d B_p' d^p \left(1 + \frac{d}{2}\right)^{-p} \qquad \text{(for some } B_p' < \infty \text{ by lemma 1 with } z = 1 + \frac{d}{2} \geq \frac{3}{2}\text{)} \tag{53}$$

$$\leq 2^p B_p' \qquad \text{(since } d^p \left(1 + \frac{d}{2}\right)^{-p} = \left(\frac{2d}{d+2}\right)^p \leq 2^p \text{ for } d \geq 1\text{)} \tag{54}$$

$$< \infty. \tag{55}$$

Since, for all $p \in (0, 1]$, $(a + b)^p \leq a^p + b^p$ and, for all $p \geq 1$, $(a + b)^p \leq 2^{p-1}(a^p + b^p)$, there exists a finite constant $A_p$ such that $P_d^p \leq A_p(S_d^p + T_d^p)$ for all $p > 0$. Taking expectations and using the above two bounds, we have

$$\sup_d \mathbb{E}[P_d^p] \leq A_p \left(\sup_d \mathbb{E}[S_d^p] + \sup_d \mathbb{E}[T_d^p]\right) < \infty, \tag{56}$$

for any fixed $p > 0$. Since $Q_d \leq P_d$ by application of $\epsilon_d \geq 0$ and eq. (42), $\sup_d \mathbb{E}[Q_d^p] < \infty$.

Set $p = \frac{1+\delta}{2}$ for some $\delta > 0$. Note that $Q_d \geq 0$, hence $Q_d = |Q_d|$. lemma 2 implies that $\{Q_d^{1/2}\}_{d\geq 1}$ is uniformly integrable.

Since $P_d = S_d + T_d \geq T_d$ from $S_d \geq 0$, and $x \mapsto x^{-q}$ is decreasing on $(0, \infty)$ for $q > 0$, we have $P_d^{-q} \leq T_d^{-q}$. For any $q \in (0, 1)$, we have

$$\sup_d \mathbb{E}[T_d^{-q}] = \sup_d d^{-q} \mathbb{E}[Y_d^{-q}] \qquad \text{(since } q > 0\text{)} \tag{57}$$

$$= \sup_d d^{-q} \frac{\Gamma\left(1-q\right)\Gamma\left(1+\frac{d}{2}\right)}{\Gamma\left(1\right)\Gamma\left(1+\frac{d}{2}-q\right)} \tag{58}$$

(by lemma 5, eq. (24) and since $q < 1$)

$$\leq \max\left\{\mathbb{E}[T_1^{-q}], \; \sup_{d \geq 2} d^{-q} \Gamma\left(1-q\right)\left(1+\frac{d}{2}-q\right)^q\right\} \tag{59}$$

(by Gautschi's inequality (Olver, 2010, §5.6.4), $\frac{\Gamma(z+q)}{\Gamma(z)} \leq z^q$ for $z \geq 1$ and $q \in (0, 1)$)

$$\leq \max\left\{\mathbb{E}[T_1^{-q}], \; \left(\frac{3}{2}\right)^q \Gamma(1-q)\right\} \tag{60}$$

$$< \infty. \tag{61}$$

Thus, for any $q \in (0, 1)$, $\sup_d \mathbb{E}[P_d^{-q}] < \infty$.

The AM-GM inequality gives $\epsilon_d = S_d T_d \leq \frac{1}{4}(S_d + T_d)^2 = \frac{P_d^2}{4}$. Since $0 \leq S_d, T_d \leq d$, we have $0 \leq P_d = S_d + T_d \leq 2d$. Using eq. (42) $Q_d = P_d - \frac{\epsilon_d}{d} \geq P_d - \frac{P_d^2}{4d} = P_d(1 - \frac{P_d}{4d}) \geq \frac{P_d}{2}$. Consequently, we have the upper bound $Q_d^{-1/2} \leq \sqrt{2} P_d^{-1/2}$.

Pick any $\delta \in (0, 1)$ and set $q = \frac{1+\delta}{2}$ so that $q \in (\frac{1}{2}, 1) \subset (0, 1)$. Then,

$$\sup_d \mathbb{E}[(Q_d^{-1/2})^{1+\delta}] = \sup_d \mathbb{E}[Q_d^{-q}] \leq \sup_d 2^{(1+\delta)/2} \mathbb{E}[P_d^{-q}] < \infty. \tag{62}$$

lemma 2 implies that $\{Q_d^{-1/2}\}_{d \geq 1}$ is uniformly integrable. $\qquad \square$

**Lemma 8.** *As $d \to \infty$,*

$$\mathbb{E}[Z_d^{1/2}] = \frac{\mathbb{E}[Q^{1/2}]}{\sqrt{d}}[1 + o(1)], \tag{63}$$

$$\mathbb{E}[Z_d^{-1/2}] = \sqrt{d}\,\mathbb{E}[Q^{-1/2}][1 + o(1)], \tag{64}$$

$$\frac{\mathbb{E}[Z_d^{1/2}]}{\mathbb{E}[Z_d^{-1/2}]} = \frac{2}{d}[1 + o(1)], \tag{65}$$

*where $o(1)$ is a function such that $\lim_{d \to \infty} o(1) = 0$. Here, $\mathbb{E}[Q^{1/2}] = 2\sqrt{\frac{2}{\pi}}$ and $\mathbb{E}[Q^{-1/2}] = \sqrt{\frac{2}{\pi}}$.*

*Proof.* Since $Q_d \xrightarrow{\mathcal{D}} Q$ (lemma 6) and the families $\{Q_d^{1/2}\}$ and $\{Q_d^{-1/2}\}$ are uniformly integrable (lemma 7), lemma 3 implies $\mathbb{E}[Q_d^{1/2}] \to \mathbb{E}[Q^{1/2}]$ and $\mathbb{E}[Q_d^{-1/2}] \to \mathbb{E}[Q^{-1/2}]$.

From lemma 6 and the moments of the Gamma distribution (eq. (22)), it follows that $\mathbb{E}[Q^{1/2}] = 2\sqrt{\frac{2}{\pi}}$ and $\mathbb{E}[Q^{-1/2}] = \sqrt{\frac{2}{\pi}}$. Since $\mathbb{E}[Q^{-1/2}] > 0$, we have

$$d\frac{\mathbb{E}[Z_d^{1/2}]}{\mathbb{E}[Z_d^{-1/2}]} = \frac{\mathbb{E}[Q_d^{1/2}]}{\mathbb{E}[Q_d^{-1/2}]} \to \frac{\mathbb{E}[Q^{1/2}]}{\mathbb{E}[Q^{-1/2}]} = 2, \tag{66}$$

as $d \to \infty$. Equivalently, $\frac{\mathbb{E}[Z_d^{1/2}]}{\mathbb{E}[Z_d^{-1/2}]} = \frac{2}{d}[1 + o(1)]$.

$\qquad \square$

### E.1.3 Lemmas for ellipsoids

Let the random vector $v_d \in S^{d-1}$ be distributed uniformly on the unit sphere. Let $A_d \in \mathbb{R}^{d \times d}$ be a symmetric positive definite matrix and define the quadratic form $q_d := v_d^T A_d v_d > 0$. Note that $q_d$ is a random variable. Further define the first moment $\mu_d := \frac{1}{d}\text{Tr}(A_d)$ and second moment $s_d := \frac{1}{d}\text{Tr}(A_d^2)$.

**Lemma 9** (Convergence of quadratic forms). *Consider a sequence of symmetric positive definite matrices $(A_d)_{d \geq 1}$, each with spectrum $\{\lambda_{i,d}\}_{i=1}^d$ and induced quadratic form $q_d$. Assume that the spectrum of $A_d$ is bounded uniformly away from 0 and $\infty$, i.e. there exist constants $\lambda_-$ and $\lambda_+$ with $0 < \lambda_- \leq \lambda_+ < \infty$ such that for all $d$ and all $i \in \{1, \ldots, d\}$, $\lambda_- \leq \lambda_{i,d} \leq \lambda_+$.*

*Then, as $d \to \infty$,*

$$\mathbb{E}[q_d^{1/2}] = \mu_d^{1/2} + O\left(\frac{1}{d}\right), \tag{67}$$

$$\mathbb{E}[q_d^{-1/2}] = \mu_d^{-1/2} + O\left(\frac{1}{d}\right), \tag{68}$$

$$\frac{\mathbb{E}[q_d^{-1/2}]}{\mathbb{E}[q_d^{1/2}]} = \frac{1}{\mu_d}\left[1 + O\left(\frac{1}{d}\right)\right], \tag{69}$$

$$\text{Var}(q_d) = O\left(\frac{1}{d}\right), \tag{70}$$

*where the error bound is uniform in $d$, i.e. $O\left(\frac{1}{d}\right)$ is a function such that there exists a finite $d$-independent constant $C$ with $O\left(\frac{1}{d}\right) \leq \frac{C}{d}$ for all $d$.*

*Proof.* The second moment $\mathbb{E}[v_i v_j]$ is a rank-2 isotropic symmetric tensor, so $\mathbb{E}[v_i v_j] = a\delta_{ij}$ for some $a$. Taking the trace gives $1 = \mathbb{E}[\|v\|^2] = ad$, hence $a = \frac{1}{d}$. The fourth moment $\mathbb{E}[v_i v_j v_k v_l]$ is a rank-4 isotropic symmetric tensor, so $\mathbb{E}[v_i v_j v_k v_l] = b\left(\delta_{ij}\delta_{kl} + \delta_{il}\delta_{jk} + \delta_{ik}\delta_{jl}\right)$ for some $b$. Contracting gives $\mathbb{E}[(\sum_i v_i^2)(\sum_j v_j^2)] = \mathbb{E}[(\sum_i v_i^2)^2] = 1 = b(d^2 + 2d)$ so that $b = \frac{1}{d(d+2)}$. Therefore, we are left with

$$\mathbb{E}[v_i v_j] = \frac{1}{d}\delta_{ij}, \tag{71}$$

$$\mathbb{E}[v_i v_j v_k v_l] = \frac{1}{d(d+2)}\left(\delta_{ij}\delta_{kl} + \delta_{il}\delta_{jk} + \delta_{ik}\delta_{jl}\right). \tag{72}$$

$$\tag{73}$$

The moments of $q_d$ are

$$\mathbb{E}[q_d] = \sum_{ij} a_{ij}\mathbb{E}[v_i v_j], \tag{74}$$

$$\mathbb{E}[q_d^2] = \sum_{ijkl} a_{ij}a_{kl}\mathbb{E}[v_i v_j v_k v_l]. \tag{75}$$

$$\tag{76}$$

It follows that the moments of $q_d$ are given by

$$\mathbb{E}[q_d] = \mu_d, \tag{77}$$

$$\mathbb{E}[q_d^2] = \frac{d\mu_d^2 + 2s_d}{d+2}. \tag{78}$$

$$\tag{79}$$

This implies that the variance is

$$\text{Var}(q_d) = \mathbb{E}[(q_d - \mu_d)^2] = \mathbb{E}[q_d^2] - \mathbb{E}[q_d]^2 = 2\frac{s_d - \mu_d^2}{d+2}. \tag{80}$$

Since $s_d = \frac{1}{d} \sum_i \lambda_i^2 \leq \lambda_+^2$ and $\mu_d^2 \geq 0$, we have $s_d - \mu_d^2 \leq \lambda_+^2$. Therefore

$$\mathrm{Var}(q_d) \leq \frac{2\lambda_+^2}{d+2}. \tag{81}$$

Let $f_\pm(x) = x^{\pm 1/2}$. By Taylor's theorem at fixed $d$ and using that $\mathbb{E}[q_d - \mu_d] = 0$,

$$\mathbb{E}[f_\pm(q_d)] = f_\pm(\mu_d) + \frac{1}{2} f_\pm''(\mu_d) \mathrm{Var}(q_d) + R_{\pm,d}, \tag{82}$$

where the Lagrange remainder is

$$R_{\pm,d} = \frac{1}{6} \mathbb{E}[f_\pm'''(\xi_d)(q_d - \mu_d)^3], \tag{83}$$

for some $\xi_d \in [q_d, \mu_d]$.

Note that $q_d = v_d^T A_d v_d$ is a Rayleigh quotient, so $q_d \in [\lambda_-, \lambda_+]$. Further note that $\mu_d = \mathbb{E}[q_d] = \frac{1}{d} \mathrm{Tr}(A_d)$ is the average eigenvalue, so $\mu_d \in [\lambda_-, \lambda_+]$. Hence, the third derivatives are bounded uniformly in $d$:

$$|f_+'''(x)| = \frac{3}{8x^{5/2}} \leq \frac{3}{8\lambda_-^{5/2}} =: C_+, \tag{84}$$

$$|f_-'''(x)| = \frac{15}{8x^{7/2}} \leq \frac{15}{8\lambda_-^{7/2}} =: C_-. \tag{85}$$

Using that $q_d - \mu_d \leq \lambda_+ - \lambda_-$ implies $|q_d - \mu_d|^3 \leq (\lambda_+ - \lambda_-)(q_d - \mu_d)^2$, we have

$$\mathbb{E}[|q_d - \mu_d|^3] \leq (\lambda_+ - \lambda_-)\mathrm{Var}(q_d) \tag{86}$$

$$\leq 2(\lambda_+ - \lambda_-)\frac{\lambda_+^2}{d+2} \qquad \text{(by eq. (81)).} \tag{87}$$

In conclusion, the remainders satisfy the bound

$$|R_{\pm,d}| \leq \frac{1}{6} \max\{C_+, C_-\} \mathbb{E}[|q_d - \mu_d|^3] \leq \frac{C_1}{d+2}, \tag{88}$$

for some $d$-indeppendent constant $C_1$ depending only on $\lambda_-$ and $\lambda_+$. Similarly, the second-order terms of the Taylor expansion are bounded uniformly in $d$:

$$\left| \frac{1}{2} f_+''(\mu_d) \mathrm{Var}(q_d) \right| \leq \frac{1}{8} \mu_d^{-3/2} \mathrm{Var}(q_d) \leq \frac{C_2}{d}, \tag{89}$$

$$\left| \frac{1}{2} f_-''(\mu_d) \mathrm{Var}(q_d) \right| \leq \frac{3}{8} \mu_d^{-5/2} \mathrm{Var}(q_d) \leq \frac{C_3}{d}, \tag{90}$$

for constants $C_2$ and $C_3$ depending only on $\lambda_-$ and $\lambda_+$. In summary, the errors in the Taylor expansion (eq. (82)) are uniformly bounded in $d$:

$$\mathbb{E}[q_d^{1/2}] = \mu_d^{1/2} + O\left(\frac{1}{d}\right), \tag{91}$$

$$\mathbb{E}[q_d^{-1/2}] = \mu_d^{-1/2} + O\left(\frac{1}{d}\right). \tag{92}$$

Since $\mu_d \in [\lambda_-, \lambda_+]$, this can be rewritten as

$$\mathbb{E}[q_d^{1/2}] = \mu_d^{1/2}(1 + \alpha_d), \tag{93}$$

$$\mathbb{E}[q_d^{-1/2}] = \mu_d^{-1/2}(1 + \beta_d). \tag{94}$$

where $|\alpha_d| \leq \frac{C_4}{d}$ and $|\beta_d| \leq \frac{C_5}{d}$ with constants $C_4$ and $C_5$ depending only on $\lambda_-$ and $\lambda_+$.

Since $q_d \in [\lambda_-, \lambda_+]$, we have $\mathbb{E}[q_d^{1/2}] \geq \lambda_-^{1/2}$ as well as $\mu_d^{1/2} \leq \lambda_+^{1/2}$. Hence, rearranging eq. (93) gives

$$1 + \alpha_d = \frac{\mathbb{E}[q_d^{1/2}]}{\mu_d^{1/2}} \geq \left(\frac{\lambda_-}{\lambda_+}\right)^{1/2} =: c_0 > 0. \tag{95}$$

for all $d$. This implies for the ratio

$$\left|\frac{1 + \beta_d}{1 + \alpha_d} - 1\right| = \left|\frac{\beta_d - \alpha_d}{1 + \alpha_d}\right| \leq \frac{|\beta_d| + |\alpha_d|}{|1 + \alpha_d|} \leq \frac{|\beta_d| + |\alpha_d|}{c_0} \leq \frac{C_5 + C_4}{c_0 d}. \tag{96}$$

Then,

$$\frac{\mathbb{E}[q_d^{-1/2}]}{\mathbb{E}[q_d^{1/2}]} = \frac{1}{\mu_d} \frac{1 + \beta_d}{1 + \alpha_d} = \frac{1}{\mu_d}\left[1 + O\left(\frac{1}{d}\right)\right]. \tag{97}$$

where the constant of the $O\left(\frac{1}{d}\right)$ error term only depends on $\lambda_-$ and $\lambda_+$. Therefore, the error is uniform for all $d \geq 1$.

$\square$

**Lemma 10** (Mean chord length in ellipsoids). *Let $E_d = \{x \in \mathbb{R}^d : x^T A_d x \leq 1\}$ be an ellipsoid with $A_d \in \mathbb{R}^{d \times d}$ a symmetric positive definite matrix and define $\mu_d = \frac{1}{d}\text{Tr}(A_d)$. Consider a sequence of such ellipsoids $(E_d)_{d \geq 1}$ indexed by dimension. Let the spectrum of each matrix in the corresponding sequence $(A_d)_{d \geq 1}$ satisfy the boundedness assumption in lemma 9.*

*Sample a point $x$ uniformly from $E_d$ and a direction $v$ uniformly from the unit sphere. Define the chord length as $\ell_d := |\{x + tv : t \in \mathbb{R}\} \cap E_d|$. Then, the following leading order expansion holds:*

$$\mathbb{E}[\ell_d] = 4\sqrt{\frac{2}{\pi \mu_d d}}[1 + o(1)]. \tag{98}$$

*Further, define the normalised chord length $R_d := \frac{\ell_d}{\mathbb{E}[\ell_d]}$. Then, $R_d \xrightarrow{\mathcal{D}} R_\infty$ as $d \to \infty$, where $R_\infty := \frac{Q^{1/2}}{\mathbb{E}[Q^{1/2}]}$.*

*Proof.* Draw $x_0$ uniformly from $E_d$ and a direction $v \in S^{d-1}$ uniformly on the unit sphere. Consider the line $x(t) = x_0 + tv$.

We now solve for the intersection of $x(t)$ and $E_d$. Transform to the unit ball via $x \mapsto A_d^{1/2}x$. Then $y_d := A_d^{1/2}x_0$ is uniform in the unit ball. Let $y_d = \rho_d U_d$ be the polar decomposition of $y_d$ with $\rho_d \in [0, 1]$ and $U_d \in S^{d-1}$ uniformly on the sphere. Define $p := A_d^{1/2}v$ with $q_d := \|p\|^2 = v^T A_d v$. Define the unit direction $u = \frac{p}{\|p\|} = \frac{p}{\sqrt{q_d}}$. In $y$-coordinates, the line is $y(t) = y_d + tp$ and the contraint is $\|y_d\| \leq 1$. The line intersects the ball when $\|y_d + tp\|^2 = 1$, which leads to the quadratic $q_d t^2 + 2(y_d \cdot p)t + (\rho_d^2 - 1) = 0$ with roots

$$t_\pm = \frac{1}{q_d}\left(-y_d \cdot p \pm \sqrt{(y_d \cdot p)^2 + (1 - \rho_d^2)q_d}\right). \tag{99}$$

Thus, the chord length is

$$\ell_d = \|x(t_+) - x(t_-)\| = |t_+ - t_-| = 2\sqrt{\frac{\rho_d^2(U_d \cdot u)^2 + (1 - \rho_d^2)}{q_d}}. \tag{100}$$

Now define $Z_d := \rho_d^2(U_d \cdot u)^2 + (1 - \rho_d^2)$ as in section E.1.2 so that the results therein are applicable. With these definitions, we have

$$\ell_d = 2\sqrt{\frac{Z_d}{q_d}}. \tag{101}$$

Crucially, $y_d$ is rotationally invariant and independent of $v$. Conditional on $v$ (equivalently, conditional on $u = u(v)$), the law of $(U_d \cdot u)^2$ is the same for all unit vectors $u$ by rotational invariance of $U_d$. Hence the conditional law of $Z_d$ given $v$ does not depend on $v$, and therefore $Z_d$ is independent of $v$ and of $q_d = v^T A_d v$. Therefore, the expectations factor:

$$\mathbb{E}[\ell_d] = 2\mathbb{E}[q_d^{-1/2}]\mathbb{E}[Z_d^{1/2}]. \tag{102}$$

By lemma 8 and lemma 9, we obtain

$$\mathbb{E}[\ell_d] = 2\frac{\mathbb{E}[Q^{1/2}]}{\sqrt{\mu_d d}}[1 + o(1)] = 4\sqrt{\frac{2}{\pi\mu_d d}}[1 + o(1)]. \tag{103}$$

The normalised chord length is defined as

$$R_d := \frac{\ell_d}{\mathbb{E}[\ell_d]} = \frac{q_d^{-1/2}}{\mathbb{E}[q_d^{-1/2}]}\frac{Z_d^{1/2}}{\mathbb{E}[Z_d^{1/2}]}. \tag{104}$$

Because $q_d \in [\lambda_-, \lambda_+]$, we also have $q_d^{-1/2} \in [\lambda_+^{-1/2}, \lambda_-^{-1/2}]$. Hence $\mathbb{E}[q_d^{-1/2}] \in [\lambda_+^{-1/2}, \lambda_-^{-1/2}]$ is bounded away from 0 and $\infty$ uniformly in $d$. Since the map $x \mapsto x^{-1/2}$ is Lipschitz on $[\lambda_-, \lambda_+]$ with constant $\frac{1}{2\lambda_-^{3/2}}$, we have $\mathrm{Var}(q_d^{-1/2}) \leq \frac{1}{4\lambda_-^3}\mathrm{Var}(q_d)$. By eq. (81), $\mathrm{Var}(q_d) = O(1/d)$, hence $\mathrm{Var}(q_d^{-1/2}) \to 0$. By Chebychev's inequality,

$$P\left(\left|\frac{q_d^{-1/2}}{\mathbb{E}[q_d^{-1/2}]} - 1\right| > \epsilon\right) = P\left(\left|q_d^{-1/2} - \mathbb{E}[q_d^{-1/2}]\right| > \epsilon\mathbb{E}[q_d^{-1/2}]\right) \tag{105}$$

$$\leq \frac{\mathrm{Var}(q_d^{-1/2})}{\epsilon^2\mathbb{E}[q_d^{-1/2}]^2} \tag{106}$$

$$\leq C\mathrm{Var}(q_d^{-1/2}) \tag{107}$$

$$\to 0, \tag{108}$$

as $d \to \infty$ for each fixed $\epsilon > 0$. Therefore $\frac{q_d^{-1/2}}{\mathbb{E}[q_d^{-1/2}]} \xrightarrow{P} 1$.

Write

$$\frac{Z_d^{1/2}}{\mathbb{E}[Z_d^{1/2}]} = A_d B_d \tag{109}$$

where $A_d := \frac{(dZ_d)^{1/2}}{\mathbb{E}[Q^{1/2}]}$ and $B_d := \frac{\mathbb{E}[Q^{1/2}]}{\mathbb{E}[(dZ_d)^{1/2}]}$. By lemma 6, $A_d \xrightarrow{\mathcal{D}} \frac{Q^{1/2}}{\mathbb{E}[Q^{1/2}]}$. By lemma 7, $B_d \to 1$. Then, by Slutsky's theorem we obtain $A_d B_d \xrightarrow{\mathcal{D}} \frac{Q^{1/2}}{\mathbb{E}[Q^{1/2}]}$.

A further application of Slutsky's theorem yields for eq. (104):

$$R_d \xrightarrow{\mathcal{D}} \frac{Q^{1/2}}{\mathbb{E}[Q^{1/2}]}. \tag{110}$$

$\square$

**Lemma 11** (Fixed-point contraction). *Let $R_d$ be the normalised chord length as in lemma 10 with $R_d \xrightarrow{\mathcal{D}} R_\infty$. For fixed $\kappa \geq 0$, define the function*

$$h_\kappa : [0, \infty) \to \mathbb{R}, \quad r \mapsto r\ln\left(1 + \frac{\kappa}{r}\right) \tag{111}$$

*with $h_\kappa(0) := 0$ and, for fixed $R_d$, define*

$$T_d : [0, \infty) \to \mathbb{R}, \quad \kappa \mapsto \frac{1}{2} + \mathbb{E}\left[h_\kappa(R_d)\right]. \tag{112}$$

*Let $\kappa_d$ be the solution of the fixed-point equation $\kappa_d = T_d(\kappa_d)$.*

*Then, $T_d \to T_\infty$ pointwise for each $\kappa \geq 0$ and $\kappa_d \to \kappa_\infty$, where $\kappa_\infty$ is the unique fixed point of $T_\infty$, $\kappa_\infty = T_\infty(\kappa_\infty)$.*

*Proof.* Note that $h_\kappa$ is continuous on $[0, \infty)$. For $r > 0$, $0 \leq \ln\left(1 + \frac{\kappa}{r}\right) \leq \frac{\kappa}{r}$, hence $0 \leq h_\kappa(r) \leq \kappa$. Overall, $\sup_{r \geq 0} h_\kappa(r) \leq \kappa$, so $h_\kappa$ is bounded on $[0, \infty)$.

By the portmanteau theorem, and since $R_d \xrightarrow{\mathcal{D}} R_\infty$ (lemma 10) and $h_\kappa$ is bounded and continuous, we get $\mathbb{E}[h_\kappa(R_d)] \to \mathbb{E}[h_\kappa(R_\infty)]$. It immediately follows that

$$T_d(\kappa) = \frac{1}{2} + \mathbb{E}[h_\kappa(R_d)] \to \frac{1}{2} + \mathbb{E}[h_\kappa(R_\infty)] =: T_\infty(\kappa), \tag{113}$$

pointwise for each $\kappa \geq 0$.

Define $\kappa_d$ as the unique solution of $\kappa = T_d(\kappa)$ and $\kappa_\infty$ as the unique solution of $\kappa = T_\infty(\kappa)$. We now prove that $\kappa_d \to \kappa_\infty$.

To do so, we first prove that $T_d$ and $T_\infty$ are contractions on $[c_0, \infty)$, uniformly in $d$, where we define $c_0 := \frac{1}{2} > 0$. For any $\kappa \geq c_0$ and any $r \geq 0$,

$$\frac{\partial}{\partial \kappa} h_\kappa(r) = \frac{r}{r + \kappa} \leq \frac{r}{r + c_0}, \tag{114}$$

so

$$T_d'(\kappa) = \mathbb{E}\left[\frac{\partial}{\partial \kappa} h_\kappa(R_d)\right] = \mathbb{E}\left[\frac{R_d}{R_d + \kappa}\right] \leq \mathbb{E}\left[\frac{R_d}{R_d + c_0}\right] = 1 - c_0 \mathbb{E}\left[\frac{1}{R_d + c_0}\right], \tag{115}$$

where we differentiated under the expectation using $0 \leq \frac{\partial}{\partial \kappa} h_\kappa(R_d) \leq 1$. Since the map $x \mapsto x^{-1}$ is convex on $(0, \infty)$, Jensen's inequality gives

$$\mathbb{E}\left[\frac{1}{R_d + c_0}\right] \geq \frac{1}{\mathbb{E}[R_d] + c_0} = \frac{1}{1 + c_0}. \tag{116}$$

Therefore,

$$T_d'(\kappa) \leq 1 - \frac{c_0}{1 + c_0} = \frac{1}{1 + c_0} =: \rho < 1 \tag{117}$$

for all $\kappa \geq c_0$, uniformly in $d$. The same bound holds for $T_\infty$. It follows that $T_d$ and $T_\infty$ are strict contractions on $[c_0, \infty)$ with the same contraction modulus $\rho = \frac{1}{1+c_0} < 1$, independent of $d$.

We now show the existence and uniqueness of $\kappa_d$ and $\kappa_\infty$. For each $d$, $T_d$ maps $[0, \infty)$ to $[c_0, \infty)$ and is continuous and strictly increasing ($T_d' \geq 0$). Also, $T_d(0) = c_0$ and $T_d(\kappa) \to \infty$ as $\kappa \to \infty$: for each fixed $r > 0$, $h_\kappa(r) = r \ln(1 + \kappa/r) \uparrow \infty$ as $\kappa \to \infty$, so by monotone convergence $\mathbb{E}[h_\kappa(R_d)] \to \infty$. Therefore, $T_d$ has exactly one fixed point on $[0, \infty)$ and by the contraction property, it lies in $[c_0, \infty)$. Similarly for $T_\infty$.

We now show that $|\kappa_d - \kappa_\infty| \to 0$. Write

$$|\kappa_d - \kappa_\infty| = |T_d(\kappa_d) - T_\infty(\kappa_\infty)| \tag{118}$$
$$\leq |T_d(\kappa_d) - T_d(\kappa_\infty)| + |T_d(\kappa_\infty) - T_\infty(\kappa_\infty)| \tag{119}$$
$$\leq \rho|\kappa_d - \kappa_\infty| + |T_d(\kappa_\infty) - T_\infty(\kappa_\infty)|. \tag{120}$$

Hence,

$$|\kappa_d - \kappa_\infty| \leq \frac{1}{1 - \rho}|T_d(\kappa_\infty) - T_\infty(\kappa_\infty)|. \tag{121}$$

Due to pointwise convergence of $T_d$, we have $T_d(\kappa_\infty) \to T_\infty(\kappa_\infty)$ as $d \to \infty$. Therefore the right-hand side tends to zero and we conclude $\kappa_d \to \kappa_\infty$. $\qquad\square$

### E.1.4 Optimal scaling of hit-and-run slice sampling in ellipsoids

We now consider Hit-and-Run Slice Sampling (Neal, 2003) with fixed width parameter $w$ targeting the uniform distribution on a domain $\subset \mathbb{R}^d$. We are interested in the expected cost, i.e. the number of likelihood evaluations, per step. The intersection of the domain with the chosen line, restricted to the connected component containing the starting point, is a single interval with length $\ell$. This holds for any domain (whether connected or disconnected) if we only target the connected component containing the starting point.

**Theorem 1** (Computational cost). *Let the random variables $N_{\text{out}}$ and $N_{\text{shrink}}$ denote the number of stepping-out and shrinkage steps in Hit-and-Run Slice Sampling with fixed slice width parameter $w$. Let $\ell$ be the width of the interval of the current step. Then, we have the following conditional expectations:*

$$\mathbb{E}[N_{\text{out}} \mid \ell] = \frac{\ell}{w}, \tag{122}$$

$$\mathbb{E}[N_{\text{shrink}} \mid \ell] = 1 + 2\phi\left(\frac{w}{\ell}\right), \tag{123}$$

*where*

$$\phi(u) = \frac{(1+u)\ln(1+u) - u}{u}, \tag{124}$$

*and where the expectations are taken over the sequence of random variables generated in the algorithm. Therefore, the expected number of likelihood evaluations per step, i.e. the computational cost, is*

$$\mathbb{E}[N_{\text{evals}} \mid \ell] = \frac{\ell}{w} + 1 + 2\phi\left(\frac{w}{\ell}\right). \tag{125}$$

*Proof.* We assume that the stepping-out and shrinkage procedures are used (Figure 3 and Figure 5 of Neal (2003), respectively): Suppose the slice is given by $S = [0, \ell]$ with $\ell > 0$, i.e. we pick coordinates along the slice such that $S = [0, \ell]$. We are given a current point inside the slice $x_0 \in (0, \ell)$. The bracket is initialised as $[x_0 - U, x_0 - U + w]$, where $U \sim \text{Uniform}(0, w)$, and then expanded in both directions sequentially until we step outside the domain. This yields the initial bracket $I_0 = [-a, \ell + b]$ with overshoots $a$ and $b$ satisfying $a, b \geq 0$.

The number of expansions to the left $k_L$ is given by the smallest $k \geq 0$ such that $x_0 - U - kw \leq 0$, i.e. $k_L = \lceil \frac{x_0 - U}{w} \rceil$. Similarly, the number of expansions to the right $k_R$ is given by the smallest $k \geq 0$ such that $x_0 - U + w + kw \geq \ell$, i.e. $k_R = \lceil \frac{\ell - (x_0 - U + w)}{w} \rceil = \lceil \frac{\ell - x_0}{w} - \left(1 - \frac{U}{w}\right) \rceil$. Let $V = \frac{U}{w} \sim \text{Uniform}(0, 1)$ and define $\alpha_L = \frac{x_0}{w}$ and $\alpha_R = \frac{\ell - x_0}{w}$. Then

$$k_L = \lceil \alpha_L - V \rceil, \tag{126}$$
$$k_R = \lceil \alpha_R - (1 - V) \rceil. \tag{127}$$

Since for any $\alpha \geq 0$, $\mathbb{E}[\lceil \alpha - V \rceil] = \alpha$, we obtain

$$\mathbb{E}[k_L \mid x_0, \ell] = \mathbb{E}[\lceil \alpha_L - V \rceil] = \alpha_L = \frac{x_0}{w}, \tag{128}$$

$$\mathbb{E}[k_R \mid x_0, \ell] = \mathbb{E}[\lceil \alpha_R - (1 - V) \rceil] = \alpha_R = \frac{\ell - x_0}{w}, \tag{129}$$

since $1 - V \sim \text{Uniform}(0, 1)$. Thus, the total expected number of stepping-out steps is

$$\mathbb{E}[N_{\text{out}} \mid x_0, \ell] = \mathbb{E}[k_L + k_R \mid x_0, \ell] = \frac{\ell}{w}. \tag{130}$$

In particular, this does not depend on $x_0$ and hence $\mathbb{E}[N_{\text{out}} \mid \ell] = \frac{\ell}{w}$, as well.

We now consider the distribution of the overshoot $a$. Let $z = \frac{x_0 - U}{w} = c - V$ with $c := \frac{x_0}{w}$ fixed. The number of left steps is $k_L = \lceil z \rceil$. The overshoot is then given by $a = k_L w - (x_0 - U) = w(\lceil z \rceil - z)$. For non-integer $z$, $\lceil z \rceil - z = 1 - \{z\}$, where $\{z\}$ is the fractional part (and the event that $z$ is an integer has probability 0 since $V$ is continuous). Since $V$ is Uniform$(0, 1)$, the random variable $\{c - V\}$ is Uniform$(0, 1)$

since translation modulo 1 and reflection preserve the uniform distribution. Hence, $1 - \{z\}$ is Uniform$(0, 1)$. Therefore, $a = w \cdot$ Uniform$(0, 1) \sim$ Uniform$(0, w)$. By symmetry, we also have $b \sim$ Uniform$(0, w)$. Note that $a$ and $b$ are dependent random variables, however they both have marginals Uniform$(0, w)$.

In the shrinkage step, we repeatedly propose a point uniformly in the current bracket. If the proposal lies outside the slice interval, the corresponding bracket end is moved to the proposal. This is repeated until a point in the slice $S$ with length $\ell = |S|$ is found. Choose coordinates such that the slice is $S = [0, \ell]$. Let the current bracket be $I_0 = [-a, \ell + b]$ with $a \geq 0$ on the left and $b \geq 0$ on the right and length $T = |I_0| = \ell + a + b$. Define $F(a, b)$ to be the expected number of shrinkage proposals starting from the bracket $I_0$.

In the algorithm, at each step, we draw $X$ uniformly on the current bracket. If $X \in [0, \ell]$, accept and stop (corresponding to a cost of 1). If $X < 0$, move the left endpoint to $X$. If $X > \ell$, move the right endpoint to $X$. Then this procedure is repeated with the updated bracket.

Condition on the first draw $X \sim$ Uniform$(-a, \ell + b)$. If $X \in [0, \ell]$, we stop with cost 1. If $X < 0$, the new left endpoint is $u = -X \in (0, a]$ while the right endpoint remains $b$. If $X > \ell$, the new right endpoint is $v = X - L \in (0, b]$ while the left endpoint remains $a$. Therefore, the following recursive relation holds:

$$F(a, b) = 1 + \mathbb{E}[1_{\{X < 0\}} F(-X, b) + 1_{\{X > L\}} F(a, X - \ell)] \tag{131}$$

$$= 1 + \frac{1}{T} \left[ \int_{-a}^{0} F(-x, b) \mathrm{d}x + \int_{\ell}^{\ell+b} F(a, x - \ell) \mathrm{d}x \right] \tag{132}$$

$$= 1 + \frac{1}{T} \left[ \int_{0}^{a} F(u, b) \mathrm{d}u + \int_{0}^{b} F(a, v) \mathrm{d}v \right], \tag{133}$$

where we changed variables to $u = -x$ and $v = x - \ell$ in the last line. This is an integral equation to be solved for $F$.

By symmetry between $a$ and $b$, we use an additive ansatz

$$F(a, b) = 1 + g(a) + g(b). \tag{134}$$

Substituting into eq. (133) gives

$$\int_{0}^{a} F(u, b) \mathrm{d}u = a + \int_{0}^{a} g(u) \mathrm{d}u + a g(b), \tag{135}$$

$$\int_{0}^{b} F(a, b) \mathrm{d}v = b + b g(a) + \int_{0}^{b} g(v) \mathrm{d}v. \tag{136}$$

Thus,

$$T g(a) + T g(b) = a + b + \int_{0}^{a} g(u) \mathrm{d}u + \int_{0}^{b} g(v) \mathrm{d}v + a g(b) + b g(a). \tag{137}$$

Rearranging by grouping terms which depend only on $a$ and $b$ respectively, and noting that each group of terms must be equal to a constant $C$ by separation of variables,

$$(\ell + a) g(a) - a - \int_{0}^{a} g(u) \mathrm{d}u = C, \tag{138}$$

$$(\ell + b) g(b) - b - \int_{0}^{b} g(v) \mathrm{d}v = -C. \tag{139}$$

The boundary condition $F(0, 0) = 1$, which is obtained by substituting $a = b = 0$ into eq. (133), implies that $g(0) = 0$. Substituting $a = 0$ into eq. (138) forces $C = 0$. Therefore the two equations for $a$ and $b$ reduce to a single integral equation for $g$ with $x \geq 0$:

$$(\ell + a) g(x) - x - \int_{0}^{x} g(u) \mathrm{d}u = 0. \tag{140}$$

Differentiating both sides with respect to $x$ gives $g'(x) = \frac{1}{\ell+x}$ and hence $g(x) = \ln(\ell+x) + C'$ for some constant $C'$. Applying the boundary condition $g(0) = 0$ finally gives $g(x) = \ln\left(1 + \frac{x}{\ell}\right)$. Hence, the explicit solution is

$$F(a,b) = 1 + \ln\left(1 + \frac{a}{\ell}\right) + \ln\left(1 + \frac{b}{\ell}\right). \tag{141}$$

We now show that the solution is unique, i.e. any solution equals the one given in eq. (141). Suppose we have another solution $K$. Define the difference $H(a,b) := K(a,b) - F(a,b)$. Subtracting the two instances of eq. (133) for $K$ and $F$ gives the equation

$$H(a,b) = \frac{1}{\ell + a + b}\left[\int_0^a H(u,b)\mathrm{d}u + \int_0^b H(a,v)\mathrm{d}v\right]. \tag{142}$$

For $s \geq 0$, define

$$M(s) := \sup\{|H(a,b)| : a, b \geq 0 \text{ and } a + b \leq s\}. \tag{143}$$

Note that $M(s)$ is nondecreasing in $s$: Let $0 \leq s_1 \leq s_2$ and define the index sets $R(s) := \{(u,v) \in [0,\infty)^2 : u + v \leq t\}$. Then $R(s_1) \subset R(s_2)$ because if $u + v \leq s_1$ and $s_1 \leq s_2$, then $u + v \leq s_2$ as well. By definition, this implies

$$M(s_1) = \sup\{|H(u,v)| : (u,v) \in R(s_1)\} \leq \sup\{|H(u,v)| : (u,v) \in R(s_2)\} = M(s_2). \tag{144}$$

Fix $s \geq 0$ and $a, b \geq 0$ with $a + b = s$. From eq. (142),

$$|H(a,b)| \leq \frac{1}{\ell + s}\left[\int_0^a |H(u,b)|\mathrm{d}u + \int_0^b |H(a,v)|\mathrm{d}v\right]. \tag{145}$$

For $u \in [0,a]$, the pair $(u,b)$ has the sum $u + b \leq a + b = s$. Similarly, for $v \in [0,b]$, the pair $(a,v)$ has the sum $a + v \leq s$. By definition of $M(s)$, this implies $|H(u,b)| \leq M(s)$ and $|H(a,v)| \leq M(s)$. Therefore,

$$|H(a,b)| \leq \frac{1}{\ell + s}[aM(s) + bM(s)] = \frac{s}{\ell + s}M(s). \tag{146}$$

Now take eq. (146) and allow $s$ to vary over $[0,t]$. Since $M(s)$ is nondecreasing in $s$ and the function $t \mapsto \frac{t}{\ell+t}$ is nondecreasing for $t \geq 0$, it follows that for any $(a,b)$ with $a + b \leq t$ (i.e. with $s := a + b \leq t$),

$$|H(a,b)| \leq \frac{s}{\ell + s}M(s) \leq \frac{t}{\ell + t}M(t). \tag{147}$$

Taking the supremum over all $(a,b)$ with $a + b \leq t$ yields

$$M(t) \leq \frac{t}{\ell + t}M(t). \tag{148}$$

Since $\frac{t}{\ell+t} < 1$ for every $t \geq 0$, this inequality forces $M(t) = 0$. Hence $H(a,b) = 0$ for all $a,b$ with $a + b \leq t$. By arbitrariness of $t$, we conclude $H(a,b) = 0$ for all $(a,b) \in [0,\infty)^2$. Thus, $K(a,b) = F(a,b)$ for all $a,b \geq 0$ and $F$ is the unique solution.

We now compute the expected number of proposals in the shrinkage procedure, unconditional on $a$ and $b$. Recall that we previously showed that the expected number of proposals conditional on the initial bracket $[-a, \ell + b]$ is given by eq. (141). Therefore, conditioning on $\ell$ and using the previously computed marginal

distributions of $a$ and $b$,

$$\mathbb{E}[N_{\text{shrink}} \mid \ell] = \mathbb{E}[F(a, b) \mid \ell] \tag{149}$$

$$= 1 + \mathbb{E}\left[\ln\left(1 + \frac{a}{\ell}\right) \mid L\right] + \mathbb{E}\left[\ln\left(1 + \frac{b}{\ell}\right) \mid L\right] \tag{150}$$

$$= 1 + 2\mathbb{E}\left[\ln\left(1 + \frac{T}{\ell}\right)\right] \qquad\qquad (\text{where } T \sim \text{Uniform}(0, \text{w})) \tag{151}$$

$$= 1 + \frac{2}{w}\int_0^w \ln\left(1 + \frac{t}{\ell}\right) \mathrm{d}t \tag{152}$$

$$= 1 + \frac{2}{w}\left[(\ell + w)\ln\left(1 + \frac{w}{\ell}\right) - w\right] \tag{153}$$

$$= 1 + 2\phi\left(\frac{w}{\ell}\right), \tag{154}$$

where $\phi(u) := \frac{(1+u)\ln(1+u)-u}{u}$.

Putting the above together, the expected total cost (stepping-out + shrinkage) conditional on $\ell$ is

$$\mathbb{E}[N_{\text{evals}} \mid \ell] = \mathbb{E}[N_{\text{out}} \mid \ell] + \mathbb{E}[N_{\text{shrink}} \mid \ell] = \frac{\ell}{w} + 1 + 2\phi\left(\frac{w}{\ell}\right). \tag{155}$$

$\square$

*Remark* 1. This does not depend on the geometry of the domain beyond $\ell$. It is valid for any compact set as long as we sample from the connected component containing the current point and use the randomised stepping-out + shrinkage procedure. For nonconvex sets, the algorithm still samples from the component containing the start. $\ell$ should be interpreted as that component's length along the chosen line. If we wanted to include other components on the line, costs and correctness would change.

**Theorem 2** (Optimal slice width in a fixed slice)**.** *Consider Hit-and-Run Slice Sampling with fixed width parameter $w > 0$ on a single connected slice interval of deterministic length $\ell > 0$. Then, the expected number of likelihood evaluations per step conditional on $\ell$ is*

$$C_\ell(w) := \mathbb{E}[N_{\text{evals}} \mid \ell] = \frac{\ell}{w} + 1 + 2\phi\left(\frac{w}{\ell}\right), \tag{156}$$

*where $\phi(u) = \frac{(1+u)\ln(1+u)-u}{u}$. Moreover, $C_\ell(w)$ has a unique minimiser $w_*$ given by*

$$w_* = u_* \ell, \tag{157}$$

*where $u_* > 0$ is the unique solution to*

$$u_* - \ln(1 + u_*) = \frac{1}{2}, \tag{158}$$

*equivalently,*

$$u_* = -1 - W_{-1}\left(-e^{-3/2}\right) \approx 1.357676674. \tag{159}$$

*Proof.* The expression eq. (156) is eq. (125) from theorem 1. Set $u := w/\ell$ and rewrite

$$C_\ell(w) = c(u) := \frac{1}{u} + 1 + 2\phi(u), \tag{160}$$

so that the minimisation over $w > 0$ is equivalent to minimisation over $u > 0$.

We compute derivatives. Writing $\phi(u) = \frac{(1+u)\ln(1+u)-u}{u}$, a direct calculation gives

$$\phi'(u) = \frac{u - \ln(1 + u)}{u^2}. \tag{161}$$

Therefore,

$$c'(u) = -\frac{1}{u^2} + 2\phi'(u) = \frac{-1 + 2(u - \ln(1+u))}{u^2}. \tag{162}$$

Hence $c'(u) = 0$ if and only if $u$ satisfies eq. (158).

To show uniqueness, define $g(u) := u - \ln(1+u)$ for $u \geq 0$. Then $g(0) = 0$ and

$$g'(u) = 1 - \frac{1}{1+u} = \frac{u}{1+u} > 0 \quad \text{for } u > 0, \tag{163}$$

so $g$ is strictly increasing on $(0, \infty)$. Since $g(u) \to \infty$ as $u \to \infty$, the equation $g(u) = \frac{1}{2}$ has a unique solution $u_* > 0$. Moreover, from eq. (162), the sign of $c'(u)$ is the sign of $-1 + 2g(u)$. Thus $c'(u) < 0$ for $u < u_*$ and $c'(u) > 0$ for $u > u_*$, so $u_*$ is the unique global minimiser of $c$. Consequently the unique minimiser of $C_\ell(w)$ is $w_*(\ell) = u_*\ell$.

For the Lambert-$W$ expression, eq. (158) is equivalent to

$$\ln(1+u) = u - \frac{1}{2} \quad \Longleftrightarrow \quad (1+u)e^{-u} = e^{-1/2}. \tag{164}$$

Let $y := 1 + u$. Then $ye^{-y} = e^{-3/2}$, i.e. $(-y)e^{-y} = -e^{-3/2}$. By definition of the Lambert-$W$ function, $-y = W(-e^{-3/2})$. The relevant solution has $y > 1$ and hence uses the $W_{-1}$ branch, yielding eq. (159). $\qquad\square$

*Remark* 2. The constant $u_*$ is the optimal ratio $w/\ell$ for a deterministic slice length $\ell$. In contrast, the constant $\kappa_\infty$ in theorem 3 is defined by minimising the *unconditional* expected cost $\mathbb{E}[N_{\text{evals}}]$, which averages over random chord lengths $\ell$ through $R_d = \ell/\mathbb{E}[\ell]$. The two constants differ because $\mathbb{E}[\phi(w/\ell)] \neq \phi(w/\mathbb{E}[\ell])$ in general.

**Theorem 3** (Optimal slice width in an ellipsoid). *Let $E_d = \{x \in \mathbb{R}^d : x^T A_d x \leq 1\}$ be an ellipsoid with $A_d \in \mathbb{R}^{d \times d}$ a symmetric positive definite matrix and define $\mu_d = \frac{1}{d}\text{Tr}(A_d)$. Consider a sequence of such ellipsoids $(E_d)_{d \geq 1}$ indexed by dimension. Let the spectrum of each matrix in the corresponding sequence $(A_d)_{d \geq 1}$ satisfy the boundedness assumption in lemma 9.*

*Suppose we run Hit-and-Run Slice Sampling in $E_d$. Then, the optimal choice of $w$, given by the minimiser of the expected cost $\mathbb{E}[N_{\text{evals}}]$, is*

$$w_* = 4\kappa_\infty \sqrt{\frac{2}{\pi \mu_d d}}[1 + o(1)], \tag{165}$$

*where $\kappa_\infty \approx 1.3035$ and $o(1)$ is a function such that $\lim_{d \to \infty} o(1) = 0$.*

*Proof.* The unconditional expected cost $C(w)$ for a single global $w$ is given by

$$C(w) := \mathbb{E}[N_{\text{evals}}] = \mathbb{E}[\mathbb{E}[N_{\text{evals}} \mid \ell]] = \frac{\mathbb{E}[\ell]}{w} + 1 + 2\mathbb{E}\left[\phi\left(\frac{w}{\ell}\right)\right], \tag{166}$$

using theorem 1. Taking the derivative gives

$$C'(w) = -\frac{\mathbb{E}[\ell]}{w^2} + \frac{2}{w^2}\mathbb{E}\left[w - L\ln\left(1 + \frac{w}{\ell}\right)\right] \tag{167}$$

which is zero when

$$G(w) := -\mathbb{E}[\ell] + 2\mathbb{E}[w - L\ln\left(1 + \frac{w}{\ell}\right)] = 0. \tag{168}$$

Note that $G(0) = -\mathbb{E}[\ell] < 0$ and $G(w) \to \infty$ as $w \to \infty$. Also, $G'(w) = 2\mathbb{E}\left[\frac{w}{\ell+w}\right] \in (0, 2)$, so $G$ is strictly increasing. Therefore, a unique root $w_*$ to eq. (168) exists.

eq. (168) can equivalently be written as the fixed-point equation

$$w_* = \frac{1}{2}\mathbb{E}[\ell] + \mathbb{E}\left[L\ln\left(1 + \frac{w_*}{\ell}\right)\right]. \tag{169}$$

Define the dimensionless optimality constant $\kappa_d$ via $w_* = \kappa_d \mathbb{E}[\ell_d]$. Substituting into the fixed-point equation for $w_*$ yields a fixed-point equation for $\kappa_d$,

$$\kappa_d = \frac{1}{2} + \mathbb{E}[R_d \ln\left(1 + \frac{\kappa_d}{R_d}\right)], \tag{170}$$

where we defined the normalised chord length $R_d := \frac{\ell}{\mathbb{E}[\ell]}$.

By lemma 11, $\kappa_d \to \kappa_\infty$ where $\kappa_\infty$ is the unique solution to

$$\kappa_\infty = \frac{1}{2} + \mathbb{E}[R_\infty \ln\left(1 + \frac{\kappa_\infty}{R_\infty}\right)]. \tag{171}$$

Equivalently, $\kappa_d = \kappa_\infty + o(1)$ as $d \to \infty$.

Combining this with the large $d$ expansion of $\mathbb{E}[\ell_d]$ in lemma 10 gives

$$w_* = \kappa_d \mathbb{E}[\ell_d] \tag{172}$$

$$= 4\kappa_\infty \sqrt{\frac{2}{\pi \mu_d d}}[1 + o(1)]. \tag{173}$$

To compute $\kappa_\infty$, we numerically solve eq. (171) by fixed-point iteration. For this, recall that $R_\infty = \frac{Q^{1/2}}{\mathbb{E}[Q^{1/2}]}$, where the distribution of $Q$ is given in lemma 6 and $\mathbb{E}[Q^{1/2}] = 2\sqrt{\frac{2}{\pi}}$ (lemma 8). The integrand is then written as

$$f(\kappa) := \mathbb{E}\left[R_\infty \ln\left(1 + \frac{\kappa}{R_\infty}\right)\right] = \frac{1}{\mathbb{E}[Q^{1/2}]} \int_0^\infty \sqrt{s} \ln\left(1 + \frac{\kappa \mathbb{E}[Q^{1/2}]}{\sqrt{s}}\right) f_S(s)\mathrm{d}s, \tag{174}$$

where $f_S$ is the Gamma$(\frac{3}{2}, 2)$ density, given by eq. (21), and iterating

$$\kappa_{\infty,n+1} = \frac{1}{2} + f(\kappa_{\infty,n}), \tag{175}$$

for any initial guess $\kappa_{\infty,0} \geq 0$. This yields $\kappa_\infty \approx 1.3035$. $\qquad\square$

*Remark* 3. Since the expectation is taken over the target distribution (in our case, the uniform distribution on the domain), the global geometry of the level set enters via the mean chord length $\mathbb{E}[\ell]$. The previous result (theorem 1) only depends on a particular slice, i.e. is valid locally.

*Remark* 4. Evidently, the optimal width in an ellipsoid scales as $O(d^{-1/2})$. We only needed the constant leading order term from the geometry of the ellipsoid (lemma 9) and the square-root scaling is a result of the concentration of the probability mass in a high-dimensional ball (lemma 8). This can be seen from the proof of lemma 8, wherein only results for the ball were used (i.e. the proof is independent of $A$).

For completeness, we specialise the above theorem to the case of the ball in the following

**Corollary 1** (Optimal slice width in a ball). *For a $d$-dimensional ball of radius $R$, the optimal slice width is given by*

$$w_* = 4\kappa_\infty R\sqrt{\frac{2}{\pi d}}[1 + o(1)], \tag{176}$$

*where $\kappa_\infty$ is given in theorem 3.*

*Proof.* This follows from theorem 3 applied to $A_d = \frac{1}{R^2}I$ with $I$ the identity matrix and $\mathrm{Tr}(A_d) = \frac{d}{R^2}$. Note that the spectrum of $A_d$ satisfies the boundedness assumption. $\qquad\square$

### E.2 Numerical tests

### E.2.1 Optimal scaling

In this section, we test the theoretically derived expressions for the computational cost of slice sampling (theorem 1) and for the optimal slice width $w_*$ (theorem 3).

To test the former, we consider the simplest example of slice sampling, namely sampling the interval $[0, \ell]$ uniformly for $\ell = 10$. Numerically, we first pick a random starting point $x_0 \sim \text{Uniform}(0, \ell)$. We then run the stepping out and shrinkage procedures according to Neal (2003) and record the total number of steps. This allows us to compare directly with the theoretical prediction in eq. (125).

To test the latter, we consider three different ellipsoids. Firstly, the unit ball, secondly the unit ball but with half of the axes shrunk by a factor of 10 (denoted by the semi-axis $s = 0.1$) and thirdly the unit ball but with half of the axes shrunk by a factor of 100 (denoted by the semi-axis $s = 0.01$). Note that these ellipsoids, when extended to a sequence indexed by dimension $d$ (i.e. for each $d$, shrink half of the axes of the unit ball by the appropriate factor), satisfy the boundedness condition required for theorem 3 to be applicable, i.e. the eigenvalues are always bounded uniformly in dimension and do not diverge to infinity or zero. For each of these ellipsoids we use the theoretically obtained eq. (165) to calculate $w_*$. For the numerical test, we scan a range of $w_*$ and calculate the computational cost by simulating slice sampling, and thus numericlly find the optimum $w_*$.

For both tests, the numerics agree with the theory, within the statistical error.

### E.2.2 Variance of the expected cost

In this section, we numerically analyse the standard deviation of the number of steps of Hit-and-Run Slice Sampling. The standard deviation is taken across i.i.d. samples in the domain under considertation, as well asl i.i.d. samples of the direction and any other source of randomness in the algorithm.

fig. 12 shows the standard deviation for both the ball and cube and anisotropic versions thereof. Importantly, we observe that the standard deviation remains of order 1 for a wide range of dimensions (here measured up to $d = 1000$). For the cube, the standard deviation increases with dimension, albeit slowly.

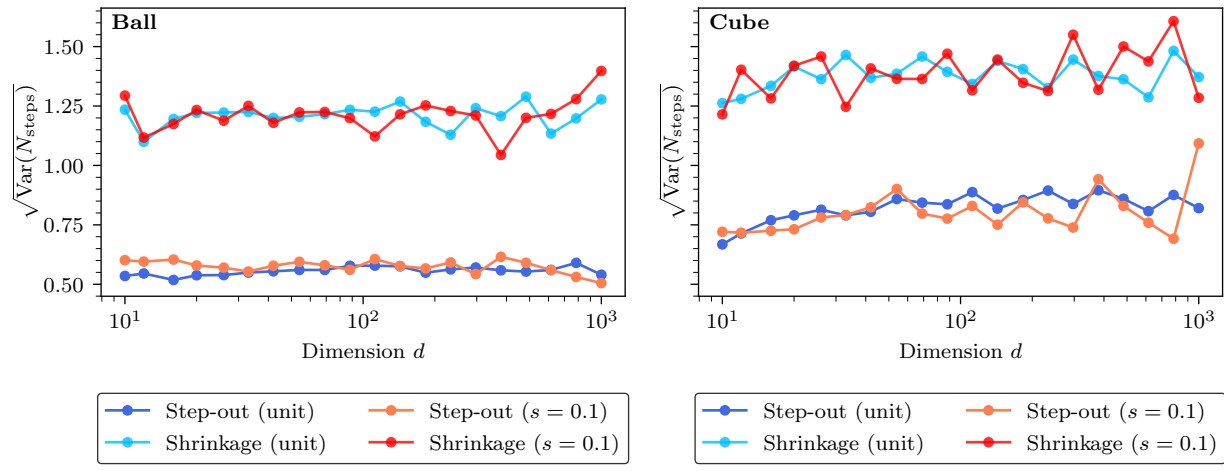

Figure 12: The standard deviation in the expected number of steps is of order 1, for a wide range of dimensions and anisotropies. Here, we compute the total number of steps of the stepping out and shrinkage procedures in slice sampling and plot the standard deviation from all source of randomness of the algorithm. **Left:** Results for ellipsoids (unit ball and with anisotropy introduced by shrinking half of the axes by a factor of 10). **Right:** Results for the cube (unit cube and with anisotropy introduced by shrinking half the axes by a factor of 10).

# F   Comparison of constrained path samplers

In this section we review and numerically compare constrained path samplers.

## F.1   Review of constrained samplers

Constrained path samplers implemented so far in the literature fall into two broad categories. Firstly, samplers which do not use gradients of the likelihood and secondly those which use gradients at the boundary to reflect the current position. Reflection-based algorithms are variants of the method described in Neal (2003).

As there is no empirical comparison of constrained samplers to date between these categories and since it is therefore unknown how individual constrained samplers scale with dimension, the following were compared:

- Galilean Monte Carlo (GMC) (Algorithm 3) Skilling (2012), which uses gradients outside the boundary to reflect a particle and reverses its velocity if the reflected particle is still outside the boundary,

- A variant of Galilean Monte Carlo published in 2019 (Galilean Monte Carlo (2019) (GMC-2019)) (Algorithm 4) Skilling (2019), which uses gradients inside the boundary and also reverses its velocity if the final point is rejected, however performs additional likelihood evaluations to remain in detailed balance,

- Reflective Slice Sampling (RSS) (Algorithm 5) Neal (2003), which is identical to Galilean Monte Carlo, except that the entire trajectory is rejected and the velocity re-randomised,

- Slice Sampling (SS) Smith (1984), which does not use gradient information and instead constructs a straight line in a randomly chosen direction using the "stepping out" and "shrinkage" procedures in Neal (2003), from which a point is then chosen uniformly.

GMC and SS are implemented in the software packages PYMATNEXT and POLYCHORD, respectively. Lemos et al. (2023) implement Langevin dynamics with a drift term and a reflection update of the velocity as in RSS.

Each sampler has hyperparameters which must be tuned for optimal performance, i.e. to decorrelate as fast as possible from the initial point. To tune the step size $\epsilon$, PYMATNEXT targets the acceptance rate along a trajectory, defined as the number of times a proposal is made inside the boundary divided by the trajectory length. The step size is increased or decreased by a factor of 1.25 until the acceptance rate lies in the range $[0.25, 0.5]$. The velocity is re-randomised after $L_{\text{traj}} = 8$ steps. The total trajectory length, $NL_{\text{traj}}$, is usually tuned manually with heuristics given in Pártay et al. (2021). In comparison, SS is self-tuning with respect to the width of the interval. POLYCHORD sets the number of velocity re-randomisations by default to $25n_{\text{dims}}$. In the following, the tuning of the step size in GMC and the POLYCHORD default setting for $N$ are adopted.

## F.2   Numerical comparison

Using each of the above samplers, the evidence $Z$ was calculated in $d$ dimensions with a uniform prior on the cube $[-r, r]^d$, where $r = 5.14$ is chosen as for the Rastrigin function Rastrigin (1974), and the test likelihood

$$L_\alpha(\boldsymbol{\theta}) = \alpha f(\boldsymbol{\theta}) + (1 - \alpha)g(\boldsymbol{\theta}), \tag{177}$$

where

$$f(\boldsymbol{\theta}) = \exp\left(-\frac{1}{2}|\boldsymbol{\theta}|^2\right), \tag{178}$$

$$g(\boldsymbol{\theta}) = \exp\left(-Ad + A\sum_{i=1}^{d}\cos(2\pi\boldsymbol{\theta}_i)\right), \quad A = 10. \tag{179}$$

---

**Algorithm 3** Galilean Monte Carlo (Skilling, 2012)

---

**Require:** Initial position $\mathbf{x}_0$, step size $\epsilon$, trajectory length $L_{\text{traj}}$, number of trajectories $N$

1: $\mathbf{x} \leftarrow \mathbf{x}_0$
2: **for** $n \leftarrow 1, N$ **do**
3:     Sample velocity $\mathbf{v} \sim \mathcal{N}(0, I)$
4:     **for** $i \leftarrow 1, L_{\text{traj}}$ **do**
5:         $\mathbf{x}_1 \leftarrow \mathbf{x} + \epsilon \mathbf{v}$
6:         **if** $L(\mathbf{x}_1) < L_\star$ **then**         ▷ If proposed point is outside, try to reflect
7:             $\hat{\mathbf{n}} \leftarrow \frac{\nabla L(\mathbf{x}_1)}{|\nabla L(\mathbf{x}_1)|}$
8:             $\mathbf{v}' \leftarrow \mathbf{v} - 2(\mathbf{v} \cdot \hat{\mathbf{n}})\hat{\mathbf{n}}$
9:             $\mathbf{x}_2 \leftarrow \mathbf{x}_1 + \epsilon \mathbf{v}'$
10:           **if** $L(\mathbf{x}_2) < L_\star$ **then**         ▷ If reflected point is also outside, go back
11:               $(\mathbf{x}, \mathbf{v}) \leftarrow (\mathbf{x}, -\mathbf{v})$
12:           **else**
13:               $(\mathbf{x}, \mathbf{v}) \leftarrow (\mathbf{x}_2, \mathbf{v}')$
14:           **end if**
15:         **else**
16:           $(\mathbf{x}, \mathbf{v}) \leftarrow (\mathbf{x}_1, \mathbf{v})$         ▷ Keep moving forward
17:         **end if**
18:     **end for**
19: **end for**

---

**Algorithm 4** Galilean Monte Carlo, 2019 variant (Skilling, 2019)

---

**Require:** Initial position $\mathbf{x}_0$, step size $\epsilon$, trajectory length $L_{\text{traj}}$, number of trajectories $N$

1: $\mathbf{x} \leftarrow \mathbf{x}_0$
2: **for** $n \leftarrow 1, N$ **do**
3:     Sample velocity $\mathbf{v} \sim \mathcal{N}(0, I)$
4:     **for** $i \leftarrow 1, L_{\text{traj}}$ **do**
5:         $N \leftarrow L(\mathbf{x} + \epsilon \mathbf{v}) \geq L_\star$
6:         **if** $N$ **then**
7:           $(\mathbf{x}, \mathbf{v}) \leftarrow (\mathbf{x} + \epsilon \mathbf{v}, \mathbf{v})$         ▷ Go North
8:         **else**
9:           $\hat{\mathbf{n}} \leftarrow \frac{\nabla L(\mathbf{x})}{|\nabla L(\mathbf{x})|}$
10:          $\mathbf{v}' \leftarrow \mathbf{v} - 2(\mathbf{v} \cdot \hat{\mathbf{n}})\hat{\mathbf{n}}$
11:          $E \leftarrow L(\mathbf{x} + \epsilon \mathbf{v}') \geq L_\star$
12:          $W \leftarrow L(\mathbf{x} - \epsilon \mathbf{v}') \geq L_\star$
13:          $S \leftarrow L(\mathbf{x} - \epsilon \mathbf{v}) \geq L_\star$
14:          **if** $S$ **and** ($E$ **and not** $W$) **then**
15:           $(\mathbf{x}, \mathbf{v}) \leftarrow (\mathbf{x}, \mathbf{v}')$         ▷ Aim East
16:          **else if** $S$ **and** ($W$ **and not** $E$) **then**
17:           $(\mathbf{x}, \mathbf{v}) \leftarrow (\mathbf{x}, -\mathbf{v}')$         ▷ Aim West
18:          **else**
19:           $(\mathbf{x}, \mathbf{v}) \leftarrow (\mathbf{x}, -\mathbf{v})$         ▷ Aim South
20:          **end if**
21:         **end if**
22:     **end for**
23: **end for**

---

The hyperparameter $\alpha \in [0, 1]$ increases the multimodality and thus the difficulty of the sampling problem. The evidence can be computed semi-analytically as

$$Z_\alpha = \frac{1}{(2r)^d} \left\{ \alpha \left[ \int_{-r}^{r} e^{-\frac{1}{2}x^2} \mathrm{d}x \right]^d + (1-\alpha)e^{-Ad} \left[ \int_{-r}^{r} e^{A\cos(2\pi x)} \mathrm{d}x \right]^d \right\}, \tag{180}$$

---

**Algorithm 5** Reflective Slice Sampling (Neal, 2003)

---

**Require:** Initial position $\mathbf{x}_0$, step size $\epsilon$, maximum trajectory length $L_{\mathrm{traj}}$, number of trajectories $N$

  1: $\mathbf{x} \leftarrow \mathbf{x}_0$
  2: **for** $n \leftarrow 1, N$ **do**
  3:      Sample velocity $\mathbf{v} \sim \mathcal{N}(0, I)$
  4:      $A \leftarrow$ True                          ▷ Accept the trajectory unless a proposed point steps outside twice
  5:      $\mathbf{x}_1 \leftarrow \mathbf{x}$                             ▷ $\mathbf{x}_1$ is the current position of the proposed trajectory
  6:      $i \leftarrow 1$
  7:      **while** $i \leq L_{\mathrm{traj}}$ **and** $A$ **do**
  8:          $\mathbf{x}_1 \leftarrow \mathbf{x} + \epsilon\mathbf{v}$
  9:          **if** $L(\mathbf{x}_1) < L_\star$ **then**
10:              $\hat{\mathbf{n}} \leftarrow \frac{\nabla L(\mathbf{x}_1)}{|\nabla L(\mathbf{x}_1)|}$
11:              $\mathbf{v}' \leftarrow \mathbf{v} - 2(\mathbf{v} \cdot \hat{\mathbf{n}})\hat{\mathbf{n}}$
12:              $\mathbf{x}_2 \leftarrow \mathbf{x}_1 + \epsilon\mathbf{v}'$
13:              **if** $L(\mathbf{x}_2) \geq L_\star$ **then**
14:                  $(\mathbf{x}_1, \mathbf{v}) \leftarrow (\mathbf{x}_2, \mathbf{v}')$
15:              **else**
16:                  $A \leftarrow$ False                    ▷ Reject trajectory and re-randomise velocity
17:              **end if**
18:          **end if**
19:          **if** $A$ **then**
20:              $\mathbf{x} \leftarrow \mathbf{x}_1$                                ▷ Accept trajectory
21:          **end if**
22:      **end while**
23: **end for**

---

and the remaining one-dimensional integrals can be performed numerically with quadrature.

Figure 13 shows the comparison in $d \in \{2, 4, 10, 30, 100, 300\}$ dimensions for 1000 live points. It is seen that SS remains correct for all dimensions, whereas all reflection-based samplers (GMC, GMC-2019, RSS), deviate as the dimensionality increases. In particular, GMC starts to systematically deviate in $d > 30$ dimensions, GMC-2019 as well but not as strongly and RSS already produces the wrong evidence in four dimensions. Moreover, the evidence becomes systematically negative for all $\alpha$. This indicates that at each Nested Sampling (NS) iteration, the density within the contour is not uniform but has an overdensity close to the boundary, leading to lower likelihood values on average, as explained in Kroupa et al. (2025). This is consistent with the fact that the deviation for GMC-2019 is smaller because the trajectories never leave the boundary and thus move the density further to the centre of the level set.

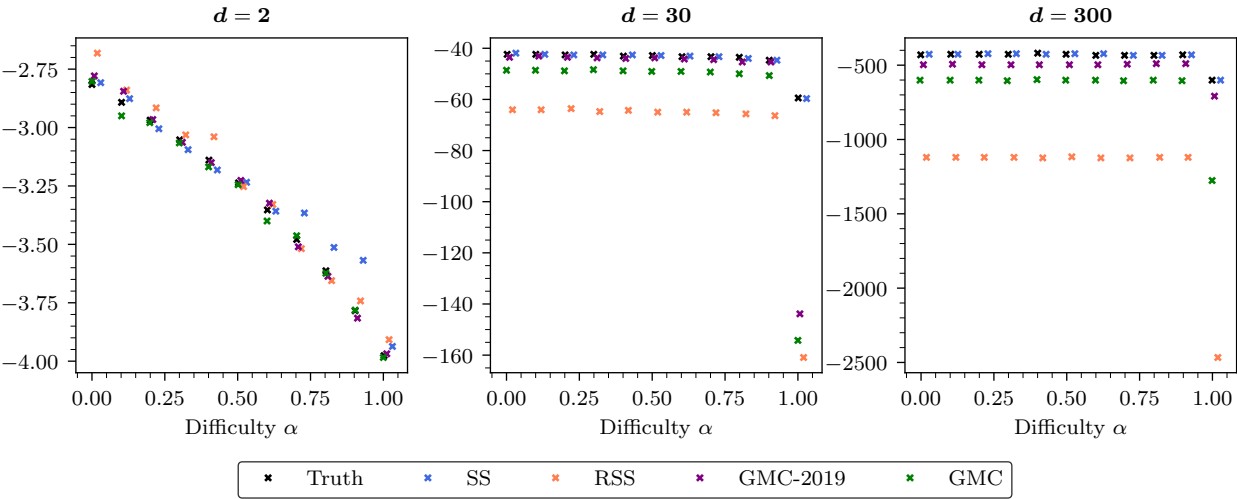

Figure 13: Evidence $\ln Z$ against difficulty hyperparameter $\alpha \in \{0, 0.1, \ldots, 0.9, 1\}$ for different constrained samplers. Each subplot shows a different parameter space dimensionality. The black crosses are the true $\log Z$ values which all samplers must agree with. Small horizontal shifts of the $\log Z$ values are purely for visualisation. The values $\alpha = 0$ and $\alpha = 1$ correspond to a unimodal Gaussian and a highly multimodal likelihood, respectively. Notably, Slice Sampling (SS) remains correct in high dimensions while all other algorithms start to deviate in dimensions $d$ larger than $\approx 30$. Slight vertical offsets of the plotted points are purely for visualisation purposes.

