# OpenReview forum: "Nested Slice Sampling: Vectorized Nested Sampling for GPU-Accelerated Inference"
_TMLR — Accepted by TMLR_

### Review · Reviewer_uaNS · 2026-02-26

**Summary Of Contributions:**

This paper presents Nested Slice Sampling (NSS), an accelerator-oriented implementation of Nested Sampling in the JAX ecosystem. The method combines a batched outer NS procedure with Hit-and-Run Slice Sampling (HRSS) as the constrained inner kernel, with the goal of making nested sampling more practical on SIMD hardware such as GPUs. A central contribution is the analysis of the slice-width hyperparameter for HRSS under ellipsoidal assumptions, which helps explain why the method can exhibit relatively stable per-step computational cost in vectorized settings. Empirically, the paper compares NSS against strong adaptive tempered SMC baselines, including gradient-based variants, and shows that NSS is competitive on multimodal synthetic benchmarks and Gaussian process hyperparameter marginalization tasks.

## Key Strengths

* **Theory and implementation are well connected.** I found it valuable that the paper does not only propose an implementation, but also provides theoretical analysis of the slice-width scaling and links this analysis to practical vectorization considerations. This connection between algorithmic design and systems-oriented efficiency is one of the strongest aspects of the paper.

* **The empirical evaluation is thoughtful and relevant.** The comparison against adaptive tempered SMC baselines, including a gradient-based variant such as SMC-HMC, is helpful for positioning the method. In particular, the experiments make it easier to understand where a gradient-free NS-based approach remains competitive and practically useful.

* **The software contribution seems useful to the community.** An open, accelerator-friendly NS implementation in JAX could be valuable for researchers and practitioners interested in Bayesian inference and evidence estimation, especially in settings where gradient information is unavailable, unreliable, or inconvenient to use.

## Key Weaknesses

* **The gap between the theoretical assumptions and the practical implementation could be discussed more clearly.** The slice-width analysis is derived under global ellipsoidal assumptions with bounded eigenvalues, whereas the implementation uses adaptive whitening based on a single empirical covariance. I was not fully sure how well the theoretical intuition carries over to highly nonlinear targets or to cases with strongly separated modes, where a single covariance estimate may become less informative.

* **The discussion of mixing uncertainty under batch deletion felt somewhat limited.** To obtain parallelism, the method deletes and duplicates (k) particles at once and then applies a finite number of HRSS updates. I may be missing something, but when (k) is large relative to the number of live points, and when the mutation chain is relatively short, it seems possible that substantial dependence between offspring particles could remain. If so, this could affect how confidently one should interpret posterior and evidence uncertainty estimates.

* **Some of the most useful intuition is deferred to the appendix.** In particular, the discussion of why anisotropy becomes less important in high dimensions, as well as the numerical validation of the slice-width theory, struck me as quite helpful for understanding the method. I think bringing part of this intuition into the main text would improve accessibility, especially for readers who are more interested in implementation and practice than in the full derivations.

**Audience:**

Yes

**Audience Explanation:**

I believe this paper would be of interest to at least a meaningful subset of the TMLR audience for several reasons.

First, scalable evidence estimation and robust posterior sampling for multimodal Bayesian models remain important problems in machine learning and scientific inference. A JAX-based nested sampling implementation that is explicitly designed for accelerator hardware is therefore likely to be of practical interest.

Second, the method is gradient-free, which makes it potentially attractive in settings involving black-box simulators, awkward constraints, or likelihood landscapes where gradient information is unavailable or not especially reliable. The comparisons against gradient-based and gradient-free SMC baselines also help readers understand the method’s practical positioning.

Third, the theoretical analysis of slice-width scaling and cost concentration in HRSS may also be of independent interest to readers working on sampling algorithms more broadly, beyond nested sampling specifically.

**Broader Impact Concerns:**

I do not have significant concerns regarding broader impact in this case. The paper proposes a general-purpose computational method for Bayesian inference and marginal likelihood estimation, and I did not identify a direct connection to sensitive data, downstream decision-making harms, or other application-specific ethical risks that would obviously require extended discussion in the present submission.

**Claims And Evidence:**

Yes

**Claims Explanation:**

Overall, I think the main claims are supported by a combination of theoretical analysis and empirical evidence. The paper gives a clear derivation of the slice-width scaling under its stated assumptions, and the experimental section provides meaningful comparisons against strong adaptive tempered SMC baselines, including gradient-based variants. In that sense, I found the central narrative of the paper convincing.

That said, I think some claims would benefit from slightly more explicit qualification. In particular, the theoretical guarantees rely on assumptions such as ellipsoidal geometry and bounded eigenvalues, and it would help to discuss more directly how the practical behavior may change once these assumptions are only weakly satisfied. I also think the uncertainty discussion could more clearly separate the geometric uncertainty that is analyzed in the paper from the additional dependence that may arise from batched deletion and finite-length mutation chains. These points do not undermine the main results, but clarifying them would make the claims more precisely bounded.

**Requested Changes:**

### 1. Discussion on limitations related to batch deletion and mixing

For parallelization, the proposed method deletes and duplicates particles in batches of size $k$, followed by a finite number of HRSS updates. While Appendix D.2 discusses geometric uncertainty, I was not fully sure how the paper recommends thinking about the additional dependence that may arise when many offspring are generated from a comparatively small number of parents. In particular, when $k$ is large and the mutation chain is relatively short, it seems plausible that residual dependence between offspring particles could remain non-negligible.

* **Actionable request:** Please consider adding a short quantitative discussion, caveat, or failure-mode analysis regarding how batch size and mutation-chain length may affect decorrelation and uncertainty estimation in practice.

### 2. Clarification of behavior when the ideal geometric assumptions are only weakly satisfied

The paper’s theoretical analysis provides useful intuition under idealized geometric assumptions, but I was not fully sure how directly this intuition transfers to the practical implementation, which relies on adaptive whitening from a single empirical covariance matrix.
The theory for the slice-width choice appears to rely on ellipsoidal assumptions with bounded eigenvalues, while the practical implementation uses adaptive whitening based on a single empirical covariance. I think the paper would be stronger if it discussed more explicitly how this approximation may degrade in highly nonlinear targets or in landscapes with widely separated modes.

* **Actionable request:** Please add a discussion clarifying how the adaptive whitening heuristic should be expected to behave when a single covariance matrix is a poor summary of the constrained region, and whether there are practical indicators that the theoretical intuition is becoming less reliable in such cases.

### 3. Impact of implementation caps in the JAX version

The JAX implementation introduces hard caps on the stepping-out and shrinkage procedures in order to guarantee bounded execution under JIT compilation. The manuscript notes that these caps were not reached in the reported experiments, which is reassuring. Still, for more difficult or unfamiliar targets, I think it would be helpful to clarify how one should interpret runs in which these caps are reached frequently.

* **Actionable request:** Please consider adding a short discussion of whether frequent cap hits could compromise the ideal slice-sampling behavior, and whether a runtime warning or diagnostic counter in the implementation would be useful for practitioners.

### 4. Integration of theoretical intuition and visual support into the main text

Some of the most intuitive and practically helpful explanations currently appear in the appendix, including the discussion that anisotropy becomes less important in high dimensions and the numerical validation of the slice-width theory.

* **Suggestion:** I would encourage the authors to summarize some of these intuitions in the main text, and possibly move one key figure (or a simplified version of it) from the appendix into the main paper. I think this would make the paper more accessible, especially to readers approaching it from the implementation or systems side.

### 5. Clearer guidance on when practitioners should prefer NSS

I appreciated that the experiments include SMC-HMC as a strong baseline, since this makes it easier to understand the practical positioning of NSS. At the same time, the paper could be even more useful if it more explicitly discussed the kinds of settings in which NSS should be expected to be a particularly good default choice, versus settings in which gradient-based alternatives may be preferable.

* **Suggestion:** Without requiring additional experiments, I think the Discussion section would benefit from a short “boundary of applicability” paragraph, for example addressing factors such as dimensionality, smoothness of the target, availability of gradients, and whether the application is more simulator-like or more amenable to gradient-based inference.

---

> ### Author Response · Authors · 2026-03-20
> **Response to requested changes**
>
> We thank the reviewer for their thorough comments and suggestions. We have implemented some minor changes as a result of these, adding some new information to relevant appendix sections, as well as promoting some key information from the appendix to the main text that we think helps the narrative. With regards to the specific requests for changes we have made the following alterations,
>
> 1. We have included an additional plot showing the estimate of the evidence error as a function of k, we promoted some key material from the appendix to Section 3.2, particularly addressing this point. We think this both strengthens the general claims around how close to embarassingly parallel our implementation manages to get. With the additional exploration of the error as a function of k in these experiments, we use this to give concrete guidance on setting the main hyperparameters of m and k (number of live points and size of batch deletion).
>
> 2. We have added a clarifying paragraph “Remarks on assumptions” addressing this point. This is included in section 3.3 alongside the promotion of some key summary of the theoretical tuning results from the appendix to the main text.
>
> 3. There are two caps, the stepping out cap is implemented as in Neal 2003 and the resulting algorithm is unbiased regardless of hitting this limit as shown in that prior work. The more problematic potential cap is the shrinkage, where we set a default of 100 max shrinkage steps. We have added text to highlight that if this is hit then something has gone seriously wrong (this is included in the appendix). The areas we have seen this actually get hit in practice are a. When the likelihood hits a flat peak (more likely to happen in reduced precision), in this case hitting the cap should trigger the end of the algorithm (i.e. include this alongside the compression based termination criteria), the result is still unbiased and the cap just serves as an augmented stopping criteria. b) when the likelihood consumes a random seed and is non-deterministic – this necessitates an extension (Which we have underway) to the procedure to work with noisy likelihood estimates. Hitting the cap in this case would result in a bias however this is true of vanilla versions of many MCMC algorithms that require a deterministic likelihood.
>
> 4. We have moved the intuitions behind the results from the appendix into the main text, as well as a figure, see the new section “Optimal slice width tuning”.
>
> 5. We expanded the discussion section to include more clear identification of what class of problems we think NSS in particular is performant and a good choice.

---

> ### Comment · Reviewer_uaNS · 2026-03-21
> **Response from Reviewer uaNS**
>
> Thank you for the detailed and constructive response. I appreciate that the authors addressed the main points in a concrete way and made corresponding revisions to the manuscript.
>
> I found the additional discussion and plot on the dependence of the evidence error on $k$ helpful. This clarifies the practical trade-off around batch deletion, and the added guidance on choosing $m$ and $k$ enhances the paper from a practitioner’s perspective.
>
> I also appreciate the addition of the new paragraph discussing assumptions and limitations. This helps clarify how the theoretical analysis should be interpreted relative to the practical implementation, which addresses one of my main concerns.
>
> The clarification regarding the implementation caps was also helpful. In particular, the distinction between the stepping-out cap and the shrinkage cap, and the discussion of the situations in which the latter may become problematic, addressed my concern on this point to a large extent.
>
> Finally, I think moving some of the key intuition and supporting material from the appendix into the main text improves the accessibility and overall narrative of the paper. I also found the expanded discussion of the problem settings in which NSS is likely to be a good choice useful.
>
> Overall, thank you again for the thoughtful response and revisions. I found the rebuttal helpful, and I believe the paper has improved meaningfully as a result.
>
> Sincerely,
>
> --Reviewer uaNS

---

### Review · Reviewer_bufA · 2026-03-02

**Summary Of Contributions:**

This paper introduces Nested Slice Sampling (NSS), with the focus on GPU-accelerated implementation of the nested sampling on batched particles. Its primary contributions are twofold:
1. The authors propose a batched, vectorized outer loop for Nested Sampling to enable massive parallelization. Specifically, this vectorizes the energy evaluation, thresholding, and resampling steps. For the inner loop---the mutation step that generates approximate particles from a constrained prior---they adopt Hit-and-Run Slice Sampling (HRSS).

2. HRSS traditionally does not fully benefit from vectorization due to the stochastic number of steps in the inner loop (which results in a synchronization penalty on SIMD hardware). The authors attempt to solve this via principled tuning of HRSS; they derive an "optimal slice width" that causes the computational cost of different chains to concentrate (meaning most chains take a similar number of steps). They provide a rigorous theoretical derivation for this optimal scaling result.

**Audience:**

Yes

**Audience Explanation:**

The paper sits at the intersection of Probabilistic Programming, Bayesian Inference, and Accelerator-Aware Algorithm Design. With increasing power of modern hardware, the need for robust, parallelizable evidence estimation tools is growing. Furthermore, the detailed theoretical analysis of HRSS is a standalone contribution to the MCMC literature that would interest researchers looking for better gradient-free samplers.

**Claims And Evidence:**

Yes

**Claims Explanation:**

The claims made in the submission are generally supported by clear and rigorous evidence. Specifically:

- Batch Parallelization Efficiency: The computational advantages of vectorizing NSS, when compared against legacy NS implementations, are demonstrated with clear empirical evidence in Appendix C.1.
- Optimal Slice Width Scaling: The theoretical scaling results for the optimal slice width are supported by rigorous mathematical derivations provided in the appendix.
- Robustness on Multimodal Targets: The claim that NSS is more robust than SMC on challenging multimodal problems is well-supported by the experimental results on the $10d$ and $20d$ Mixture of Gaussians (MoG) benchmarks.

**Requested Changes:**

Missing important benchmarks:

- Direct Comparison with JAXNS: The authors compare against UltraNest (a CPU-centric code) and show massive speedups. However, a direct comparison against jaxns, which is also JAX-based, would clarify if the performance gains come primarily from the HRSS kernel or the general use of JAX.

- Benchmarking against GPU-Accelerated AIS: A highly relevant competitor would be GPU-accelerated Annealed Importance Sampling (AIS), such as OAIS ( proposed in Optimised Annealed Sequential Monte Carlo Samplers, 2025, Syed, et.al.) Unlike SMC, AIS, is embarrassingly parallel because it drops the resampling step entirely, allowing independent chains to run in isolation across GPU threads.
This would be helpful to demonstrate whether the complex interacting particle system of NSS actually outperforms independent annealed chains on a GPU.

Missing citations and discussions:
- Dance et al. (2025) Efficiently Vectorized MCMC on Modern Accelerators. The authors explicitly discusses the "synchronization penalty" of running variable-length loops (like slice sampling) on SIMD hardware (GPUs) and try to resolve it through optimal tuning. Instead, Dance et al. (2025) tackle the exact same problem for the exact same algorithms in JAX, but offer a hardware-scheduling solution using Finite State Machines (FSMs) to decouple the chains. Omitting this leaves a major gap in the discussion of the current state-of-the-art for GPU MCMC.

- Add a substantial discussion addressing why AIS variants were excluded and how NSS's theoretical GPU scaling compares to embarrassingly parallel importance sampling methods.

---

> ### Author Response · Authors · 2026-03-20
> **Response to requested changes**
>
> We thank the reviewer for these helpful suggestions. In revision, we have included some new experimentation addressing comparison to embarrassingly parallel samplers, as well as better positioning how well parallelised the NSS algorithm is in the main text. With regards to the specific requests for changes we have made the following alterations,
>
> 1. We have run JAXNS and included it in an expanded appendix C.1. We note that jaxns doesn’t execute it’s chains in parallel so it is unlikely to be competitive without some specific tweaks (you could split a GPU into multiple devices but this is an anti-pattern). We chose Ultranest for this brief comparison as we can use jit compiled likelihoods executing on a hardware accelerator, and there is in theory a population slice sampler we can run, however we don’t find that these implementations are performant for the goal of rapid parallel implementation. We also promoted some material on parallelism in this appendix into the main text as it’s a rather punchy summary of the improvement we make, as well as sharpening our statement that we know of no other effective parallel implementation of the algorithm
> 2. This was a really interesting and useful prompt. We included all of this in a new appendix section A.2 (Adaptive non-equilibrium samplers and parallelism). NS is inherently adaptive and we implemented the experiments in the paper in a basic form (although it is simple enough to scan over many outer steps, typically outer kernel moves in SMC or NS are expensive enough that compiling a single outer kernel move is sufficient hardware utilisation). We included AIS as suggested, and consider this as the next logical step to include after our existing SMC-IRMH. This helps isolate the cost of resampling, and we also took the opportunity to examine the overhead cost of compiling a known temperature ladder as well. We think this nicely frames how much overhead there is. Even in the scenario of a simple Gaussian problem where the likelihood has minimal computation cost, so the overhead cost the sampling algorithm introduces is at it’s most problematic, we find this is still fairly minimal. We include a reference to OAIS as a potential motivation for future investigation.
> 3. We thank the reviewer for raising this paper to our attention. Our methods in NSS are compatible with the reformulation of slice sampling as a finite state machine, and their combination could potentially yield even larger speedups. We have included a reference in the section “Additional baselines”.
> 4. Thanks for pointing out this direction, We added a paragraph to 3.2 on GPU scaling and practical performance. Where we pull out the material from the appendix that we think best highlights how close to embarrassingly parallel we manage to get in the best case, as well as the previously mentioned experiment we included in A.2. Alongside the headline scaling we included some detail on how the variance of the estimates on normalizing constants degrades with higher $k$, and hence give some more clear concrete guidance on setting these hyperparameters.

---

> ### Comment · Reviewer_bufA · 2026-03-23
>
> Thank you for the detailed point-by-point response and for incorporating my comments into the revision. After reviewing the revised draft and the comments from the other reviewers, I find that my major concerns have been successfully addressed.
>
> However, a few minor but important points remain:
>
> - The current discussion of Dance et al. feels a bit superficial---it is only briefly mentioned in the appendix as a "means to soften the synchronization." Because the synchronization bottleneck of variable-length loops on SIMD hardware is a core motivation for this work, the existing architectural solution to this exact problem (FSMs) deserves to be treated more in det6ail, ideally in the main text's Sec 3.2 or Discussion section. (Additionally, please update this reference; Dance et al. is a published work, but the current draft cites the arXiv preprint).
>
> - While I appreciate the addition of the AIS benchmark, the current experimental setup does not fully capture its capabilities on modern accelerators. AIS performance depends heavily on operating in the large particle limit. Evaluating AIS with a population of only $m=1000$ particles is too small to reveal its true performance (for context, the OAIS framework typically scales to $N≥2^{14}$). A major innovation of the round-based tuning scheme in OAIS is that it completely avoids the global synchronization steps required by online annealing schedules, allowing it to leverage massive GPU parallelization. To be clear, the core contributions of this work remain entirely valid even if OAIS method demonstrates advantages in certain regimes. My request is simply that the text properly contextualizes these comparisons and acknowledges the limitations of benchmarking AIS at small particle counts.

---

> > ### Author Response · Authors · 2026-03-25
> > **Response to Official comment**
> >
> > Thank you for clarifying the points intended on these two items in particular.
> >
> > * We will fix the reference, apologies for that, and also do a general check for arXiv references where published versions now exist. In terms of the content of this paper, it is certainly relevant contemporary work, so thank you for prompting us to include it. I’ll include our preliminary take on it here, as this may help clarify our level of discussion, and the reason for including it in the appendix rather than the main text.
> >   (i) The speedup is fairly modest at best, for TESS on German Credit logistic regression, which is the closest setting to ours, the paper reports a speedup of 1.0x.
> >   (ii) It does not seem to discuss the case of short chains, which is the regime most relevant to SMC and nested sampling, where my understanding is that the returns would be even smaller.
> >   (iii) Our work goes to some lengths to show that HRSS with stepping-out has very low variance and a short tail in trajectory length. Fig. 6, for example, illustrates that we are likely quite far from the very long tail in NUTS trajectory length that can kill parallelisation. In particular, when targeting the constrained prior, which is approximately flat in general, my understanding is that this would further diminish the expected gain.
> >
> > * With regard to particle counts, we have been working extensively on downstream applications of this, and we noted a couple of these in the discussion, particularly in astrophysics, using neural surrogate models for physical processes and JAX-native versions of gravitational-wave likelihoods with massive data vectors. In both cases, the level of parallel throughput that can actually be achieved at the likelihood level is often a long way from the $2^{14}$-particle regime used in the OAIS paper. If you compare the runtime scaling at fixed compression (the experiment where we vary $m$ while fixing $k/m$) from Fig. 2 to Fig. 11, there is already something instructive there. Figure 2 (logistic regression) is a much cheaper, more parallel-friendly likelihood than Figure 11 (Gaussian process model). While one does not go out of memory on an A100 with parallel GP evaluations until around 10k particles in this case, one does start to see a linear trend in evaluation time well before that point. This is completely independent of the nested sampling or SMC setup, if one simply times likelihood batches while sweeping over $N$, the evaluation time starts to scale linearly. It is probably fair to say that GP models are not especially amenable to batch evaluation, for reasons including the need for fp64. This is a challenge we have found repeatedly when JAX-ifying analysis problems in applied sciences, not all likelihoods are amenable to taking the particle count asymptotic. In practice, I think the parallelism level we fix to is much more generally achievable for the kinds of problems we have in mind. Although on this MoG problem it is trivial to scale much further, doing so is not very instructive for the actual applications. I have also seen many AIS applications using on the order of 1000 particles, so this seems at least serviceable. We also sweep up to 4000 particles in these runtime-scaling illustrations and, echoing sentiments in the OAIS paper, our comment added in review was:
> >
> >   > “Based on these results we recommend setting $m$ as large as the available hardware allows (since additional particles are nearly free for fixed compression rate $k/m$), with $k/m \approx 0.1$ as a conservative default and $k/m \le 0.5$ as a practical upper bound before evidence accuracy degrades.”
> >
> > Lastly, this comparison to OAIS, and a more detailed discussion of SMC annealing schedules, probably does deserve more attention in follow-up work. From the nested sampling perspective one of the key features is robustness to phase transitions that are typically hard for annealing approaches, so a comparison on a statistical physics problem that exhibits this would be a really interesting direction. However, we are reasonably happy with the section we added, as it also gave us a good excuse to discuss the slightly more mundane issue of Python callback overhead, so we err on the side of leaving the discussion as is.
> >
> > Thanks again for the detailed discussion of these points.

---

### Review · Reviewer_Tee7 · 2026-03-10

**Summary Of Contributions:**

The authors propose a GPU implementation of Nested Slice Sampling (NSS) in blackjax and validate it against established samples, such as NUTS, with promising results.

However, at least to me, it was not sufficiently clear what were the concrete technical innovations the authors had to make to adapt samplers and what was directly inherited from existing nested sampling, slice sampling, and constrained MCMC ideas. As written, it is difficult to disentangle the paper’s original contributions, especially for readers who are not already deeply familiar with the SMC and MCMC literature and existing polemics therein.

A further practical concern is software and usability. The paper states that an open-source implementation is released and also says that the method is implemented in blackjax, but I could not find any code in the submission materials I reviewed, so it is hard to assess how straightforward the method is to use in practice, whether it will become a maintained part of blackjax, or whether it is instead a standalone research codebase. This matters because blackjax is pretty general and the paper’s practical impact depends on its integration with blackjax.

**Audience:**

Yes

**Audience Explanation:**

I am answering "Yes", because the main text is generally easy to follow and well written (and the question of "at least some readers" is logically trivial and requires > 1 interested readers). At the same time, there is a great deal of important material deferred to the appendix, including implementation details, uncertainty, tuning analysis, and experimental specifics. That balance made me wonder whether the paper may in fact be a **better fit for a (computational) statistics journal than for an ML venue**. In its current form, the work reads less like a clearly delimited ML-methods paper and more like a substantial computational statistics paper with strong implementation and commendable methodological and theoretical detail which unfortunately hides in the Appendix, but could be very interesting for statisticians.

**Claims And Evidence:**

Yes

**Claims Explanation:**

Partially, see requested changes.

**Requested Changes:**

- The authors should sharpen the contribution statement early and explicitly: what exactly is new algorithmically, what is an implementation contribution, and what is mainly synthesis? This issue is particularly noticeable because the paper itself acknowledges that many of the ingredients are known and frames the contribution as making them “practically GPU-viable” end-to-end. This pertains mainly to **Section 3**, but also **Section 2** should prime readers to what the problems of existing implementations are.
- It would be very helpful if authors added an **Algorithm** outlining their implementation in pseudocode.
- I don't understand why metrics changes from table from experiment to experiment. For instance, Table 1 and 2 show ESS/s, but Table 3 shows raw ESS values (which are rather impressive for the experiment). For experiment 3 which lacks a "gold standard ground truth", I would have liked to see simulation-based calibration (SBC, see Modrak et al., 2025) results or at least a measure of (expected) calibration error to actually show "well-calibrated uncertainty". This would also complement the current "single-dataset" experiments which a larger-scale, (synthetic) multi-dataset example. Furthermore, the comparatively low MMD values in Table 1 contrasting with the larger variability in W2 makes me wonder if a proper (multi-scale) kernel was used for the implementation and how useful MMD is for the comparisons (it doesn't seem to consistently differentiate between the samplers in nay of the experiments).
- On a related note, none of the benchmarks is actually "high-dimensional" according to modern ML standards. Please, correct me if I am wrong, but even though some dimensionalities are > 100 for **Inference Gym**, these problems factorize nicely into "hierarchical models". I would have liked to see something of equivalent difficulty to a Bayesian neural network (BNN) which is even showcased in the blackjax documentation or at least an argument from the authors against such a benchmark.
- Any reason why the authors opted for an arbitrary GP model as the "interesting example" instead of going for a more established set of benchmarks with already pre-computed ground-truth posterior samples, such as the open posteriordb (Magnusson et al., 2025, https://github.com/stan-dev/posteriordb)?
- Closely related, can the authors argue why something like bridge sampling or harmonic mean estimation from NUTS samples is not feasible for the GP experiment as a contender for evidence estimation?


**References**

1. Modrák, M., Moon, A. H., Kim, S., Bürkner, P., Huurre, N., Faltejsková, K., ... & Vehtari, A. (2023). Simulation-based calibration checking for Bayesian computation: The choice of test quantities shapes sensitivity. Bayesian Analysis, 20(2), 461.
2. Magnusson, M., Torgander, J., Bürkner, P. C., Zhang, L., Carpenter, B., & Vehtari, A. (2025). posteriordb: Testing, Benchmarking and Developing Bayesian Inference Algorithms. In International Conference on Artificial Intelligence and Statistics (pp. 1198-1206). PMLR.

---

> ### Author Response · Authors · 2026-03-16
>
> We thank the reviewer for the thoughtful comments on our work. We will provide a brief initial response to some of the higher level questions here. We will then follow up with more detailed changes once we have completed them.
>
> Firstly, we realise that it is hard to evaluate a submission that is intended to at least be in part an implementation contribution, without access to an implementation. The code is already public, however we weren’t aware of tools to anonymise a GitHub repo at submission time. The main implementation code is here:
> [https://anonymous.4open.science/r/blackjax-7354/README.md](https://anonymous.4open.science/r/blackjax-7354/README.md)
>
> With the contributions being `blackjax/mcmc/ss.py` (slice sampling), and the entirety of the `blackjax/ns/` folder. As BlackJAX is primarily lower-level kernel abstractions, we also include with this submission some experiments and examples of rolling both the BlackJAX NS kernels as well as competing SMC implementations here:
> [https://anonymous.4open.science/r/nss-D354/README.md](https://anonymous.4open.science/r/nss-D354/README.md)
>
>
> We are also sympathetic to the general sentiments around positioning, both of our contributions and in terms of what algorithms we compare to and what experiments we chose. We will give a brief candid description here and then follow up on the more specific points in revision.
>
> We would characterise the work as a significant amount of implementation work (we hope the code we link helps make this more transparent), a bit of algorithmic refinement (synthesising what works and doesn’t for parallel hardware) as well as theoretical exploration, and a bit of validation against similar algorithms. The aim of the experimentation that is included is twofold. Firstly to point out some synthetic examples with known pathologies where NS is performant, and secondly to extend this to some common data tasks that are often used to validate sampling algorithms (the growing literature on “neural samplers” also typically picks up on similar problems in similar dimensionality, which is perhaps more where we are spiritually aiming than scaling with dimension like NUTS). However, given the nature of the tasks, this experimentation is more validation that NSS keeps pace with similar GPU-ified algorithms, rather than broader claims of being SoTA on these tasks that would require much more experimental scrutiny.
>
> We will address the comments in revision and attempt to focus particularly on the framing and positioning.

---

> ### Author Response · Authors · 2026-03-20
> **Response to requested changes**
>
> We thank the reviewer for these helpful suggestions. In revision, we have made a number of focused changes to address the main concerns. We summarize these as follows (numbering attempting to follow listing in the original comment):
>
> 1. Clarifying the contribution and implementation focus.
> We have moved some of the theoretical tuning results from the appendix into the main text, and also promoted headline results on the vectorized implementation so that the practical contribution is visible earlier (in sections 3.2 and 3.3). We now clarify more explicitly that the main contribution is an end-to-end GPU-oriented formulation of nested sampling, and that this is, to the best of our knowledge, the first natively parallelized implementation of the full NS algorithm.
>
> 2. Adding explicit algorithmic description.
> We have added pseudocode for both the NS outer step and the generic SMC outer step in Appendix A.3. This is intended to make the implementation structure clearer and to better distinguish inherited ingredients from our concrete implementation choices.
>
> 3. Improving metric consistency and calibration language.
> For consistency with Tables 1 and 2, we have added an ESS/s column to the GP table. We have also softened the claim about “well-calibrated uncertainty” in the GP section, and instead reference simulation-based calibration (Modrák et al., 2025) as a useful complementary validation tool in settings without ground-truth posteriors. In addition, we added clarifying text in Sections 4.2 and 4.3 noting that broadly similar metrics across methods are expected here: these benchmarks are primarily intended to validate that NSS keeps pace with strong GPU-accelerated baselines, rather than to claim uniform superiority. Across these experiments, NSS posteriors remain broadly consistent with NUTS reference draws.
>
> 4. Positioning the benchmark scope more clearly.
> In some sense these are high dimensional but indeed this is subjective, to clarify what we mean more in the draft. We added a new paragraph to the discussion clarifying the broader motivation and positioning of the work, including expanded references to contemporary neural sampling papers that validate on similar classes of problems. We already signpost in the discussion that these are not intended as extreme high-dimensional benchmarks, and that slice-sampling-based methods have known scaling limitations in sufficiently challenging high-dimensional settings.
>
> 5. Clarifying the GP and other benchmark choices.
> We added the following motivation for the GP model: “Beyond its scientific interest, GP regression also serves as a useful proxy for models that place non-trivial computational load on the likelihood evaluation, making it a sharper test of GPU acceleration.” We also added a reference to PosteriorDB and note that it overlaps substantially with Inference Gym in benchmark construction and model families; our main reason for using Inference Gym is that it is implemented in JAX and therefore integrates more naturally with the hardware-accelerated sampling setting we target.
>
> 6. Discussing bridge sampling and related evidence estimators.
> We agree that bridge sampling and related post hoc evidence-estimation methods are relevant, and thank the reviewer for pointing this out. We now include references to these methods in the revised manuscript. We did not include them as primary baselines because our main focus is on sampling algorithm development: post hoc estimators can be very useful in practice, but they depend on the quality of the underlying posterior samples and therefore are not a direct like-for-like comparison with methods that jointly target posterior sampling and evidence estimation. More broadly, we agree that it would be interesting to study how these evidence estimates scale on higher-dimensional examples where NUTS remains effective. We view that as an important downstream model-comparison question, but somewhat separate from the present paper’s main goal, which is to develop and validate GPU-amenable nested sampling methodology.

---

> > ### Comment · Reviewer_Tee7 · 2026-03-23
> >
> > Thank you for the edits, which definitely help clarifying the positioning of the paper. I have read the other reviews and have a few small requests:
> >
> > - Ensure that all references and link render properly. For example, Modrak et al. is now not showing the year. Please, also ensure that figures are consistently references (currently, there are different versions, e.g., Figure, .fig, ...).
> > - Section 2 remains a jungle of background, theory, scattered related work, and design choices relevant to the proposed sampler. I suggest clearly delineating Background (the theoretical framework for nested slice sampling) and Related Work (other approaches to GPU-accelerated MC sampling / efficient nested slice sampling). For a non-expert reader, Section 3, among introducing clever scaling techniques, should make it obvious which components introduced in Section 2 (the background) were vectorized. Conversely, Section 2 should clearly introduce what are the cpu-bound components. Even at the heading level, the two sections should try communicate better.
> > - I maintain that it would greatly strengthen the results it the authors showed at least some calibration results for the GP example; after all, one advantage of a GPU-accelerated sampler should be that it makes cumbersome simulation-based validation of Bayesian samplers faster. I leave it up to the authors to agree or disagree with my point and just stick with a discussion.
> > - Table 1: how many modes does MoG ($D = 10$) have?

---

> > > ### Author Response · Authors · 2026-03-25
> > > **Response to comment**
> > >
> > > Thanks for this follow up, we have made some additional edits based on these comments that we hope refine the draft further:
> > >
> > > * We will do a pass through checking bib entries for arxiv vs published, biblatex bibtex clashes, cref vs Cref usage correctly, and hopefully catch the remainder.
> > > * We have made the following changes to try and increase the readibility of this that we hope helps:
> > >   - Renamed Section 2 to "Background and Theoretical Framework" to clarify its scope as setup rather than contributions
> > >   - Moved all theoretical analysis (optimal tuning, cost concentration, vectorization consequences) out of Section 2 and into Section 3.3, where it now lives alongside
> > >   the implementation. This was probably the largest change that we missed when adding content in the review process, and hopefully this adds a cleaner division between background and contributions.
> > >   - Added bridging text at the end of Section 2.2 identifying the adaptive control flow challenge and pointing forward to Section 3
> > >   - Revised Section 3 opening to clearly distinguish the generic vectorized NS framework from the specific NSS instantiation
> > >   - Clarified the parallelism model in Section 3.1: replacement kernels are synchronised at the likelihood level, executing in lockstep as a single batched operation
> > > * We agree that SBC is a valuable suggestion, and more broadly that it is perhaps a more natural validation tool for Bayesian inference than discrepancy metrics such as MMD. We did not include it in the current revision, but we now reference it explicitly and agree it would be a useful complementary check in future Bayesian validation studies. Our broader framework is phrased in terms of generic energy functions and partition function estimation, with an eye to applications beyond Bayesian inference where SBC is not applicable, but we agree that since the applications in the present paper are Bayesian and that this point is a bit of a weaker reason to not implement it!
> > > * We corrected the table/text to note that this problem uses 5 components. In the same pass, we clarified that, unlike the 2D MoG example, this experiment uses randomized covariances.

---

### Decision · Action_Editor_qML7 · 2026-05-03

**Recommendation:** Accept as is

**Additional Comments:**

All three reviewers support acceptance of the paper due to the reasons outlined above.

**Audience:**

Yes

**Audience Explanation:**

The paper can be of interest to TMLR's audience, especially for statisticians and researchers working on Bayesian inference and probabilistic computation. This is also acknowledged by the reviewers. Reviewer Tee7 highlights the paper's contributions in the Appendix could be a nice fit for a statistics journal.

**Claims And Evidence:**

Yes

**Claims Explanation:**

This paper makes Nested Sampling (NS) amenable to GPU parallelization, therefore enabling GPU acceleration for this method. The approach, called Nested Slice Sampling (NSS), combines an outer NS procedure with Hit-and-Run Slice Sampling as the inner kernel. The paper evaluates NSS against Sequential Monte Carlo (SMC)-based baselines, including gradient-based variants, concluding that NSS is competitive on multimodal synthetic benchmarks and Gaussian process hyperparameter marginalization. NSS is implemented within the JAX ecosystem.

All three reviewers agree that the claims made in the submission are supported by convincing and clear evidence. Specifically:
- The proposed sampler is theoretically sound.
- The theoretical claims (regarding optimal slice width) are supported by rigorous mathematical derivations.
- The experiments show clear computational advantages of NSS against legacy NS implementations, as well as improved robustness of NSS against SMC on challenging problems.
- The method is open-sourced, which facilitates reproduciblity and adoption.

Even though the method is not exempt of limitations that readers would need to consider, the reviewers find that these limitations are adequately acknowledged.